# Understanding Sharpness Dynamics in NN Training with a Minimalist Example: The Effects of Dataset Difficulty, Depth, Stochasticity, and More

Geonhui Yoo [1]  Minhak Song [2]  Chulhee Yun [1]

## Abstract

When training deep neural networks with gradient descent, sharpness often increases—a phenomenon known as *progressive sharpening*—before saturating at the *edge of stability*. Although commonly observed in practice, the underlying mechanisms behind progressive sharpening remain poorly understood. In this work, we study this phenomenon using a minimalist model: a deep linear network with a single neuron per layer. We show that this simple model effectively captures the sharpness dynamics observed in recent empirical studies, offering a simple testbed to better understand neural network training. Moreover, we theoretically analyze how dataset properties, network depth, stochasticity of optimizers, and step size affect the degree of progressive sharpening in the minimalist model. We then empirically demonstrate how these theoretical insights extend to practical scenarios. This study offers a deeper understanding of sharpness dynamics in neural network training, highlighting the interplay between depth, training data, and optimizers.

## 1. Introduction

Understanding the learning dynamics of neural network training is challenging due to the non-convex nature of its loss landscape. Recent empirical studies have highlighted that when training deep neural networks using gradient descent with step size $\eta$, sharpness often increases—a phenomenon known as *progressive sharpening*—and eventually hovers near $2/\eta$, a regime known as the *edge of stability* (Jastrzębski et al., 2019; 2020; Cohen et al., 2021).

The sharpness dynamics in deep learning have garnered significant interest in recent years, with several studies proposing mechanisms to explain its behavior (Ahn et al., 2022; 2023; Arora et al., 2022; Damian et al., 2023; Song & Yun, 2023; Wang et al., 2022; Zhu et al., 2023). Notably, Damian et al. (2023) introduce the *self-stabilization* mechanism, attributing the edge of stability to a negative feedback loop formed by the third-order term in the Taylor expansion of the loss. However, their analysis relies on the assumption that progressive sharpening occurs, i.e., that the sharpness tends to increase along the negative gradient direction.

Although commonly observed in practice, the underlying mechanisms behind progressive sharpening remain poorly understood. Analyzing this phenomenon is particularly challenging due to its dependence on various factors, including network architecture, training data, and optimizers. Cohen et al. (2021) conduct systematic experiments to investigate how these factors influence the degree of progressive sharpening, and we summarize their observations in Section 2.

In this work, we study progressive sharpening using a minimalist model: a *deep linear network* with a single neuron per layer. We show that this simple model effectively captures the sharpness dynamics observed in recent empirical studies. Furthermore, we empirically demonstrate that our theoretical findings from the minimalist model extend to practical scenarios. Our main contributions are summarized below:

- In Section 2, we identify key factors influencing progressive sharpening (dataset size, network depth, batch size, and learning rate) based on Cohen et al. (2021), and summarize them in Phenomenon 1.

- In Section 3, we propose a minimalist model that successfully replicates sharpness dynamics in deep learning, including the effects of key factors on progressive sharpening (Phenomenon 1) and behavior at the edge of stability.

- In Section 4 and Section 5, we provide a rigorous theoretical analysis of the minimalist model. We introduce the concept of *dataset difficulty* and derive bounds on the sharpness using this quantity. Furthermore, we show that the predicted sharpness from these bounds aligns well with empirical observations, even beyond our theoretical setup.

[1] KAIST AI [2] KAIST Math. Correspondence to: Chulhee Yun <chulhee.yun@kaist.ac.kr>.

*Proceedings of the 42nd International Conference on Machine Learning*, Vancouver, Canada. PMLR 267, 2025. Copyright 2025 by the author(s).

## 1.1. Related Works

The dynamics of sharpness during neural network training have garnered significant interest in recent years. Jastrzębski et al. (2019; 2020) observe that sharpness increases during the initial phase of training and step size influences sharpness along the optimization trajectory. Cohen et al. (2021) formalize the phenomena of *progressive sharpening* and the *edge of stability* through extensive controlled experiments, laying the groundwork for subsequent theoretical and empirical studies. In this subsection, we briefly mention some of the related works. Refer to Appendix A for more related works.

**Progressive Sharpening.**    Several recent works have explored the mechanisms behind progressive sharpening. Wang et al. (2022) employ the output-layer norm as a proxy for sharpness to explain progressive sharpening. In a two-layer linear network, they prove progressive sharpening under certain conditions, but their characterization is limited to a certain interval and does not specify the limit behavior of sharpness. Agarwala et al. (2023) analyze a quadratic regression model and showed progressive sharpening occurs at initialization. Rosenfeld & Risteski (2024) empirically observe that the training dynamics of neural networks are heavily influenced by outliers with opposing signals, suggesting that progressive sharpening arises due to these outliers. However, they do not quantify the degree of progressive sharpening or analyze its correlation with data properties. In addition, they restrict their theoretical analysis to synthetic data. Closely related to our work, Marion & Chizat (2024) study sharpness dynamics in deep linear networks and characterize the sharpness of solutions found by gradient flow. However, their analysis focus primarily on establishing an upper bound on sharpness, whereas our work provides both lower and upper bounds. Furthermore, we introduce the concept of *dataset difficulty* and demonstrate its correlation with these bounds.

**Sharpness Dynamics of SGD.**   While existing works on progressive sharpening and the edge of stability primarily focus on GD dynamics, several recent studies have analyzed how SGD differs from GD. For instance, SGD with a large step size has been observed to operate at a *stochastic edge of stability*, where sharpness stabilizes at a threshold smaller than that of GD (Lee & Jang, 2023; Agarwala & Pennington, 2024). Agarwala & Pennington (2024) analyze sharpness dynamics in SGD for a quadratic regression model, showing that SGD noise reduces sharpness increase compared to GD.

## 2. Key Factors of Progressive Sharpening

*Progressive sharpening* refers to the phenomenon where sharpness increases during gradient descent (GD) training. In this section, we discuss how the degree of progressive

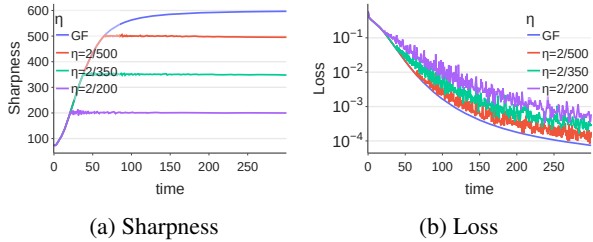

(a) Sharpness    (b) Loss

*Figure 1.* GF closely tracks GD dynamics before EoS. Sharpness of GF saturates as loss converges to zero. Sharpness of GD saturates as it enters the EoS regime. For experimental details, refer to Appendix C.1.

sharpening depends on problem parameters, based on the observations of Cohen et al. (2021). Their experiments examine the influence of factors such as network architecture and training data on the degree of progressive sharpening, which they quantify as the maximum sharpness along the gradient flow trajectory (GD with infinitesimal step size).

Notably, Cohen et al. (2021) observe that GD with a fixed step size $\eta$ closely follows the gradient flow trajectory until the sharpness approaches $2/\eta$ (see Figure 1). Once the sharpness reaches $2/\eta$, the training switches to the Edge of Stability (EoS) regime. In this regime, the GD trajectory deviates from the gradient flow trajectory and instead oscillates along the central flow trajectory (Cohen et al., 2025). Thus, gradient flow effectively represents *what GD would do if GD didn't have to worry about instability* (Cohen et al., 2021). Therefore, if we know the sharpness at convergence of GF, we can predict whether the same training of GD enters the EoS regime, based on the step size.

In Phenomenon 1, we summarize how problem parameters influence the degree of progressive sharpening, as observed in Cohen et al. (2021). We focus on the mean squared loss setting since, with cross-entropy loss, sharpness decreases at the end of training due to margin maximization, making comparisons less straightforward.

**Phenomenon 1** (Key factors of progressive sharpening)**.** The degree of progressive sharpening depends on the following problem parameters:

- **Dataset size**: Progressive sharpening occurs to a greater degree as the size of the training dataset increases. For example, when training on different-sized subsets of CIFAR10 using gradient flow, progressive sharpening is more pronounced with larger datasets (see Figure 2a).

- **Network depth**: Progressive sharpening occurs to a greater degree as network depth increases. For instance, when training fully-connected networks with fixed width and varying depth using gradient flow, deeper networks exhibit more pronounced progressive sharpening (see Figure 2b).

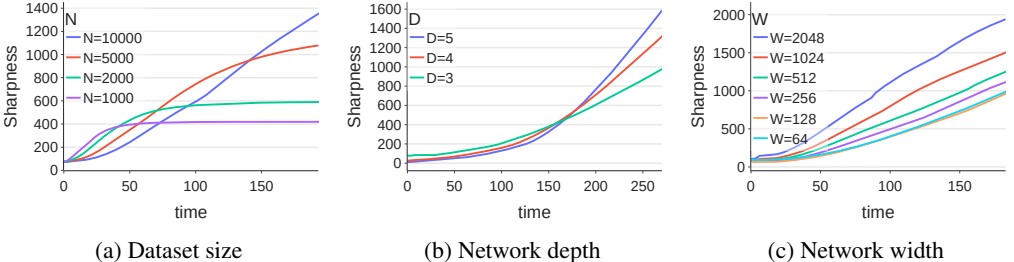

(a) Dataset size          (b) Network depth          (c) Network width

*Figure 2.* Effect of dataset size, network depth, and network width of tanh NN, for experimental details, refer to Appendix C.1.

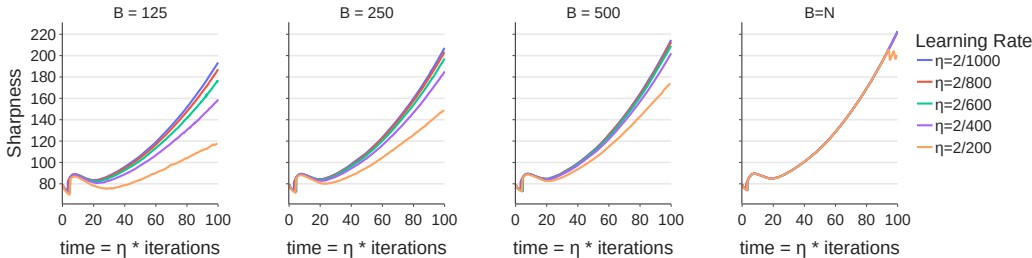

*Figure 3.* Effect of batch size, and learning rate in SGD and GD, for experimental details, refer to Appendix C.1.

- **Batch size**: Progressive sharpening occurs to a greater degree as the SGD batch size increases. When training with SGD, larger batch sizes lead to more pronounced progressive sharpening (see Figure 3).

- **Learning rate of SGD**: Progressive sharpening occurs to a lesser degree as the SGD learning rate increases, especially when the batch size is small (see Figure 3).

Cohen et al. (2021) also observe that wider networks exhibit less progressive sharpening with cross-entropy loss. However, our experiments suggest that this trend may not hold for squared loss, as shown in Figure 2c. Due to this discrepancy, we exclude the network width from the key factors of progressive sharpening.

## 3. Minimalist Model for Sharpness Dynamics

In this section, we introduce a *minimalist model*—a deep linear network with a single neuron per layer—that effectively captures the key characteristics of progressive sharpening (Phenomenon 1) as well as the edge of stability phenomenon observed in practical setups.

### 3.1. Problem Setup

**Notation.** For a positive integer $n$, we use $[n]$ to denote $\{1, \ldots, n\}$. For a vector $v$, $\|v\|$ denotes its Euclidean norm. For a matrix $A$, let $\|A\|_2$ detnoes its spectral norm, and $\mathrm{col}(A)$ and $\mathrm{row}(A)$ denote its column space and row space, respectively. For a symmetric matrix $M$, let $\lambda_{\max}(M)$ denote its maximum eigenvalue. For a linear subspace $S$ of

$\mathbb{R}^d$, we use $S^\perp$ to denote its orthogonal complement.

**Minimalist Model.** We consider a simple deep linear network $f : \mathbb{R}^d \to \mathbb{R}$, where each hidden layer consists of a single neuron with the identity activation function and $D \geq 2$ is the depth. The network is defined as

$$f(x; \theta) := (x^\top u) \prod_{i=1}^{D-1} v_i, \tag{1}$$

where $\theta = (u, v_1, \ldots, v_{D-1})$ represents the collection of all model parameters. Here, $u \in \mathbb{R}^d$ is the weight of the first layer, and $v_i \in \mathbb{R}$ is the weight of the $(i+1)$-th layer. We use $p := d + D - 1$ to denote the total number of parameters.

**Task.** We study the empirical risk minimization problem under the squared loss. Let the training dataset be defined by the data matrix $X = \begin{bmatrix} x_1^\top & \cdots & x_N^\top \end{bmatrix}^\top \in \mathbb{R}^{N \times d}$ and the label vector $y = \begin{bmatrix} y_1 & \cdots & y_N \end{bmatrix}^\top \in \mathbb{R}^N$, which satisfy $y \neq \mathbf{0}$. The loss function $L : \mathbb{R}^p \to \mathbb{R}$ is given by

$$L(\theta) := \frac{1}{2N} \sum_{i=1}^{N} (f(x_i; \theta) - y_i)^2 = \frac{1}{2N} \left\| Xu \prod_{i=1}^{D-1} v_i - y \right\|^2.$$

For convenience, we abbreviate the network output for the entire dataset as

$$f(X; \theta) := (Xu) \prod_{i=1}^{D-1} v_i,$$

and the residual vector as

$$z(\theta) := f(X; \theta) - y,$$

so we can write $L(\theta) = \frac{1}{2N} \|f(X; \theta) - y\|^2 = \frac{1}{2N} \|z(\theta)\|^2$.

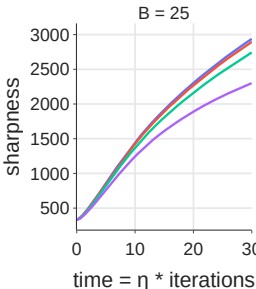
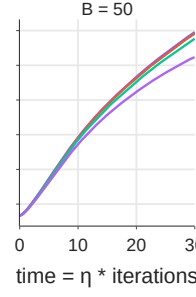
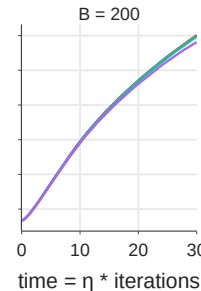
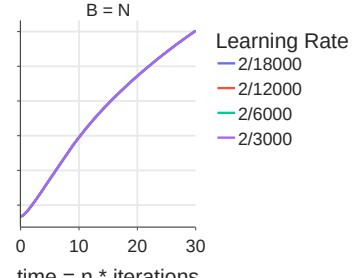

*Figure 4.* Effect of batch size and learning rate in our minimalist model ($D = 2$). We use a 2-label subset of $N = 1000$ from CIFAR10.

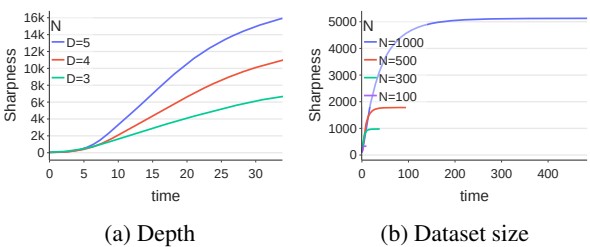

(a) Depth    (b) Dataset size

*Figure 5.* Effects of depth and dataset size in minimalist models. All experiments used a 2-label subset of $N = 1000$ from CIFAR10. In (b), except for $N = 1000$, runs terminated after $L(\theta) < 10^{-7}$.

**Optimizers.** We consider three optimization algorithms:

- Gradient Flow (GF): $\dot{\theta}(t) = -\nabla L(\theta(t))$,

- Gradient Descent (GD): $\theta^{(t+1)} \leftarrow \theta^{(t)} - \eta \nabla L(\theta^{(t)})$,

- Mini-batch Stochastic Gradient Descent (SGD):

$$\theta^{(t+1)} \leftarrow \theta^{(t)} - \eta \nabla \tilde{L}(\theta^{(t)}; P^{(t)}),$$

where we use $\eta$ to denote the learning rate (a.k.a. step size). For SGD, $\tilde{L}(\theta; P^{(t)})$ is the mini-batch loss at step $t$:

$$\tilde{L}(\theta; P^{(t)}) = \frac{1}{2B} \left\| P^{(t)}(f(X; \theta) - y) \right\|^2 = \frac{1}{2B} \left\| P^{(t)} z(\theta) \right\|^2,$$

defined by an independently sampled random diagonal matrix $P^{(t)} \in \mathbb{R}^{N \times N}$ with exactly $B$ diagonal entries chosen uniformly at random and set to 1 and the rest set to 0.

**Sharpness.** The primary goal of this paper is to understand how much the *sharpness* of the loss increases along the training trajectory. The quantity **sharpness** $S(\theta)$ at $\theta$ is defined as the maximum eigenvalue of the loss Hessian at $\theta$:

$$S(\theta) := \lambda_{\max}(\nabla^2 L(\theta)).$$

### 3.2. Minimalist Model Replicates Sharpness Dynamics

We now demonstrate that our minimalist model successfully replicates most of the interesting phenomena in the sharpness dynamics of neural network training. Here, we empirically showcase that the model not only reproduces the key observations made in Phenomenon 1, but also the self-stabilization dynamics in the edge of stability regime. Our observations with minimalist model suggest that this simple deep linear network could offer a useful testbed for understanding the sharpness dynamics in deep learning.

**Progressive Sharpening.** For Figure 4 and Figure 5a, we trained our minimalist model on a 2-label (cat vs dog) subset of CIFAR10 ($d = 3072$) with $N = 1000$, where labels are set to $\pm 1$ depending on the class. For Figure 5b, we changed $N$ for different runs. In Figure 4, we used random-reshuffling SGD for $B < N$ cases, and GD for $B = N$ cases. For Figure 5a and Figure 5b, we used Runge-Kutta 4 algorithm with adaptive step size of $\frac{1.0}{S(\theta)}$, where $S(\theta)$ is measured once every 50 iterations. In Figure 4 and Figure 5, we can observe the same trend described in Phenomenon 1 with our minimalist model: progressive sharpening happens to a greater degree with larger datasets, deeper networks, larger batch size, and smaller learning rate (when $B$ is small). While preserving the essential properties of progressive sharpening, the simplicity of our model makes it amenable to rigorous theoretical analyses, which we present in the subsequent sections.

**Edge of Stability.** Before we present further investigations into progressive sharpening, we discuss our empirical findings on the edge of stability regime here. To see whether our minimalist model also captures the characteristics of sharpness dynamics in this regime, we chose $X \in \mathbb{R}^{2 \times 2}$ and $y \in \mathbb{R}^2$ ($N = d = 2$) whose entries were sampled from the standard normal distribution. We trained a depth $D = 2$ minimalist model and observed the evolution of sharpness and training loss throughout the run of GD. For comparison, we also trained a Transformer on SST2 dataset (Socher et al., 2013). For more details, refer to Appendix C.3.

In Figure 6a and Figure 6b, we observe that the sharpness dynamics exhibited by the minimalist model closely resembles the typical sharpness curve at the edge of stability. The model first goes through progressive sharpening until the

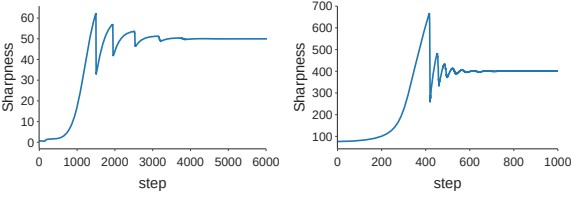

(a) Sharpness of minimalist    (b) Sharpness of Transformer

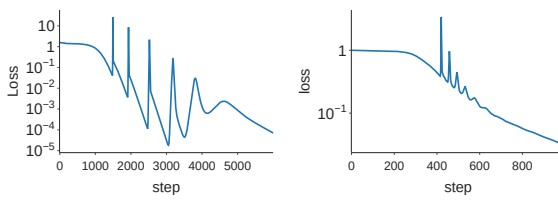

(c) Loss of minimalist      (d) Loss of Transformer

*Figure 6.* Our model captures typical "edge of stability" behaviors.

sharpness reaches $2/\eta$, and then the sharpness oscillates around the threshold $2/\eta$. Figure 6c and Figure 6d show the loss curve for the two models. As usually seen in the loss curves of practical models, the loss of our minimalist model decreases *non-monotonically* with *occasional spikes* during the edge of stability phase. Replicating this convergence behavior with a minimalist model is intriguing, because many existing studies based on other minimalist models fail to do so (Zhu et al., 2023; Kreisler et al., 2023; Kalra et al., 2025).

In practical models, the magnitude of the sharpness oscillation around $2/\eta$ often decays over time, as can be seen in Figure 6b and also Figure 3 of Damian et al. (2023). To the best of our knowledge, the reason for this attenuated oscillation is not well-understood. Interestingly, we find that our minimalist model captures this characteristic as well (Figure 6a), suggesting its potential usefulness for deeper theoretical understanding. However, it looks quite challenging to analyze this decay theoretically, because we discovered that the sharpness at which the rapid sharpness drop starts in fact depends on *machine precision*, and high precision can even make the loss *blow up* instead of decaying with occasional spikes. See Appendix D for details.

## 4. Sharpness at Minimizer of Training Loss

In this section, we provide a theoretical characterization of sharpness $S(\theta^\star)$ for any given minimizer $\theta^\star$ of the minimalist model. We start by introducing some necessary notation.

We let $r := \mathrm{rank}(X)$, and denote the singular value decomposition (SVD) of the data matrix $X \in \mathbb{R}^{N \times d}$ as

$$X = \sum\nolimits_{i=1}^{r} \sigma_i e_i w_i^\top, \tag{2}$$

where $\sigma_1 \geq \ldots \geq \sigma_r > 0$ are the singular values, $e_i \in \mathbb{R}^N$ are the left singular vectors, and $w_i \in \mathbb{R}^d$ are the right singular vectors.

For simplicity of exposition, we assume without loss of generality that $y \in \mathrm{col}(X)$, which allows the model to attain zero training loss $L(\theta^\star) = 0$ at global minima.[1] The label vector $y$ can then be decomposed as

$$y = \sum\nolimits_{i=1}^{r} d_i e_i, \tag{3}$$

where $d_1, \ldots, d_r$ are scalars. Let $W := \mathrm{row}(X) = \mathrm{span}(w_1, \ldots, w_r)$, and define $\Pi_W$ as the projection onto $W$, and $\Pi_W^\perp$ as the projection onto its orthogonal complement $W^\perp$. Then, the first-layer weight $u \in \mathbb{R}^d$ can be decomposed as

$$u = \sum\nolimits_{i=1}^{r} o_i w_i + \Pi_W^\perp u.$$

Then, we can express the GF, GD, and SGD dynamics in terms of $\sigma_i, d_i, o_i$ for $i \in [r]$ and $v_1, \ldots, v_{D-1}$. A detailed derivation is provided in Appendix B.1. It is worth noting that the updates to $u$ occur only within the subspace $W$.

Now we introduce a key concept that determines the degree of progressive sharpening.

**Definition 4.1** (Dataset Difficulty)**.** The *difficulty* of a dataset $(X, y)$ is defined as

$$Q := \sum\nolimits_{i=1}^{r} \frac{d_i^2}{\sigma_i^2}.$$

Intuitively, the quantity $Q$ captures the overall difficulty for a model to perfectly fit the dataset. For illustration, let us temporarily consider the linear model: we would like to learn a vector $\beta \in \mathbb{R}^d$ that satisfies $X\beta = y$. From the decompositions in (2) and (3), it is straightforward to check that any solution $\beta^\star$ must satisfy $\beta^\star = \sum_{i=1}^{r} \frac{d_i}{\sigma_i} w_i + \Pi_W^\perp \beta^\star$. Hence, each $\frac{d_i}{\sigma_i}$ can be thought of as "distance to travel" for the model to achieve $e_i^\top (X\beta - y) = 0$. The squared sum of such distances thus captures the total amount of effort required to fit the entire dataset.

Notice that $Q$ is only dependent on the dataset, independent of any architecture or optimization algorithm. We will show later that not only $Q$ is useful in our theoretical analysis of minimalist models, but also $Q$ can be used to predict the degree of progressive sharpening in larger fully-connected networks with nonlinear activations. Table 1 shows average values of $Q$ computed for 2-label subsets of CIFAR10, SVHN and Google speech commands datasets. We see that $Q$ increases as $N$ increases.

---

[1]This is without loss of generality, because any $y$ can be decomposed into $y = y_\| + y_\perp$ with $y_\| \in \mathrm{col}(X)$ and $y_\perp \in \mathrm{col}(X)^\perp$. This then allows decomposing the loss into: $L(\theta) = \frac{1}{2N}\|Xu\prod_{i=1}^{D-1} v_i - y_\||^2 + \frac{1}{2N}\|y_\perp\|^2$. The second constant term becomes the global minimum value of $L(\theta)$.

## 4.1. Two-layer Linear Networks

We first consider the two-layer case, i.e., when $D = 2$. Here, we prove upper and lower bounds on the sharpness $S(\theta^\star)$ at a given solution $L(\theta^\star) = 0$ as a function of the *imbalance* between two layers of $\theta^\star$.

**Definition 4.2** (Layer Imbalance). The *layer imbalance* of the two-layer minimalist model ($D = 2$) with parameter $\theta = (u, v_1)$ is defined by

$$C(\theta) := \|\Pi_W u\|^2 - v_1^2 = \Big( \sum_{i=1}^r o_i^2 \Big) - v_1^2 \,.$$

As will be seen in Section 5, layer imbalance is a preserved quantity for the trajectory of GF, and a slowly increasing quantity for GD and SGD.

In the next theorem, we show that the sharpness of a global minimum $\theta^\star$ can be characterized using $C(\theta^\star)$ and $Q$.

**Theorem 4.3** (Sharpness at minimizer, two-layer case). *For a two-layer minimalist model (1) trained on a dataset $(X, y)$ with difficulty $Q$, the sharpness at a global minimizer $\theta^\star = (u^\star, v_1^\star)$ of $L(\theta)$ is bounded by*

$$\frac{1}{N} \left[ \sigma_1^2 (v_1^\star)^2 + \frac{d_1^2}{(v_1^\star)^2} \right] \leq S(\theta^\star) \leq \frac{1}{N} \left[ \sigma_1^2 (v_1^\star)^2 + \frac{\sum_{i=1}^r d_i^2}{(v_1^\star)^2} \right],$$

*where $v_1^\star$ satisfies*

$$(v_1^\star)^2 = \frac{\sqrt{C(\theta^\star)^2 + 4Q} - C(\theta^\star)}{2} \,.$$

*Specifically, if the layers are balanced, i.e., $C(\theta^\star) = 0$, then*

$$\frac{1}{N} \left[ \sigma_1^2 Q^{1/2} + d_1^2 Q^{-1/2} \right] \leq S(\theta^\star)$$
$$\leq \frac{1}{N} \left[ \sigma_1^2 Q^{1/2} + \Big( \sum_{i=1}^r d_i^2 \Big) Q^{-1/2} \right].$$

Theorem 4.3 provides bounds on the sharpness at a global minimizer $\theta^\star$ when the layer imbalance $C(\theta^\star)$ is known. We can easily observe that if $(v_1^\star)^2 \geq \sigma_1^{-1} \sqrt{\sum_{i=1}^r d_i^2}$, both the lower and upper bounds of $S(\theta^\star)$ increase with $(v_1^\star)^2$. Furthermore, $(v_1^\star)^2$ is an increasing function of the dataset difficulty $Q$ and a decreasing function of the layer imbalance $C(\theta^\star)$. Therefore, if $C(\theta^\star)$ increases while $(v_1^\star)^2 \geq \sigma_1^{-1} \sqrt{\sum_{i=1}^r d_i^2}$, then both the lower and upper bounds decreases. The condition $(v_1^\star)^2 \geq \sigma_1^{-1} \sqrt{\sum_{i=1}^r d_i^2}$ holds if and only if

$$C(\theta^\star) \leq \tilde{C} := \frac{\sigma_1 Q}{\sqrt{\sum_{i=1}^r d_i^2}} - \frac{\sqrt{\sum_{i=1}^r d_i^2}}{\sigma_1} \,.$$

Note that $\tilde{C}$ is again only dependent on the dataset, and Table 2 shows its average numerical value for CIFAR10 (see Table 8 for SVHN and Google speech commands). As

can be seen later in Figure 9, we observe in practice that $C(\theta^{(t)})$ is substantially smaller than $\tilde{C}$ throughout training. Thus, under the assumption $C(\theta^\star) \leq \tilde{C}$, we establish the following relationship:

**Remark 4.4.** The sharpness at a global minimizer $S(\theta^\star)$ increases with the largest singular value $\sigma_1$ of the data matrix, increases with dataset difficulty $Q$, and decreases with layer imbalance $C(\theta^\star)$.

This observation will play a crucial role in Section 5.2.

## 4.2. Deep Linear Networks

Now we extend our analysis to the general case, i.e., a minimalist model with arbitrary depth $D \geq 2$. For the deeper case, we focus on the case where all layers are balanced, for the sake of simplicity. To this end, we first introduce the following assumption.

**Assumption 4.5** (Balanced Layers). For a minimalist model of depth $D \geq 2$ with parameter $\theta = (u, v_1, \ldots, v_{D-1})$, we say the parameter $\theta$ is *balanced* if

$$\|\Pi_W u\| = |v_1| = \cdots = |v_{D-1}| \,.$$

The balancedness assumption is widely adopted in the deep linear network literature (Arora et al., 2018; 2019). In our setting, the key difference is that we consider the norm of the first-layer weight $u$ projected onto the subspace $W$.

**Theorem 4.6** (Sharpness at minimizer, general case). *For the minimalist model (1) of depth $D \geq 2$ trained on a dataset $(X, y)$ with difficulty $Q$, let $\theta^\star$ be a global minimizer of $L(\theta)$ that is balanced (Assumption 4.5). Then, the sharpness at $\theta^\star$ is bounded by*

$$\frac{1}{N} \left[ \sigma_1^2 Q^{\frac{D-1}{D}} + (D-1) d_1^2 Q^{-\frac{1}{D}} \right] \leq S(\theta^\star)$$
$$\leq \frac{1}{N} \left[ \sigma_1^2 Q^{\frac{D-1}{D}} + (D-1) \Big( \sum_{i=1}^r d_i^2 \Big) Q^{-\frac{1}{D}} \right].$$

In Theorem 4.6, the effect of depth is dependent on the scale of $Q$. If $Q > 1$, the increase of depth will result in higher sharpness, and vice versa if $Q < 1$. In datasets of practical size, $Q$ is usually much larger than 1, as Table 1 suggests.

## 5. Optimizers of Minimalist Models

So far, we have discussed how dataset difficulty and layer imbalance determine the sharpness at a global minimum of the training loss $L(\theta)$ in our minimalist model. In this section, we consider the trajectory of GF, GD, and SGD on $L(\theta)$. For GF, using the fact that the layer imbalance is a conserved quantity, we can characterize the amount of progressive sharpening until convergence. Through numerical experiments we show that this prediction aligns well with

*Table 1.* Average $Q$ for different dataset size $N$. Samples were taken from two labels (cat vs dog, 3 vs 5, yes vs no) of each dataset, with standard deviation in parenthesis.

| Name | $N = 100$ | $N = 300$ | $N = 1000$ |
|---|---|---|---|
| CIFAR10 | 0.22(0.04) | 1.70(0.16) | 44.44(3.32) |
| SVHN | 1.16(0.26) | 21.13(2.90) | 859.4(67.2) |
| Google speech | 0.26(0.04) | 1.67(0.18) | 26.34(2.35) |

*Table 2.* Average $\tilde{C}$ that minimizes upper bound of Theorem 4.3, computed from 2-label subsets of CIFAR10.

| $N = 100$ | $N = 300$ | $N = 500$ | $N = 1000$ |
|---|---|---|---|
| 6.0(1.2) | 47.4(5.3) | 155.9(15.4) | 1246.1(100.2) |

*Table 3.* Average $\hat{S}_D = \frac{\sigma_1^2}{N} Q^{\frac{D-1}{D}}$ computed from 2-label subsets of CIFAR10.

| $D$ | $N = 100$ | $N = 300$ | $N = 500$ | $N = 1000$ |
|---|---|---|---|---|
| 2 | 365(61) | 1017(91) | 1851(128) | 5243(296) |
| 3 | 283(53) | 1111(114) | 2465(202) | 9876(662) |
| 4 | 249(50) | 1161(127) | 2845(252) | 13555(983) |
| 5 | 231(48) | 1193(136) | 3101(287) | 16391(1244) |

*Table 4.* Average $\frac{D-1}{N} \left( \sum_{i=1}^{r} d_i^2 \right) Q^{-\frac{1}{D}}$ computed from 2-label subsets of CIFAR10.

| $D$ | $N = 100$ | $N = 300$ | $N = 500$ | $N = 1000$ |
|---|---|---|---|---|
| 2 | 2.18(0.18) | 0.77(0.04) | 0.43(0.02) | 0.15(0.01) |
| 3 | 3.36(0.19) | 1.68(0.05) | 1.13(0.03) | 0.56(0.01) |
| 4 | 4.42(0.18) | 2.63(0.06) | 1.96(0.04) | 1.16(0.02) |
| 5 | 5.46(0.18) | 3.60(0.07) | 2.84(0.05) | 1.87(0.03) |

the actual post-training sharpness value in both linear and *nonlinear* neural networks. For GD and SGD, we study how the discrete and stochastic nature of the algorithms affects the evolution of layer imbalance $C(\theta^{(t)})$, which has implications on the sharpness of final solutions (Remark 4.4).

### 5.1. Gradient Flow

We introduce following assumption for theoretical results.

**Assumption 5.1** (Gradient flow converges to a global minimum)**.** The gradient flow dynamics $\theta(t)$ converges to the solution $\theta(\infty) := \lim_{t \to \infty} \theta(t)$, where $L(\theta(\infty)) = 0$.
We introduce the following useful property of gradient flow:

**Lemma 5.2** (Conservation of layer imbalance under gradient flow)**.** *Let $\theta(t)$ be a gradient flow trajectory trained on the minimalist model* (1)*. Then,*

- *For $D = 2$, the layer imbalance remains constant along the gradient flow trajectory: $C(\theta(t)) = C(\theta(0))$ for all $t \geq 0$.*

- *For $D > 2$ and balanced initialization $\theta(0)$, $\theta(t)$ remains balanced for all $t \geq 0$.*

Combining Lemma 5.2 with Theorem 4.3 and Theorem 4.6:

**Corollary 5.3** (two-layer case)**.** *Consider a two-layer minimalist model* (1) *trained with the gradient flow. Under Assumption 5.1, the sharpness at convergence $S(\theta(\infty))$ satisfies the bounds of Theorem 4.3 with $C(\theta^\star) = C(\theta(0))$.*

**Corollary 5.4** (general case)**.** *Consider a minimalist model* (1) *trained with the gradient flow using balanced initialization (Assumption 4.5). Under Assumption 5.1, the sharpness at convergence $S(\theta(\infty))$ satisfies the bounds of Theorem 4.6.*

These corollaries show that, for GF, we can get the bounds of sharpness at convergence in Theorem 4.3 and Theorem 4.6 only based on the information of initialization, without the need to run GF. The bounds obtained for $S(\theta(\infty))$ align with the observation in Phenomenon 1 that larger datasets (larger $Q$) and deeper models (larger $D$ when $Q > 1$) induce stronger progressive sharpening.

Our theoretical results so far only characterize the sharpness at convergence. We now present theoretical results based on the initial weight assumption specifically for our two-layer minimalist model, thereby deriving theoretical bounds for both the initial sharpness and the converged sharpness.

**Assumption 5.5** ($\alpha\beta$ Initialization)**.** For a two-layer minimalist model with parameter $\theta = (u, v_1)$, we say the parameter $\theta$ uses $\alpha\beta$ *initialization* if $u^{(0)} \sim \mathcal{N}\left(0, \alpha^2 I_d\right)$ and $v_1^{(0)} \sim \mathcal{N}\left(0, \beta^2\right)$, where $\alpha \in \mathbb{R}^+$ and $\beta \in \mathbb{R}^+$.

Under Assumption 5.5, our analysis of the minimalist model can encompass many widely used weight-initialization techniques, such as those of Glorot & Bengio (2010), He et al. (2015), and LeCun et al. (2002).

We now present two theorems, one characterizing expected sharpness at initialization and the other at convergence.

**Theorem 5.6** (Initial Sharpness Bound with $\alpha\beta$ Initialization)**.** *Under the Assumption 5.5, for a two-layer minimalist model trained on a dataset $(X, y)$, the expected sharpness at the initialization $\theta^{(0)} = (u^{(0)}, v_1^{(0)})$ is bounded by*

$$\frac{\sigma_1^2}{N}\left(\alpha^2 + \beta^2\right) \leq \mathbb{E}\left[S(\theta^{(0)})\right]$$
$$\leq \frac{1}{N}\left[\sum_{i=1}^{r} \alpha^2 \sigma_i^2 + \beta^2 \sigma_1^2 + \sqrt{\sum_{i=1}^{r} \sigma_i^2 \left(\alpha^2 \beta^2 \sigma_i^2 + d_i^2\right)}\right].$$

Theorem 5.6 provides bounds of the sharpness at initialization under the Assumption 5.5. We can easily observe that both the lower and upper bounds of $\mathbb{E}[S(\theta^{(0)})]$ increase with $\alpha^2$ and $\beta^2$, meaning that the larger variance of the initialization scheme results in sharper initialization.

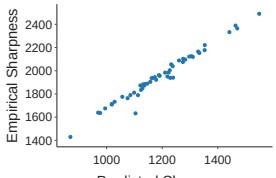 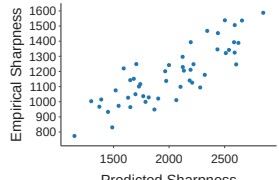

(a) identity activation, depth 5, width 2048 in CIFAR 2 label ($N = 300$, Correlation 0.99)

(b) tanh activation, depth 4, width 1024 in SVHN 2 label ($N = 100$, Correlation 0.81)

Figure 7. Correlation of $\hat{S}_D$ vs empirical $S(\theta(\infty))$, for more results, refer to Appendix C.5.

**Theorem 5.7** (Sharpness Bound at Convergence with $\alpha\beta$ Initialization). *Under the Assumption 5.1 and Assumption 5.5, for a two-layer minimalist model trained on a dataset $(X, y)$ with difficulty $Q$ and gradient flow, the expected sharpness at convergence $\theta^\star$ is lower bounded by:*

$$\mathbb{E}[S(\theta^\star)] \geq \frac{1}{2N}\left[\left(\sigma_1^2 + \frac{d_1^2}{Q}\right)\sqrt{(\mathbb{E}[C(\theta^\star)])^2 + 4Q} + \left(\frac{d_1^2}{Q} - \sigma_1^2\right)\mathbb{E}[C(\theta^\star)]\right]$$

*and upper bounded by:*

$$\mathbb{E}[S(\theta^\star)] \leq \frac{1}{2N}\left[\left(\sigma_1^2 + \frac{\sum_{i=1}^r d_i^2}{Q}\right)\sqrt{(\mathbb{E}[C(\theta^\star)])^2 + 2r\alpha^4 + 2\beta^4 + 4Q} + \left(\frac{\sum_{i=1}^r d_i^2}{Q} - \sigma_1^2\right)\mathbb{E}[C(\theta^\star)]\right],$$

*where $\mathbb{E}[C(\theta^\star)] = \mathbb{E}[C(\theta^{(0)})] = r\alpha^2 - \beta^2$*

These bounds are analogous to Theorem 4.3. The lower bound of Theorem 4.3 can be reparameterized as follows:

$$\frac{1}{N}\left[\sigma_1^2(v_1^\star)^2 + \frac{d_1^2}{(v_1^\star)^2}\right]$$
$$= \frac{1}{2N}\left[\left(\sigma_1^2 + \frac{d_1^2}{Q}\right)\sqrt{C(\theta^\star)^2 + 4Q} + \left(\frac{d_1^2}{Q} - \sigma_1^2\right)C(\theta^\star)\right].$$

The only difference between the lower bound of Theorem 5.7 and Theorem 4.3 is whether we consider the expectation or a specific value for $C(\theta^\star)$. For the upper bound, reparameterization shows similar results, except for the term $2r\alpha^4 + 2\beta^4$. Therefore, the same characterization we stated in Remark 4.4 can also be applied to Theorem 5.7.

**Experiments.** To verify if our theory provides a good prediction of the post-training sharpness, we numerically calculate the terms that appear in Theorem 4.6. We used 2-label (cat vs dog) subset of CIFAR10 ($d = 3072$) in Table 3 and Table 4. We randomly selected 50 datasets of size $N$, while ensuring a balanced distribution between labels. For each dataset, we computed the quantities and report the average and standard deviation. We also run the

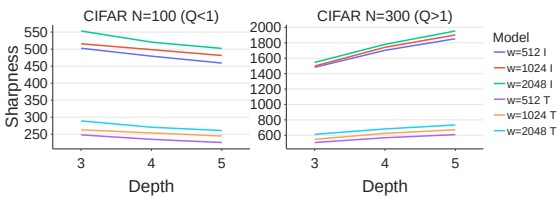

Figure 8. Depth vs Sharpness plot. "I" denotes identity activation, "T" denotes tanh activation, and "w" means width.

same experiments on SVHN and Google speech commands; see Appendix C.6. Based on Table 3 and Table 4, we can observe that the term $\frac{\sigma_1^2}{N}Q^{\frac{D-1}{D}}$ dominates both bounds, and the gap between the upper and lower bounds should be orders of magnitude smaller than $\frac{\sigma_1^2}{N}Q^{\frac{D-1}{D}}$. Therefore, we give a name for the quantity:

**Definition 5.8** (Predicted Sharpness). We define the *predicted sharpness* as the following: $\hat{S}_D := \frac{\sigma_1^2}{N}Q^{\frac{D-1}{D}}$.

For fully-connected networks of varying width, depth, activation, and dataset size, we trained the model 50 times using different random 2-label subsets of size $N$ from CIFAR10, SVHN, and Google speech commands. For each run, GF was randomly initialized using the default Pytorch initialization scheme. All training runs were terminated when $L(\theta(t)) < 10^{-6}$, and we treated the iterate at termination as $\theta(\infty)$.

Figure 7 shows the scatter plots of final sharpness $S(\theta(\infty))$ vs our predicted sharpness $\hat{S}_D$ on a 5-layer linear network of width 2048 trained on CIFAR10, and tanh-activated 4-layer network of width 1024 trained on SVHN. We emphasize that the predicted sharpness provides a reasonable estimate of the post-training sharpness, even though the experiment settings were different from our theory in a number of ways[2]: width was not fixed to 1, activation was nonlinear, and initialization was not balanced.

In Figure 8, we show that our theory captures the effect of dataset difficulty $Q$ and depth $D$. As expected from predicted sharpness, we observe different effects of depth $D$ on $S(\theta(\infty))$, depending on $Q > 1$ (when $N = 300$), and $Q < 1$ (when $N = 100$). More detailed results are deferred to Table 5 and Table 6 in Appendix C.5.

We detail experiments for Theorem 5.6 and 5.7 in Appendix C.7.

## 5.2. Gradient Descent and Stochastic Gradient Descent

For GD and SGD, we specifically focus on the case of $D = 2$ and address the last two parts of Phenomenon 1: how batch size and step size affect progressive sharpening.

---

[2]Due to this reason, the scale of empirical sharpness and predicted sharpness in Figure 7 doesn't match exactly.

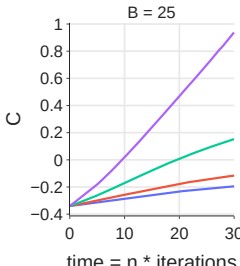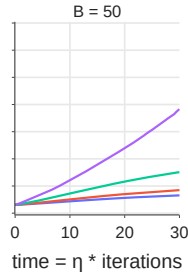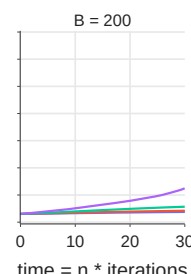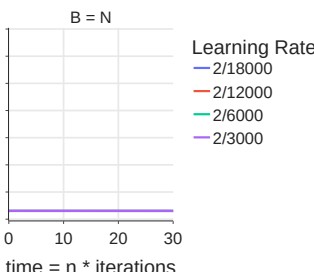

Figure 9. Effect of batch size and learning rate in minimal models in the dynamics of layer imbalance $C(\theta^{(t)})$, in $D = 2$ & $N = 1000$.

By Theorem 4.3, knowing $C(\theta^\star)$ at convergence determines the bound of sharpness. Therefore, we focus on the *change* of $C$ after an update of GD and SGD. Starting at the same point $\theta$, let $\theta_{\mathrm{GD}}^+$ and $\theta_{\mathrm{SGD}}^+$ be the parameters after one step of GD and SGD, respectively. In the theorem below, we show that a step of SGD incurs a greater increase of $C$ compared to GD. Together with Remark 4.4, Theorem 5.9 sheds light on why SGD induces less progressive sharpening.

**Theorem 5.9** (Increase of Layer Imbalance). *For the minimalist model* (1) *with $D = 2$, the following holds for the change of $C$ for an update of GD and SGD:*

$$C(\theta_{GD}^+) - C(\theta) = \tfrac{\eta^2}{N^2}[-\Psi_1(\theta)C(\theta) + \Omega_1(\theta)]$$

$$\mathbb{E}[C(\theta_{SGD}^+)] - C(\theta_{GD}^+) = \tfrac{\eta^2(N-B)}{BN^2(N-1)}[-(\Psi_2(\theta) - \Psi_1(\theta))C(\theta) + (\Omega_2(\theta) - \Omega_1(\theta))],$$

*where*

$$\Psi_1(\theta) := \sum_{i=1}^r \sigma_i^2(z(\theta)^\top e_i)^2,$$

$$\Psi_2(\theta) := N\sum_{i=1}^r \sigma_i^2\|z(\theta) \odot e_i\|^2,$$

$$\Omega_1(\theta) := \sum_i \sum_{j>i}[\sigma_i(z(\theta)^\top e_i)o_j - \sigma_j(z(\theta)^\top e_j)o_i]^2,$$

$$\Omega_2(\theta) := N\sum_i \sum_{j>i}\|\sigma_i(z(\theta) \odot e_i)o_j - \sigma_j(z(\theta) \odot e_j)o_i\|^2.$$

*Also, $\Psi_2(\theta) \geq \Psi_1(\theta) \geq 0$ and $\Omega_2(\theta) \geq \Omega_1(\theta) \geq 0$.*

The symbol $\odot$ in the theorem statement denotes the element-wise product. Theorem 5.9 shows that whenever $C(\theta) \leq \frac{\Omega_1(\theta)}{\Psi_1(\theta)} =: T_1(\theta)$, GD is guaranteed to increase $C$ and SGD increases $C$ even more whenever $C(\theta) \leq \frac{\Omega_2(\theta) - \Omega_1(\theta)}{\Psi_2(\theta) - \Psi_1(\theta)} =: T_2(\theta)$. In Figure 9, we visualize how $C(\theta^{(t)})$ evolves in our minimal models with a 2-label subset of CIFAR10 dataset. The plots are obtained from the same runs as Figure 4. First, we observe that $C(\theta^{(t)})$ keeps increasing. Also in Figure 21 and 22, we observe that $C(\theta^{(t)}) \leq T_1(\theta^{(t)})$ and $C(\theta^{(t)}) \leq T_2(\theta^{(t)})$ hold across all settings.

There are a few important implications of the theorem. Indeed, this theorem does not fully prove that SGD has larger value of $C(\theta^\star)$ at convergence and hence smaller final sharpness $S(\theta^\star)$ (Remark 4.4) than GD. Nevertheless, the theorem offers useful insights on how the batch size $B$ and step size $\eta$ affects the degree of progressive sharpening.

Note that $\mathbb{E}[C(\theta_{\mathrm{SGD}}^+)] - C(\theta_{\mathrm{GD}}^+)$ is more pronounced for smaller batch size $B$, highlighting the role of stochasticity. Smaller $B$ results on greater increase of $C$ in SGD, resulting in less progressive sharpening. Larger step size $\eta$ also amplifies this mechanism and leads to even smaller increase of sharpness. In Figure 23–26, we numerically calculate $\Psi_1(\theta^{(t)})$, $\Psi_2(\theta^{(t)})$, $\Omega_1(\theta^{(t)})$ and $\Omega_2(\theta^{(t)})$ values of GD and SGD, which offers helpful insights on how to interpret Theorem 5.9 and Figure 9. As for GD, although the increase of $C$ is proportional to $\eta^2$, its $\Psi_1(\theta^{(t)})$ and $\Omega_1(\theta^{(t)})$ are relatively small, so combined with the factors $\frac{\eta^2}{N^2}$ the increase becomes very small and largely unaffected by the step size. For SGD, $\Psi_2(\theta^{(t)})$ is far larger than $\Psi_1(\theta^{(t)})$, and likewise $\Omega_2(\theta^{(t)})$ greatly exceeds $\Omega_1(\theta^{(t)})$. Consequently, $C$ increases much more noticeably when $B$ is small and $\eta$ is large. This dynamics of $C$ nicely correlates with the degree of progressive sharpening in Figure 4.

## 6. Conclusion

Throughout this paper, we studied how problem parameters influence the sharpness dynamics of neural networks. We introduced a minimalist model that reproduces progressive sharpening and edge of stability behavior observed in practice. Theoretically, we derived sharpness bounds at both initialization and convergence as functions of problem parameters. Empirically, we showed these bounds are numerically tight and can predict convergence sharpness under gradient flow. For GD and SGD, we established a theorem that offers insights into how batch size and learning rate influence sharpness dynamics.

As a step toward broader generalization, Appendix E presents a preliminary theoretical analysis of networks with nonlinear activations. We believe that further extending the problem setup, such as varying the architecture, width, or optimizer, can help bridge the gap between theoretical understanding and empirical observations of progressive sharpening.

## Acknowledgements

This work was supported by two National Research Foundation of Korea (NRF) grants funded by the Korean government (MSIT) (No. RS-2023-00211352; No. RS-2024-00421203).

## Impact Statement

This paper presents work whose goal is to advance the field of Machine Learning. There are many potential societal consequences of our work, none which we feel must be specifically highlighted here.

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

# A. More Related Works

**Edge of Stability.** The edge of stability regime, where sharpness stabilizes near $2/\eta$, has been extensively studied in recent years (Ahn et al., 2022; Arora et al., 2022; Damian et al., 2023; Wu et al., 2023). Damian et al. (2023) propose the self-stabilization mechanism, attributing this phenomenon to a negative feedback loop arising from the third-order term in the loss's Taylor expansion. Recent theoretical works have further analyzed training dynamics under simplified models. For instance, Ahn et al. (2023) study a loss function of the form $(x, y) \mapsto \ell(xy)$, and Song & Yun (2023) extend these results to 2-layer linear networks. Similarly, Zhu et al. (2023) characterize the edge of stability in a 4-layer scalar network, and Kreisler et al. (2023) generalize this analysis to deep scalar networks.

Specifically, Kreisler et al. (2023) consider a scalar linear network with loss $\mathcal{L}(\mathbf{w})$, for depth $D \in \mathbb{N}$ and weights $\mathbf{w} \in \mathbb{R}^{\mathbf{D}}$. Their Theorem 3.2 shows that gradient descent does not increase the sharpness of the gradient flow solution initialized at GD iterates (referred to as GFS sharpness). Similarly, our Theorem 5.9 shows that $C(\theta)$ increases over time when mild conditions on $C(\theta)$ are satisfied. Together with our Remark 4.4, these results imply that GFS sharpness decreases as training progresses under GD/SGD in our minimalist model.

While these works provide valuable insights, they focus on training with a single data point, limiting their applicability to more general settings. In contrast, our minimalist model considers general training data, enabling us to capture how dataset properties influence sharpness dynamics.

**Linear Diagonal Neural Networks.** Our minimalist model shares similarities with diagonal linear networks in a sparse regression setting. Pesme et al. (2021) show that SGD leads to solutions with better generalization than GD. Similarly, our Theorem 5.9 shows that SGD induces less progressive sharpening than GD, leading to lower sharpness at convergence. Considering that lower sharpness correlates with improved generalization in diagonal linear networks (Nacson et al., 2022), they both unveil how stochasticity can help generalization.

**Connection between sharpness and generalization.** SAM (Foret et al., 2021) introduce the hypothesis that minimizing sharpness improves generalization, and its benefit is demonstrated in practical training. Moreover, GD with a large learning rate is shown to implicitly find flatter solutions (Cohen et al., 2021), which often generalize better than those obtained with small learning rates (Li et al., 2019). While these works suggest a correlation between sharpness and generalization, Andriushchenko et al. (2023) show that this relationship is data-dependent.

**Potential practical implications on learning rate scheduling.** The study by Zhu et al. (2024) highlights that the catapult mechanism contributes positively to model generalization, and catapults can be induced by designing a proper learning rate schedule. In light of this, predicting sharpness evolution can offer practical value when designing such schedulers.

# B. Technical Details

In this section, we present proofs of the main theorems and derivations of key formulas. For simplicity, we use the following notation: for a function $Z$ mapping model parameters to a scalar, vector, or matrix, we write $Z(t) := Z(\theta(t))$ for gradient flow (GF) and $Z^{(t)} := Z(\theta^{(t)})$ for gradient descent (GD) and stochastic gradient descent (SGD).

## B.1. Reparameterization of the Minimalist Model

In this subsection, we reparameterize the gradient flow (GF), gradient descent (GD), and stochastic gradient descent (SGD) dynamics for the minimalist model (1) in terms of $\sigma_i$, $d_i$, $o_i$ for $i \in [r]$, and $v_1, \ldots, v_{D-1}$, as introduced in Section 4. This reparameterization serves as the foundation for subsequent theoretical proofs.

**Network output and residual.** We can decompose network output $f(X;\theta)$ into $e_1, \ldots, e_r$ components by

$$f(X;\theta) = \sum_{i=1}^{r}(e_i^\top f(X;\theta))e_i = \sum_{i=1}^{r}\left(e_i^\top Xu \prod_{j=1}^{D-1} v_j\right)e_i = \sum_{i=1}^{r}\left(\sigma_i o_i \prod_{j=1}^{D-1} v_j\right)e_i\,, \tag{4}$$

where the last equality is obtained by replacing $X$ with $\sum_{i=1}^{r}\sigma_i(e_i w_i^\top)$. Similarly, we can decompose residual $z(\theta)$ by

$$z(\theta) = f(X;\theta) - y = \sum_{i=1}^{r}\left(\sigma_i o_i \prod_{j=1}^{D-1} v_j - d_i\right)e_i\,. \tag{5}$$

Hence, network output (4) and residual (5) can be reparameterized in terms of $\sigma_i$, $d_i$, $o_i$ for $i \in [r]$, and $v_1, \ldots, v_{D-1}$.

**Gradient Flow.** Recall that the loss at $\theta$ is $L(\theta) = \frac{1}{2N}\|f(X;\theta) - y\|^2 = \frac{1}{2N}\|z(\theta)\|^2$. The GF dynamics is given by

$$\dot{u}(t) = -\frac{\partial L}{\partial u}(t) = -\frac{1}{N}X^\top z(t)\prod_{j=1}^{D-1} v_j(t)\,, \tag{6}$$

$$\dot{v}_j(t) = -\frac{\partial L}{\partial v}(t) = -\frac{1}{N}z(t)^\top Xu(t)\prod_{q\neq j} v_q(t)\,, \quad \forall j \in [D-1]\,. \tag{7}$$

We can replace $X$ with $\sum_{i=1}^{r}\sigma_i(e_i w_i^\top)$ in (7) and obtain

$$\dot{v}_j(t) = -\frac{1}{N}\sum_{i=1}^{r}\sigma_i(z(t)^\top e_i)(w_i^\top u(t))\prod_{q\neq j} v_q(t) = -\frac{1}{N}\sum_{i=1}^{r}\sigma_i(z(t)^\top e_i)o_i(t)\prod_{q\neq j} v_q(t)\,,$$

for each $j \in [D-1]$. Hence, we have

$$\dot{v}_j(t) = -\frac{1}{N}\sum_{i=1}^{r}\sigma_i\left(e_i^\top z(t)\right)o_i(t)\prod_{q\neq j} v_q(t)\,. \tag{8}$$

Similarly, inner product with $w_i$ to both hand sides of (6) and replacing $X$ with $\sum_{i=1}^{r}\sigma_i(e_i w_i^\top)$ gives

$$\dot{o}_i(t) = w_i^\top \dot{u}(t) = -\frac{1}{N}\sigma_i\left(e_i^\top z(t)\right)\prod_{j=1}^{D-1} v_j(t)\,, \quad \forall i \in [r]. \tag{9}$$

Therefore, (8) and (9) together give the reparameterization of the GF dyanmics.

**Gradient Descent.** Similar to the GF dynamics, the GD dynamics with step size $\eta$ can be reparameterized as

$$v_j^{(t+1)} = v_j^{(t)} - \frac{\eta}{N}\sum_{i=1}^{r}\sigma_i\left(e_i^\top z^{(t)}\right)o_i^{(t)}\prod_{q\neq j} v_q^{(t)}\,, \quad \forall j \in [D-1]. \tag{10}$$

and

$$o_i^{(t+1)} = o_i^{(t)} - \frac{\eta}{N}\sigma_i\left(e_i^\top z^{(t)}\right)\prod_{j=1}^{D-1} v_j^{(t)}, \quad \forall i \in [r]. \tag{11}$$

**Mini-batch Stochastic Gradient Descent.** Recall that the mini-batch loss $\tilde{L}(\theta; P^{(t)})$ at step $t$ is given by

$$\tilde{L}(\theta; P^{(t)}) = \frac{1}{2B}\left\|P^{(t)}(f(X;\theta)-y)\right\|^2 = \frac{1}{2B}\left\|P^{(t)}z(\theta)\right\|^2 = \frac{1}{2B}z(\theta)^\top P^{(t)}z(\theta),$$

where $P^{(t)} \in \mathbb{R}^{N\times N}$ is an independently sampled random diagonal matrix with exactly $B$ diagonal entries chosen uniformly at random and set to 1 and the rest set to 0.

The update rule of SGD is given by

$$u^{(t+1)} = u^{(t)} - \frac{\eta}{B}X^\top P^{(t)}z^{(t)}\prod_{j=1}^{D-1} v_j^{(t)}, \tag{12}$$

$$v_j^{(t+1)} = v_j^{(t)} - \frac{\eta}{B}(z^{(t)})^\top P^{(t)}Xu^{(t)}\prod_{q\neq j} v_q^{(t)}, \quad \forall j \in [D-1]. \tag{13}$$

We can replace $X$ with $\sum_{i=1}^r \sigma_i(e_i w_i^\top)$ and rewrite (13) as

$$v_j^{(t+1)} = v_j^{(t)} - \frac{\eta}{B}\sum_{i=1}^r \sigma_i\left(e_i^\top P^{(t)}z^{(t)}\right)o_i^{(t)}\prod_{q\neq j} v_q^{(t)} \quad \forall j \in [D-1], \tag{14}$$

and similarly rewrite (12) as

$$o_i^{(t+1)} = o_i^{(t)} - \frac{\eta}{B}\sigma_i\left(e_i^\top P^{(t)}z^{(t)}\right)\prod_{j=1}^{D-1} v_j^{(t)} \quad \forall i \in [r]. \tag{15}$$

Note that GD is a special case of SGD when $B = N$ and $P^{(t)} = I$, where $I$ is an $N$-by-$N$ identity matrix.

### B.2. Proof of Theorem 4.3

Let $\theta^\star = (u^\star, v_1^\star)$ be a global minimizer of $L(\theta)$ for a two-layer minimalist model (1) trained on a dataset $(X, y)$ with difficulty $Q$. We denote $o_i^\star = w_i^\top u^\star$ for each $i \in [r]$. Since $L(\theta^\star) = \frac{1}{2N}\|z(\theta^\star)\|^2$, the residual $z(\theta^\star)$ is a zero vector. Combining with (5), we have

$$e_i^\top z(\theta^\star) = \sigma_i o_i^\star v_1^\star - d_i = 0, \quad \forall i \in [r].$$

Moreover, we have

$$C(\theta^\star) = \left(\sum_{i=1}^r (o_i^\star)^2\right) - (v_1^\star)^2,$$

by the definition of the layer imbalance. Substituting $o_i^\star = \frac{d_i}{\sigma_i v_1^\star}$ gives

$$C(\theta^\star) = \left(\sum_{i=1}^r \frac{d_i^2}{\sigma_i^2(v_1^\star)^2}\right) - (v_1^\star)^2 = \frac{Q}{(v_1^\star)^2} - (v_1^\star)^2,$$

which can be rewritten as a quadratic equation in $(v_1^\star)^2$:

$$((v_1^\star)^2)^2 + C(\theta^\star)(v_1^\star)^2 - Q = 0.$$

This quadratic equation has a unique positive solution, given by

$$(v_1^\star)^2 = \frac{-C(\theta^\star) + \sqrt{C(\theta^\star)^2 + 4Q}}{2} \,. \tag{16}$$

It is worth noting that, for a given dataset *difficulty* and *layer imbalance* at a global minimum $\theta^\star = (u^\star, v_1^\star)$, the second-layer weight $v_1^\star$ and $o_i^\star$ for each $i \in [r]$ are *uniquely* determined up to sign, as given by (16) and $o_i^\star = \frac{d_i}{\sigma_i v_1^\star}$.

Based on these facts, we will bound the sharpness $S(\theta^\star)$. First, note that the loss Hessian matrix exactly matches with the (normalized) Gauss-Newton (GN) matrix at a global minimum:

$$\nabla^2 L(\theta^\star) = \frac{1}{N} J(\theta^\star)^\top J(\theta^\star) \,,$$

where $J(\theta^\star) = \frac{\partial f}{\partial \theta}(\theta^\star) = \begin{bmatrix} Xv_1^\star & Xu^\star \end{bmatrix} \in \mathbb{R}^{N \times p}$ is a Jacobian matrix of the minimalist model. Hence, the sharpness at $\theta^\star$ matches (up to scaling) the spectral norm of the NTK matrix $J(\theta^\star)J(\theta^\star)^\top$ (Jacot et al., 2018):

$$S(\theta^\star) = \lambda_{\max}(\nabla^2 L(\theta^\star)) = \frac{1}{N}\|J(\theta^\star)^\top J(\theta^\star)\|_2 = \frac{1}{N}\|J(\theta^\star)J(\theta^\star)^\top\|_2$$

The NTK matrix can be written as

$$J(\theta^\star)J(\theta^\star)^\top = XX^\top(v_1^\star)^2 + Xu^\star(u^\star)^\top X^\top = (v_1^\star)^2 \sum_{i=1}^r \sigma_i^2 e_i e_i^\top + \sum_{i_1,i_2=1}^r \sigma_{i_1}\sigma_{i_2}o_{i_1}^\star o_{i_2}^\star e_{i_1}e_{i_2}^\top \,.$$

Hence, the NTK matrix at $\theta^\star$ is *uniquely* determined by the dataset and $C(\theta^\star)$ as

$$J(\theta^\star)J(\theta^\star)^\top = \left(\frac{-C(\theta^\star) + \sqrt{C(\theta^\star)^2 + 4Q}}{2}\right) \sum_{i=1}^r \sigma_i^2 e_i e_i^\top + \left(\frac{2}{-C(\theta^\star) + \sqrt{C(\theta^\star)^2 + 4Q}}\right) \sum_{i_1,i_2=1}^r d_{i_1}d_{i_2}e_{i_1}e_{i_2}^\top \,. \tag{17}$$

Based on (17), we can lower bound the spectral norm of the NTK matrix by

$$\|J(\theta^\star)J(\theta^\star)^\top\|_2 \geq e_1^\top J(\theta^\star)J(\theta^\star)^\top e_1 = \sigma_1^2(v_1^\star)^2 + \frac{d_1^2}{(v_1^\star)^2} \,,$$

and upper bound by

$$\|J(\theta^\star)J(\theta^\star)^\top\|_2 \leq \|XX^\top(v_1^\star)^2\|_2 + \|Xu^\star(u^\star)^\top X^\top\|_2 \leq \sigma_1^2(v_1^\star)^2 + \|Xu^\star\|_2^2 = \sigma_1^2(v_1^\star)^2 + \frac{\sum_{i=1}^r d_i^2}{(v^\star)^2} \,.$$

Therefore, we can obtain the desired bound on the sharpness:

$$\frac{1}{N}\left[\sigma_1^2(v_1^\star)^2 + \frac{d_1^2}{(v_1^\star)^2}\right] \leq S(\theta^\star) \leq \frac{1}{N}\left[\sigma_1^2(v_1^\star)^2 + \frac{\sum_{i=1}^r d_i^2}{(v_1^\star)^2}\right] \,.$$

### B.3. Proof of Theorem 4.6

The proof of Theorem 4.6 is analogous to the proof of Theorem 4.3 presented in Appendix B.2. Let $\theta^\star = (u^\star, v_1^\star, \ldots, v_{D-1}^\star)$ be a global minimizer of $L(\theta)$ for a minimalist model (1) of depth $D$ trained on a dataset $(X, y)$ with difficulty $Q$. We assume that $\theta^\star$ is balanced, i.e., $\|\Pi_W u\| = |v_1| = \cdots = |v_{D-1}|$. We denote $o_i^\star = w_i^\top u^\star$ for each $i \in [r]$. Since $L(\theta^\star) = \frac{1}{2N}\|z(\theta^\star)\|^2$, the residual $z(\theta^\star)$ is a zero vector. Combining with (5), we have

$$e_i^\top z(\theta^\star) = \sigma_i o_i^\star \prod_{j=1}^{D-1} v_j^\star - d_i = 0, \quad \forall i \in [r] \,.$$

Moreover, since $\theta^\star$ is balanced,

$$\sum_{i=1}^r o_i^2 = \|\Pi_W u\|^2 = (v_1^\star)^2 = \cdots = (v_{D-1}^\star)^2 \,.$$

Substituting $o_i^\star = d_i/(\sigma_i \prod_{j=1}^{D-1} v_j^\star)$ gives

$$\sum_{i=1}^r \frac{d_i^2}{\sigma_i^2 \prod_{j=1}^{D-1}(v_j^\star)^2} = (v_1^\star)^2 = \cdots = (v_{D-1}^\star)^2 \,.$$

Multiplying both sides by $(v_1^\star)^{2D-2}$ gives

$$(v_1^\star)^{2D} = \cdots = (v_{D-1}^\star)^{2D} = \sum_{i=1}^r \frac{d_i^2}{\sigma_i^2} = Q \,.$$

Hence, we can observe that

$$(v_1^\star)^2 = \cdots = (v_{D-1}^\star)^2 = Q^{\frac{1}{D}} \,,$$

and

$$(o_i^\star)^2 = \frac{d_i^2}{\sigma_i^2 \prod_{j=1}^{D-1}(v_j^\star)^2} = \frac{d_i^2}{\sigma_i^2} Q^{\frac{1-D}{D}} \quad \forall i \in [r] \,.$$

It is worth noting that $v_1^\star, \ldots, v_{D-1}^\star$ and $o_1^\star, \ldots, o_r^\star$ are *uniquely* determined up to sign.

Based on these facts, we will bound the sharpness $S(\theta^\star)$. First, note that the loss Hessian matrix exactly matches with the (normalized) Gauss-Newton (GN) matrix at a global minimum:

$$\nabla^2 L(\theta^\star) = \frac{1}{N} J(\theta^\star)^\top J(\theta^\star) \,,$$

where $J(\theta^\star) = \frac{\partial f}{\partial \theta}(\theta^\star) = (\prod_{j=1}^{D-1} v_j^\star)\left[ X \frac{1}{v_1^\star} X u^\star \cdots \frac{1}{v_{D-1}^\star} X u^\star \right] \in \mathbb{R}^{N \times p}$ is a Jacobian matrix of the minimalist model. Hence, the sharpness at $\theta^\star$ matches (up to scaling) the spectral norm of the NTK matrix $J(\theta^\star)J(\theta^\star)^\top$ (Jacot et al., 2018):

$$S(\theta^\star) = \lambda_{\max}(\nabla^2 L(\theta^\star)) = \frac{1}{N}\|J(\theta^\star)^\top J(\theta^\star)\|_2 = \frac{1}{N}\|J(\theta^\star)J(\theta^\star)^\top\|_2$$

The NTK matrix can be written as

$$J(\theta^\star)J(\theta^\star)^\top = XX^\top \prod_{j=1}^{D-1}(v_1^\star)^2 + \left( \sum_{j=1}^{D-1} Xu^\star(u^\star)^\top X^\top (v_j^\star)^{-2} \right) \prod_{j=1}^{D-1}(v_1^\star)^2 \tag{18}$$

$$= Q^{\frac{D-1}{D}} \sum_{i=1}^r \sigma_i^2 e_i e_i^\top + (D-1)Q^{\frac{D-2}{D}} \sum_{i_1,i_2}^r \sigma_{i_1}\sigma_{i_2}o_{i_1}o_{i_2}e_{i_1}e_{i_2}^\top \tag{19}$$

$$= Q^{\frac{D-1}{D}} \sum_{i=1}^r \sigma_i^2 e_i e_i^\top + (D-1)Q^{\frac{D-2}{D}} \sum_{i_1,i_2}^r d_{i_1}d_{i_2}Q^{\frac{1-D}{D}} e_{i_1}e_{i_2}^\top \tag{20}$$

$$= Q^{\frac{D-1}{D}} \sum_{i=1}^r \sigma_i^2 e_i e_i^\top + (D-1)Q^{-\frac{1}{D}} \sum_{i_1,i_2}^r d_{i_1}d_{i_2} e_{i_1}e_{i_2}^\top \,. \tag{21}$$

Note that the NTK matrix is also *uniquely* determined. Based on (18)-(21), we can lower bound the spectral norm of the NTK matrix by

$$\|J(\theta^\star)J(\theta^\star)^\top\|_2 \geq e_1^\top J(\theta^\star)J(\theta^\star)^\top e_1 = \sigma_1^2 Q^{\frac{D-1}{D}} + (D-1)d_1^2 Q^{-\frac{1}{D}}$$

and upper bound by

$$\|J(\theta^\star)J(\theta^\star)^\top\|_2 \leq \|XX^\top\|_2 \prod_{j=1}^{D-1}(v_1^\star)^2 + \left(\sum_{j=1}^{D-1}\|Xu^\star(u^\star)^\top X^\top\|_2 (v_j^\star)^{-2}\right)\prod_{j=1}^{D-1}(v_1^\star)^2$$

$$= \sigma_1^2 Q^{\frac{D-1}{D}} + (D-1)Q^{\frac{D-2}{D}}\|Xu^\star(u^\star)^\top X^\top\|_2$$

$$= \sigma_1^2 Q^{\frac{D-1}{D}} + (D-1)Q^{\frac{D-2}{D}}\|Xu^\star\|_2^2$$

$$= \sigma_1^2 Q^{\frac{D-1}{D}} + (D-1)Q^{\frac{D-2}{D}}\sum_{i=1}^{r}\sigma_1^2(o_i^\star)^2$$

$$= \sigma_1^2 Q^{\frac{D-1}{D}} + (D-1)Q^{\frac{D-2}{D}}\sum_{i=1}^{r}d_i^2 Q^{\frac{1-D}{D}}$$

$$= \sigma_1^2 Q^{\frac{D-1}{D}} + (D-1)Q^{-\frac{1}{D}}\sum_{i=1}^{r}d_i^2 .$$

Therefore, we can obtain the desired bound on the sharpness:

$$\frac{1}{N}\left[\sigma_1^2 Q^{\frac{D-1}{D}} + (D-1)d_1^2 Q^{-\frac{1}{D}}\right] \leq S(\theta^\star) \leq \frac{1}{N}\left[\sigma_1^2 Q^{\frac{D-1}{D}} + (D-1)\left(\sum_{i=1}^{r}d_i^2\right)Q^{-\frac{1}{D}}\right] .$$

### B.4. Proof of Lemma 5.2

We provide a proof for $D = 2$ and $D > 2$ separately.

**(1) $D = 2$.** It suffices to prove that $\dot{C}(\theta(t)) = 0$ for all $t \geq 0$. Recall that the layer imbalance is defined as

$$C(\theta(t)) = \left(\sum_{i=1}^{r}o_i(t)^2\right) - v_1(t)^2 .$$

Differentiating both sides with respect to $t$ gives

$$\dot{C}(\theta(t)) = \left(\sum_{i=1}^{r}2o_i(t)\dot{o}_i(t)\right) - 2v_1(t)\dot{v}_1(t)$$

$$= -\frac{2}{N}\sum_{i=1}^{r}\sigma_i\left(e_i^\top z(t)\right)o_i(t)v_1(t) + \frac{2}{N}\sum_{i=1}^{r}\sigma_i\left(e_i^\top z(t)\right)o_i(t)v_1(t)$$

$$= 0 ,$$

where we used (8) and (9) for the second equality.

**(2) $D > 2$.** It suffices to prove that

$$\frac{\partial}{\partial t}(\|\Pi_W u(t)\|^2) = \frac{\partial}{\partial t}(v_1(t)^2) = \cdots = \frac{\partial}{\partial t}(v_{D-1}(t)^2)$$

holds for any $t \geq 0$. Using (8) and (9), we have

$$\frac{\partial}{\partial t}(\|\Pi_W u(t)\|^2) = \sum_{i=1}^{r}\frac{\partial}{\partial t}(o_i(t)^2) = \sum_{i=1}^{r}2\dot{o}_i(t)o_i(t) = -\frac{2}{N}\sum_{i=1}^{r}\sigma_i(e_i^\top z(t))o_i(t)\prod_{j=1}^{D-1}v_j(t) ,$$

and

$$\frac{\partial}{\partial t}(v_j(t)^2) = 2\dot{v}_j(t)v_j(t) = -\frac{2}{N}\sum_{i=1}^{r}\sigma_i(e_i^\top z(t))o_i(t)\prod_{j'=1}^{D-1}v_{j'}(t) ,$$

for any $j \in [D-1]$. This completes the proof.

## B.5. Auxiliary Lemmas and Proofs

In this subsection, we provide auxiliary lemmas for the proof of Theorem 5.6 and 5.7.

**Lemma B.1.** *Given a matrix* $A = \begin{bmatrix} 0_{k \times k} & v \\ v^\top & 0 \end{bmatrix}$ *, and* $k \in \mathbb{N}$ *dimensional arbitrary vector* $v \in \mathbb{R}^k$,

$$\|A\|_2 = \|v\|.$$

*Proof.* Since $A$ is symmetric, its spectral norm is equal to the maximum absolute eigenvalue of $A$.

Let $(x, y)^\top \in \mathbb{R}^{k+1}$ be an eigenvector corresponding to an eigenvalue $\lambda$, where $x \in \mathbb{R}^k$ and $y \in \mathbb{R}$. Then the eigenvalue equation

$$A \begin{bmatrix} x \\ y \end{bmatrix} = \lambda \begin{bmatrix} x \\ y \end{bmatrix}$$

becomes

$$\begin{bmatrix} 0_{k \times k} & v \\ v^\top & 0 \end{bmatrix} \begin{bmatrix} x \\ y \end{bmatrix} = \begin{bmatrix} v\,y \\ v^\top x \end{bmatrix} = \lambda \begin{bmatrix} x \\ y \end{bmatrix}.$$

This yields the system:

$$\begin{cases} v\,y = \lambda x, \\ v^\top x = \lambda y. \end{cases}$$

**Case 1:** $y \neq 0$**.** In this case, from the first equation we obtain

$$x = \frac{y}{\lambda} v,$$

assuming $\lambda \neq 0$. Substituting this expression into the second equation gives:

$$v^\top \left( \frac{y}{\lambda} v \right) = \lambda y \quad \implies \quad \frac{y}{\lambda} \|v\|^2 = \lambda y.$$

Since $y \neq 0$, canceling $y$ we get:

$$\|v\|^2 = \lambda^2 \quad \implies \quad \lambda = \pm \|v\|.$$

**Case 2:** $y = 0$**.** If $y = 0$, the first equation becomes:

$$0 = \lambda x.$$

For a nontrivial eigenvector (i.e., $x \neq 0$), we must have $\lambda = 0$.

Thus, the eigenvalues of $A$ are:

$$\lambda = \|v\|, \quad \lambda = -\|v\|, \quad \text{and} \quad \lambda = 0 \quad \text{(with multiplicity at least } k - 1\text{)}.$$

Therefore, the spectral norm of $A$ is

$$\|A\|_2 = \max\{|\lambda|\} = \|v\|.$$

$\square$

**Lemma B.2.** *Define*

$$g(x) = \sqrt{x^2 + 4Q},$$

*where* $Q \geq 0$*. Then* $g$ *is convex on* $\mathbb{R}$*.*

*Proof.* Compute the first and second derivatives:

$$g'(x) = \frac{x}{\sqrt{x^2 + 4Q}}, \qquad g''(x) = \frac{\sqrt{x^2 + 4Q} - \frac{x^2}{\sqrt{x^2+4Q}}}{x^2 + 4Q} = \frac{4Q}{(x^2 + 4Q)^{3/2}} \geq 0.$$

Since $g''(x) \geq 0$ for all $x \in \mathbb{R}$, $g$ is convex.

$\square$

**Lemma B.3** (Sharpness Bounds for Two-Layer Minimalist Model with Arbitrary $\theta$). *For a two-layer minimalist model trained on a dataset $(X, y)$, the sharpness at $\theta = (u, v_1)$ is bounded by*

$$\frac{1}{N}\left[\sigma_1^2 v_1^2 + \sigma_1^2 o_1^2 + \frac{2\sigma_1^4 o_1^2 v_1^2}{v_1^2\sigma_1^2 + \sigma_1^2 o_1^2} - \frac{2\sigma_1^3 d_1\, o_1 v_1}{v_1^2\sigma_1^2 + \sigma_1^2 o_1^2}\right] \le S(\theta) \le \frac{1}{N}\left[\sigma_1^2 v_1^2 + \sum_{i=1}^{r}\sigma_i^2 o_i^2 + \sqrt{\sum_{i=1}^{r}\left(\sigma_i\left(\sigma_i o_i v_1 - d_i\right)\right)^2}\right].$$

*Proof.* Consider a two-layer minimalist model with parameters $\theta = (u, v_1)$ and Jacobian $J(\theta)$. By differentiating the loss twice,

$$\nabla^2 \frac{1}{2N}\left(z(\theta)^\top z(\theta)\right) = \nabla\frac{1}{N}\left(z(\theta)^\top J(\theta)\right) = \frac{1}{N}\left(J(\theta)^\top J(\theta) + \langle H(\theta), z(\theta)\rangle\right),$$

where we define $H(\theta) \in \mathbb{R}^{n \times p \times p}$ as $H_{i,j,k}(\theta) = \frac{\partial^2 z_i(\theta)}{\partial\theta_j\partial\theta_k}$ for $i \in [N]$, $j, k \in [p]$ and

$$[\langle H(\theta), z(\theta)\rangle]_{jk} = \sum_{i=1}^{N} z_i(\theta) H_{i,jk}(\theta), \quad j, k \in [p].$$

Therefore, sharpness at $\theta$ is given by

$$S(\theta) = \left\|\frac{1}{N}\left[J(\theta)^\top J(\theta) + \langle H(\theta), z(\theta)\rangle\right]\right\|_2.$$

We may write $J(\theta)J(\theta)^\top = XX^\top v_1^2 + Xuu^\top X^\top$. Using the singular value decomposition $XX^\top = \sum_{i=1}^{r}\sigma_i^2\, e_i e_i^\top$, and representing $Xu = \sum_{i=1}^{r}\sigma_i o_i e_i$, it follows that

$$J(\theta)J(\theta)^\top = v_1^2\sum_{i=1}^{r}\sigma_i^2\, e_i e_i^\top + \sum_{i,j=1}^{r}\sigma_i\sigma_j o_i o_j\, e_i e_j^\top.$$

**Lower bound**    For any unit vector $w \in \mathbb{R}^{d+1}$, we have

$$\|J(\theta)^\top J(\theta) + \langle H(\theta), z(\theta)\rangle\|_2 \ge w^\top (J(\theta)^\top J(\theta) + \langle H(\theta), z(\theta)\rangle)w.$$

Choosing $\frac{e_1^\top J(\theta)}{\|e_1^\top J(\theta)\|}$, a straightforward calculation on the term $J(\theta)^\top J(\theta)$ shows that

$$\frac{1}{\|e_1^\top J(\theta)\|^2}\, e_1^\top\left(J(\theta)J(\theta)^\top\right)^2 e_1 = \frac{(v_1)^4\sigma_1^4 + 2(v_1)^2\sigma_1^4(o_1)^2 + \sum_{i=1}^{r}\sigma_1^2\sigma_i^2(o_1)^2(o_i)^2}{(v_1)^2\sigma_1^2 + \sigma_1^2(o_1)^2}.$$

Since $\sum_{i=1}^{r}(o_i)^2 \ge (o_1)^2$, it follows that

$$\frac{1}{\|e_1^\top J(\theta)\|^2}\, e_1^\top\left(JJ^\top\right)^2 e_1 \ge (v_1)^2\sigma_1^2 + \sigma_1^2(o_1)^2.$$

Next, noting that

$$\langle H(\theta), z(\theta)\rangle = \begin{bmatrix} \mathbf{0}_{d\times d} & X^\top z(\theta) \\ z(\theta)^\top X & 0 \end{bmatrix},$$

a similar calculation shows that

$$e_1^\top J(\theta)\langle H(\theta), z(\theta)\rangle J(\theta)^\top e_1 = 2\sigma_1^3\, o_1\, v_1\left(\sigma_1\, o_1 v_1 - d_1\right),$$

where $e_1^\top J(\theta) = \sigma_1\begin{bmatrix} v_1 w_1 \\ o_1 \end{bmatrix}$. Dividing by $\|e_1^\top J(\theta)\|^2$ yields the term

$$\frac{2\sigma_1^4(o_1)^2(v_1)^2}{(v_1)^2\sigma_1^2 + \sigma_1^2(o_1)^2} - \frac{2\sigma_1^3 d_1\, o_1 v_1}{(v_1)^2\sigma_1^2 + \sigma_1^2(o_1)^2}.$$

Thus, combining these results, we obtain the following inequality:

$$\frac{1}{N}\left[\sigma_1^2(v_1)^2 + \sigma_1^2(o_1)^2 + \frac{2\sigma_1^4(o_1)^2(v_1)^2}{(v_1)^2\sigma_1^2 + \sigma_1^2(o_1)^2} - \frac{2\sigma_1^3 d_1\, o_1 v_1}{(v_1)^2\sigma_1^2 + \sigma_1^2(o_1)^2}\right] \le S(\theta).$$

**Upper bound**   To provide an upper bound, we use the triangular inequality as follows:

$$\frac{1}{N}\left\|J(\theta)^\top J(\theta) + \langle H(\theta), z(\theta)\rangle\right\|_2 \le \frac{1}{N}\left\|J(\theta)^\top J(\theta)\right\|_2 + \frac{1}{N}\|\langle H(\theta), z(\theta)\rangle\|_2$$

By standard norm inequalities,

$$\|J(\theta)^\top J(\theta)\|_2 = \|J(\theta)J(\theta)^\top\|_2 \le \|XX^\top\|_2 (v_1)^2 + \|Xu\|_2^2.$$

Since $\|XX^\top\|_2 = \sigma_1^2$ and $\|Xu\|_2^2 = \sum_{i=1}^r \sigma_i^2(o_i)^2$, we have

$$\|J(\theta)^\top J(\theta)\|_2 \le \sigma_1^2(v_1)^2 + \sum_{i=1}^r \sigma_i^2(o_i)^2.$$

Similarly,

$$\|\langle H(\theta), z(\theta)\rangle\|_2 = \|z(\theta)^\top X\| = \left\|\sum_{i=1}^r \sigma_i(e_i^\top z(\theta))w_i\right\|$$

$$= \left\|\sum_{i=1}^r \sigma_i(\sigma_i o_i v_1 - d_i)w_i\right\| = \sqrt{\sum_{i=1}^r \left(\sigma_i\left(\sigma_i o_i v_1 - d_i\right)\right)^2}.$$

The first equality is based on Lemma B.1.

Therefore, the upper bound for the sharpness is

$$S(\theta) \le \frac{1}{N}\left[\sigma_1^2(v_1)^2 + \sum_{i=1}^r \sigma_i^2(o_i)^2 + \sqrt{\sum_{i=1}^r \left(\sigma_i\left(\sigma_i o_i v_1 - d_i\right)\right)^2}\right].$$

$\square$

## B.6. Proof of Theorem 5.6

Note that the initialization follows
$$u^{(0)} \sim \mathcal{N}\left(0, \alpha^2 I_d\right), \quad v_1^{(0)} \sim \mathcal{N}\left(0, \beta^2\right).$$

Then,
$$o_i^{(0)} \sim \mathcal{N}\left(0, \alpha^2\right), \quad \mathbb{E}\left[(o_i^{(0)})^2\right] = \alpha^2, \quad \mathbb{E}\left[(v_1^{(0)})^2\right] = \beta^2.$$

From the lower bound of Lemma B.3,

$$\frac{1}{N}\left[\sigma_1^2(v_1^{(0)})^2 + \sigma_1^2(o_1^{(0)})^2 + \frac{2\sigma_1^4(o_1^{(0)})^2(v_1^{(0)})^2}{v_1^2\sigma_1^2 + \sigma_1^2 o_1^2} - \frac{2\sigma_1^3 d_1 o_1^{(0)} v_1^{(0)}}{(v_1^{(0)})^2\sigma_1^2 + \sigma_1^2(o_1^{(0)})^2}\right] \le S(\theta^{(0)}).$$

Hence, the expectation of the lower bound becomes

$$\mathbb{E}\left[S(\theta^{(0)})\right] \ge \frac{\sigma_1^2}{N}\left(\alpha^2 + \beta^2\right),$$

where the contribution of $-\frac{2\sigma_1^3 d_1 o_1^{(0)} v_1^{(0)}}{(v_1^{(0)})^2\sigma_1^2 + \sigma_1^2(o_1^{(0)})^2}$ term vanishes by symmetry, and we may drop $\frac{2\sigma_1^4(o_1^{(0)})^2(v_1^{(0)})^2}{(v_1^{(0)})^2\sigma_1^2 + \sigma_1^2(o_1^{(0)})^2} \ge 0$ for brevity.

From the upper bound of Lemma B.3,

$$S(\theta^{(0)}) \leq \frac{1}{N} \left[ \sigma_1^2 (v_1^{(0)})^2 + \sum_{i=1}^{r} \sigma_i^2 (o_i^{(0)})^2 + \sqrt{\sum_{i=1}^{r} \left( \sigma_i \left( \sigma_i \, o_i^{(0)} v_1^{(0)} - d_i \right) \right)^2} \right]. \tag{22}$$

Applying Jensen's inequality to the square-root term in Equation (22) and taking the expectation yields

$$\mathbb{E} \left[ \sqrt{\sum_{i=1}^{r} \left( \sigma_i \left( \sigma_i \, o_i^{(0)} v_1^{(0)} - d_i \right) \right)^2} \right] \leq \sqrt{\sum_{i=1}^{r} \mathbb{E} \left[ \sigma_i^2 \left( \sigma_i \, o_i^{(0)} v_1^{(0)} - d_i \right)^2 \right]}$$

$$= \sqrt{\sum_{i=1}^{r} \mathbb{E} \left[ \sigma_i^4 \left( o_i^{(0)} \right)^2 \left( v_1^{(0)} \right)^2 - 2 d_i \sigma_i^3 o_i^{(0)} v_1^{(0)} + \sigma_i^2 d_i^2 \right]}$$

$$= \sqrt{\sum_{i=1}^{r} \left( \alpha^2 \beta^2 \sigma_i^4 + \sigma_i^2 d_i^2 \right)}$$

Therefore, taking the expectation to Equation (22) results in the following inequality.

$$\mathbb{E} \left[ S(\theta^{(0)}) \right] \leq \frac{1}{N} \left[ \sum_{i=1}^{r} \alpha^2 \sigma_i^2 + \beta^2 \sigma_1^2 + \sqrt{\sum_{i=1}^{r} \sigma_i^2 \left( \alpha^2 \beta^2 \sigma_i^2 + d_i^2 \right)} \right].$$

Thus, we obtain the overall bounds

$$\frac{\sigma_1^2}{N} \left( \alpha^2 + \beta^2 \right) \leq \mathbb{E} \left[ S(\theta^{(0)}) \right] \leq \frac{1}{N} \left[ \sum_{i=1}^{r} \alpha^2 \sigma_i^2 + \beta^2 \sigma_1^2 + \sqrt{\sum_{i=1}^{r} \sigma_i^2 \left( \alpha^2 \beta^2 \sigma_i^2 + d_i^2 \right)} \right].$$

This finishes the proof.

### B.7. Proof of Theorem 5.7

Our analysis aims to derive the expectation of both the lower and upper bounds of Theorem 4.3. By substituting the expression for $(v_1^\star)^2$ and simplifying the resulting terms for the lower bound, we obtain:

$$\begin{aligned} S(\theta^\star) &\geq \frac{1}{N} \left[ \sigma_1^2 (v_1^\star)^2 + \frac{d_1^2}{(v_1^\star)^2} \right] = \frac{1}{N} \left[ \sigma_1^2 \frac{\sqrt{C(\theta^\star)^2 + 4Q} - C(\theta^\star)}{2} + \frac{2 d_1^2}{\sqrt{C(\theta^\star)^2 + 4Q} - C(\theta^\star)} \right] \\ &= \frac{1}{N} \left[ \sigma_1^2 \frac{\sqrt{C(\theta^\star)^2 + 4Q} - C(\theta^\star)}{2} + \frac{d_1^2 \left( \sqrt{C(\theta^\star)^2 + 4Q} + C(\theta^\star) \right)}{2Q} \right] \\ &= \frac{1}{2N} \left[ \left( \sigma_1^2 + \frac{d_1^2}{Q} \right) \sqrt{C(\theta^\star)^2 + 4Q} + \left( \frac{d_1^2}{Q} - \sigma_1^2 \right) C(\theta^\star) \right]. \end{aligned} \tag{23}$$

For the upper bound, we obtain the following in the same manner:

$$S(\theta^\star) \leq \frac{1}{N} \left[ \sigma_1^2 (v_1^\star)^2 + \frac{\sum_{i=1}^{r} d_i^2}{(v_1^\star)^2} \right] = \frac{1}{2N} \left[ \left( \sigma_1^2 + \frac{\sum_{i=1}^{r} d_i^2}{Q} \right) \sqrt{C(\theta^\star)^2 + 4Q} + \left( \frac{\sum_{i=1}^{r} d_i^2}{Q} - \sigma_1^2 \right) C(\theta^\star) \right]. \tag{24}$$

To derive expectations for both the lower bound and the upper bound, we first characterize $\mathbb{E}[C(\theta^\star)]$:

$$\mathbb{E}[C(\theta^\star)] = \mathbb{E}[C(\theta^{(0)})] = \mathbb{E}\left[\sum_{i=1}^{r}\left[(o_i^{(0)})^2\right] - (v_1^{(0)})^2\right] = r\alpha^2 - \beta^2\,, \tag{25}$$

where the first equality can be directly derived by Corollary 5.3

We characterize $\mathbb{E}[\sqrt{C(\theta^\star)^2 + 4Q}]$ by introducing two-sided bounds, using the following auxiliary lemma.

We now apply Jensen's inequality to bound $\mathbb{E}\left[\sqrt{C(\theta^\star)^2 + 4Q}\right]$ from below and above.

- **Lower bound via convexity.** By Lemma B.2, $g(x) = \sqrt{x^2 + 4Q}$ is convex. Hence

$$g\left(\mathbb{E}[C(\theta^\star)]\right) \leq \mathbb{E}\left[g(C(\theta^\star))\right] \implies \sqrt{\left(\mathbb{E}[C(\theta^\star)]\right)^2 + 4Q} \leq \mathbb{E}\left[\sqrt{C(\theta^\star)^2 + 4Q}\right].$$

- **Upper bound via concavity.** The function $h(y) = \sqrt{y}$ is concave on $[0, \infty)$. Let $Y = C(\theta^\star)^2 + 4Q \geq 0$. Jensen's inequality then gives

$$\mathbb{E}\left[h(Y)\right] \leq h\left(\mathbb{E}[Y]\right) \implies \mathbb{E}\left[\sqrt{C(\theta^\star)^2 + 4Q}\right] \leq \sqrt{\mathbb{E}\left[C(\theta^\star)^2\right] + 4Q}.$$

Combining the two yields the desired two-sided bound:

$$\sqrt{\left(\mathbb{E}[C(\theta^\star)]\right)^2 + 4Q} \leq \mathbb{E}\left[\sqrt{C(\theta^\star)^2 + 4Q}\right] \leq \sqrt{\mathbb{E}[C(\theta^\star)^2] + 4Q}\,. \tag{26}$$

For numerical bounds for both sides, we need to derive $\mathbb{E}[C(\theta^\star)^2]$ as follows:

$$\begin{aligned}
\mathbb{E}[C(\theta^\star)^2] &= \mathbb{E}[C(\theta^{(0)})^2] \\
&= \mathbb{E}\left[\left(\left(\sum_{i=1}^{r}\left(o_i^{(0)}\right)^2\right) - \left(v_1^{(0)}\right)^2\right)^2\right] \\
&= \mathbb{E}\left[\left(\sum_{i=1}^{r}\left(o_i^{(0)}\right)^2\right)^2 - 2\left(\sum_{i=1}^{r}\left(o_i^{(0)}\right)^2\right)\left(v_1^{(0)}\right)^2 + \left(v_1^{(0)}\right)^4\right] \\
&= \mathbb{E}\left[\left(\sum_{i=1}^{r}\left(o_i^{(0)}\right)^4\right) + \left(2\sum_{i}\sum_{j>i}\left(o_i^{(0)}\right)^2\left(o_j^{(0)}\right)^2\right) - 2\left(\sum_{i=1}^{r}\left(o_i^{(0)}\right)^2\right)\left(v_1^{(0)}\right)^2 + \left(v_1^{(0)}\right)^4\right] \\
&= r \cdot (3\alpha^4) + 2\binom{r}{2}\alpha^4 - 2r\alpha^2\beta^2 + 3\beta^4 \\
&= (r^2 + 2r)\alpha^4 - 2r\alpha^2\beta^2 + 3\beta^4\,. \tag{27}
\end{aligned}$$

Combining Equation (26) with Equation (25) and Equation (27), we can derive bounds as follows:

$$\sqrt{(r\alpha^2 - \beta^2)^2 + 4Q} \leq \mathbb{E}\left[\sqrt{C(\theta^\star)^2 + 4Q}\right] \leq \sqrt{(r^2 + 2r)\alpha^4 - 2r\alpha^2\beta^2 + 3\beta^4 + 4Q}\,. \tag{28}$$

Finally, to derive our desired lower bound for the expectation of the sharpness at convergence, we combine the expectation of Equation (23) with Equation (25) and the lower bound of Equation (28) as follows:

$$\mathbb{E}[S(\theta^\star)] \geq \frac{1}{2N}\mathbb{E}\left[\left(\sigma_1^2 + \frac{d_1^2}{Q}\right)\sqrt{C(\theta^\star)^2 + 4Q} + \left(\frac{d_1^2}{Q} - \sigma_1^2\right)C(\theta^\star)\right]$$

$$= \frac{1}{2N}\left[\left(\sigma_1^2 + \frac{d_1^2}{Q}\right)\mathbb{E}\left[\sqrt{C(\theta^\star)^2 + 4Q}\right] + \left(\frac{d_1^2}{Q} - \sigma_1^2\right)\mathbb{E}[C(\theta^\star)]\right]$$

$$\geq \frac{1}{2N}\left[\left(\sigma_1^2 + \frac{d_1^2}{Q}\right)\sqrt{(r\alpha^2 - \beta^2)^2 + 4Q} + \left(\frac{d_1^2}{Q} - \sigma_1^2\right)(r\alpha^2 - \beta^2)\right].$$

We derive the desired upper bound in the same manner:

$$\mathbb{E}[S(\theta^\star)] \leq \frac{1}{2N}\mathbb{E}\left[\left(\sigma_1^2 + \frac{\sum_{i=1}^r d_i^2}{Q}\right)\sqrt{C(\theta^\star)^2 + 4Q} + \left(\frac{\sum_{i=1}^r d_i^2}{Q} - \sigma_1^2\right)C(\theta^\star)\right]$$

$$= \frac{1}{2N}\left[\left(\sigma_1^2 + \frac{\sum_{i=1}^r d_i^2}{Q}\right)\mathbb{E}\left[\sqrt{C(\theta^\star)^2 + 4Q}\right] + \left(\frac{\sum_{i=1}^r d_i^2}{Q} - \sigma_1^2\right)\mathbb{E}[C(\theta^\star)]\right]$$

$$\leq \frac{1}{2N}\left[\left(\sigma_1^2 + \frac{\sum_{i=1}^r d_i^2}{Q}\right)\sqrt{(r^2 + 2r)\alpha^4 - 2r\alpha^2\beta^2 + 3\beta^4 + 4Q} + \left(\frac{\sum_{i=1}^r d_i^2}{Q} - \sigma_1^2\right)(r\alpha^2 - \beta^2)\right].$$

This finishes the proof.

### B.8. Proof of Theorem 5.9

For the proof, we consider the update equations of SGD, as it can unify the analysis for both GD and SGD. Starting from $\theta$ and its corresponding values of $o_1, \ldots, o_r$, the update equations under one SGD update can be written as

$$\Delta o_i = -\frac{\eta}{B}\sigma_i(z^\top Pe_i)v_1 \quad \text{for } i \in [r],$$

$$\Delta v_1 = -\frac{\eta}{B}\sum_i \sigma_i(z^\top Pe_i)o_i,$$

where we write $z := z(\theta)$ for simplicity. Note that plugging in $P = I$ and $B = N$ recovers the GD update. We now analyze change of $C$ after the update:

$$C(\theta_{\text{SGD}}^+) - C(\theta) = \sum_{i=1}^r (o_i + \Delta o_i)^2 - (v_1 + \Delta v_1)^2 - \left(\sum_{i=1}^r o_i^2 - v_1^2\right)$$

$$= \sum_{i=1}^r \left(2o_i\Delta o_i + (\Delta o_i)^2\right) - \left(2v_1\Delta v_1 + (\Delta v_1)^2\right)$$

$$= \sum_{i=1}^r (\Delta o_i)^2 - (\Delta v_1)^2. \tag{29}$$

Now, substituting the definitions of $\Delta o_i$ and $\Delta v_1$,

$$\sum_{i=1}^r (\Delta o_i)^2 - (\Delta v_1)^2 = \frac{\eta^2}{B^2}\left[\sum_{i=1}^r \sigma_i^2(z^\top Pe_i)^2 v_1^2 - \left(\sum_{i=1}^r \sigma_i(z^\top Pe_i)o_i\right)^2\right] \tag{30}$$

For simplification, we denote $\psi_i^\odot = \sigma_i(z(\theta) \odot e_i)$, and $\psi_i^P = \sigma_i(z(\theta)^\top Pe_i)$ for any arbitrary matrix $P \in \mathbb{R}^{N \times N}$. Then,

we can derive the following exact characterization:

$$
C(\theta_{\text{SGD}}^+) - C(\theta) = \sum_{i=1}^{r} (\Delta o_i)^2 - (\Delta v_1)^2
$$

$$
= \frac{\eta^2}{B^2} \left[ \sum_{i=1}^{r} \left( \psi_i^P \right)^2 v_1^2 - \left( \sum_{i=1}^{r} \psi_i^P o_i \right)^2 \right]
$$

$$
= \frac{\eta^2}{B^2} \left[ \sum_{i=1}^{r} \left[ \left( \psi_i^P \right)^2 \left( v_1^2 - o_i^2 \right) \right] - \sum_{i} \sum_{j>i} \left[ 2\psi_i^P \psi_j^P o_i o_j \right] \right]
$$

$$
= \frac{\eta^2}{B^2} \left[ \sum_{i=1}^{r} \left[ \left( \psi_i^P \right)^2 \left( v_1^2 - \sum_{j=1}^{r} o_j^2 \right) \right] + \sum_{i} \sum_{j \neq i} \left( \psi_i^P \right)^2 o_j^2 - \sum_{i} \sum_{j>i} \left[ 2\psi_i^P \psi_j^P o_i o_j \right] \right]
$$

$$
= \frac{\eta^2}{B^2} \left[ -\sum_{i=1}^{r} \left( \psi_i^P \right)^2 C(\theta) + \sum_{i} \sum_{j>i} \left[ \psi_i^P o_j - \psi_j^P o_i \right]^2 \right]. \tag{31}
$$

For GD, setting $P = I$ and $B = N$ finishes the proof. For SGD, we need to take the expectation over randomness of $P$. Recall that $P \in \mathbb{R}^{N \times N}$ a random diagonal matrix with exactly $B$ diagonal entries chosen uniformly at random and set to $1$ and the rest set to $0$.

To make the expectation easier to calculate, let us define $p \in \{0, 1\}^N$ satisfying $P = \text{diag}(p)$, where $\text{diag}(\cdot)$ denotes a diagonal matrix whose diagonal entries are equal to the components of the input vector. With this notation, we can write

$$
\mathbb{E}_P \left[ \psi_i^P \psi_j^P \right] = \mathbb{E}_p \left[ \sigma_i \sigma_j z^\top \text{diag}(e_i) p p^\top \text{diag}(e_j) z \right] = \sigma_i \sigma_j z^\top \text{diag}(e_i) \mathbb{E}_p \left[ p p^\top \right] \text{diag}(e_j) z. \tag{32}
$$

Now calculate the expectation of the random variable $pp^\top$. Recall that the $1$ entries of $p$ are sampled without replacement. For $i \in [N]$, let $p_i \in \{0, 1\}$ denote the $i$-th entry of $p$. Then,

$$
\mathbb{E}[p_i^2] = \mathbb{E}[p_i] = \frac{B}{N}, \quad \text{and} \quad \mathbb{E}[p_i p_j] = \frac{B(B-1)}{N(N-1)} \text{ for any } i \neq j.
$$

From this, $\mathbb{E}_p \left[ pp^\top \right]$ is a matrix whose diagonal entries are all $\frac{B}{N}$ and off-diagonals are $\frac{B(B-1)}{N(N-1)}$. We can write

$$
\mathbb{E}_p \left[ pp^\top \right] = \left( \frac{B}{N} - \frac{B(B-1)}{N(N-1)} \right) I + \frac{B(B-1)}{N(N-1)} \mathbf{1}\mathbf{1}^\top
$$

$$
= \frac{B(N-B)}{N(N-1)} I + \frac{B(B-1)}{N(N-1)} \mathbf{1}\mathbf{1}^\top, \tag{33}
$$

where $\mathbf{1} \in \mathbb{R}^N$ is a vector filled with $1$'s.

Substituting Equation (33) to Equation (32) yields

$$
\mathbb{E}_P \left[ \psi_i^P \psi_j^P \right] = \sigma_i \sigma_j \left[ \frac{B(N-B)}{N(N-1)} z^\top \text{diag}(e_i) \text{diag}(e_j) z + \frac{B(B-1)}{N(N-1)} z^\top \text{diag}(e_i) \mathbf{1}\mathbf{1}^\top \text{diag}(e_j) z \right]
$$

$$
= \sigma_i \sigma_j \left[ \frac{B(N-B)}{N(N-1)} (z \odot e_i)^\top (z \odot e_j) + \frac{B(B-1)}{N(N-1)} (z^\top e_i)(z^\top e_j) \right]
$$

$$
= \frac{B(N-B)}{N(N-1)} (\boldsymbol{\psi}_i^\odot)^\top \boldsymbol{\psi}_j^\odot + \frac{B(B-1)}{N(N-1)} \psi_i^I \psi_j^I. \tag{34}
$$

Combining Equation (31) and Equation (34) together, we get

$$\mathbb{E}_P\left[C(\theta_{\text{SGD}}^+)\right] - C(\theta) = \frac{\eta^2(N-B)}{BN(N-1)}\left[-\sum_{i=1}^r \|\boldsymbol{\psi}_i^{\odot}\|^2 C(\theta) + \sum_i \sum_{j>i} \|\boldsymbol{\psi}_i^{\odot}o_j - \boldsymbol{\psi}_j^{\odot}o_i\|^2\right]$$

$$+ \frac{\eta^2(B-1)}{BN(N-1)}\left[-\sum_{i=1}^r (\psi_i^I)^2 C(\theta) + \sum_i \sum_{j>i} (\psi_i^I o_j - \psi_j^I o_i)^2\right]$$

$$= \frac{\eta^2(N-B)}{BN^2(N-1)}\left[-N\underbrace{\sum_{i=1}^r \|\boldsymbol{\psi}_i^{\odot}\|^2}_{=:\Psi_2(\theta)} C(\theta) + N\underbrace{\sum_i \sum_{j>i} \|\boldsymbol{\psi}_i^{\odot}o_j - \boldsymbol{\psi}_j^{\odot}o_i\|^2}_{=:\Omega_2(\theta)}\right]$$

$$+ \frac{\eta^2(B-1)}{BN(N-1)}\left[-\underbrace{\sum_{i=1}^r (\psi_i^I)^2}_{=:\Psi_1(\theta)} C(\theta) + \underbrace{\sum_i \sum_{j>i} (\psi_i^I o_j - \psi_j^I o_i)^2}_{=:\Omega_1(\theta)}\right]$$

$$= \underbrace{\frac{\eta^2}{N^2}[-\Psi_1(\theta)C(\theta) + \Omega_1(\theta)]}_{=C(\theta_{\text{GD}}^+)-C(\theta)} + \frac{\eta^2(N-B)}{BN^2(N-1)}[-(\Psi_2(\theta) - \Psi_1(\theta))C(\theta) + (\Omega_2(\theta) - \Omega_1(\theta))],$$

where the last equality used

$$\frac{N-B}{BN^2(N-1)} + \frac{B-1}{BN(N-1)} = \frac{1}{N^2}.$$

The last thing to show is $\Psi_2(\theta) \geq \Psi_1(\theta)$ and $\Omega_2(\theta) \geq \Omega_1(\theta)$. For $\Psi_2(\theta) \geq \Psi_1(\theta)$, it suffices to show $N\|z \odot e_i\|^2 \geq (z^\top e_i)^2$. If we denote $a_k := [z \odot e_i]_k$, the $k$-th entry of $z \odot e_i$, then the inequality boils down to

$$N\sum_{i=1}^N a_i^2 \geq \left(\sum_{i=1}^N a_i\right)^2,$$

which is true by Jensen's inequality. The same holds for $\Omega_2(\theta) \geq \Omega_1(\theta)$, if we denote $a_k := [\psi_i^{\odot}o_j - \psi_j^{\odot}o_i]_k$. This finishes the proof.

## C. Experimental Details and More Results

We bootstrapped our own implementation from the work of Damian et al. (2023). We mainly used JAX for all the experiments except for Figure 28. To get the sharpness, we used LOBPCG (Knyazev, 2001) algorithm. If there's no specific mention in each experiment for initialization, we replicated pytorch default initialization, which has distribution of $\mathcal{U}\left(-\frac{1}{\sqrt{\text{fan\_in}}}, \frac{1}{\sqrt{\text{fan\_in}}}\right)$. Also, dataset difficulty $Q$ is calculated as follows to avoid the numerical instability: $Q \approx \sum_{i=1}^r \frac{d_i^2}{\max(\sigma_i^2, 10^{-9})}$.

Code is available at ⦿ Yoogeonhui/understand_progressive_sharpening.

### C.1. Experimental Details for Figure 1–3

In all cases, we trained tanh activation networks. More details are given as follows:

- Figure 1: We trained a width 512, 3-layer NN. For the dataset, we trained a $N = 2000$ subset of CIFAR10 ($d = 3072$) dataset. Except for the GF curve in the plot, we trained using GD with learning rates $2/500, 2/350, 2/200$.

- Figure 2a: We trained a width 512, 3-layer NN. For each case we trained $N = 10000$, $N = 5000$, $N = 2000$, $N = 1000$ subsets of CIFAR10 ($d = 3072$) dataset.

- Figure 2b: We trained width 512, with 3-, 4-, and 5-layer NN. For the dataset, we trained a $N = 10000$ subset of CIFAR10 ($d = 3072$) dataset.

- Figure 2c: We trained 3-layer NNs of widths 64, 128, 256, 512, 1024, and 2048. For the dataset, we trained a $N = 10000$ subset of CIFAR10 ($d = 3072$) dataset.

- Figure 3: We trained a width 512, 3-layer NN. For $B = N$ case we used GD, and in other cases, we used random reshuffling SGD (batch size 125, 250, and 500). Both are trained with learning rates 2/1000, 2/800, 2/600, 2/400, and 2/200. For the dataset, we trained on full CIFAR10 ($d = 3072$) dataset.

For Figure 2 and GF curve of Figure 1, we used Runge-Kutta 4 algorithm with the adaptive step size of $\frac{1.0}{S(\theta)}$, where $S(\theta)$ is measured once every 50 iterations. Note that our choice of experimental setting of GF followed Appendix I.5 of Cohen et al. (2021). All images have been normalized and standardized on a per-channel (RGB) basis.

## C.2. Data Processing Details on Google Speech Commands

We provide details on the preprocessing of the Google Speech Commands dataset (Warden, 2018) in Appendix C.4 and Appendix C.5.

We filtered the audio with equal or more than 1 second duration. These audio files are cropped to 1 second, and then converted to mel-spectrogram with the following code.

```
mel = librosa.feature.melspectrogram(y=audio_array, sr=16000, n_mels=64)
mel_db = librosa.power_to_db(mel, ref=np.max)
```

Then, mel-spectrograms were normalized and standardized globally.

## C.3. Experimental Details for Figure 6

For minimal model training, we sampled $X, y$ from pytorch normal random as follows:

$$X = \begin{bmatrix} 1.54099607 & -0.2934289 \\ -2.17878938 & 0.56843126 \end{bmatrix}, y = \begin{bmatrix} -1.08452237 \\ -1.39859545 \end{bmatrix}$$

for initialization, $u^{(0)} = \begin{bmatrix} 0.01 & 0.01 \end{bmatrix}, v_1^{(0)} = \begin{bmatrix} 0.01 \end{bmatrix}$. For GD learning rate we used $\eta = 2/50$.

For real-world experiments, we used the same transformer model architecture provided in Damian et al. (2023), which is a 2-layer Transformer with hidden dimension 64, and two attention heads. We used GELU activation, and $N = 100$ samples of SST2 with binary MSE loss, using GD learning rate $\eta = 2/400$.

## C.4. Experiments Figure 2–Figure 5 on SVHN and Google Speech Commands Dataset

We present experimental results on SVHN and Google Speech datasets that mirror Figure 2–Figure 5. In particular, Figure 10–Figure 13 reproduce our results in the main text, while Figure 14–Figure 17 do so for the speech data. All experimental settings are consistent across datasets, with only the dataset itself varying.

These confirm that the observed trends in Figure 2–Figure 5 hold across CIFAR-10, SVHN, and Google speech dataset.

### C.4.1. EXPERIMENTS ON SVHN

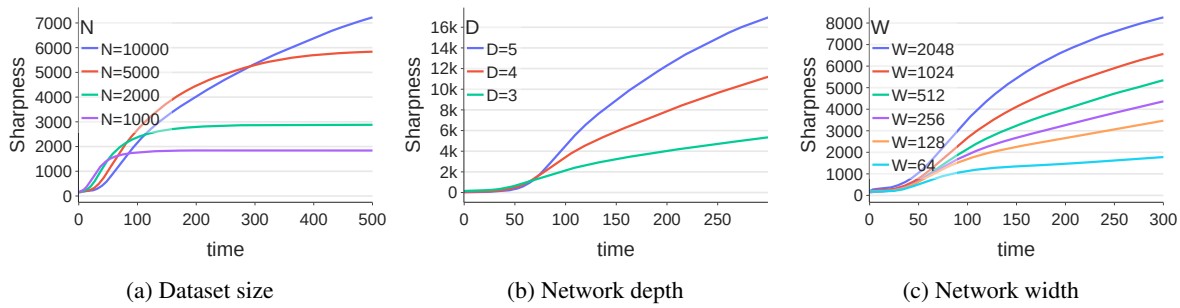

(a) Dataset size  (b) Network depth  (c) Network width

*Figure 10.* Effect of dataset size, network depth, and network width of tanh NN, SVHN dataset

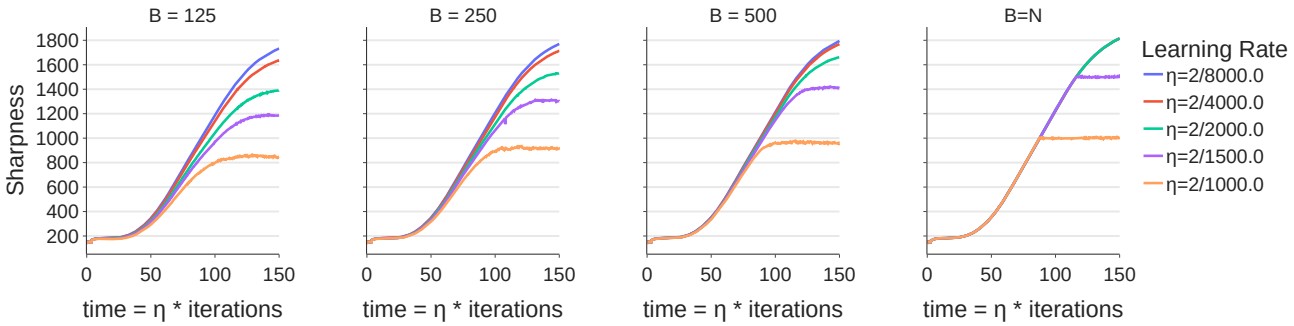

*Figure 11.* Effect of batch size, and learning rate in SGD and GD of tanh NN, SVHN dataset.

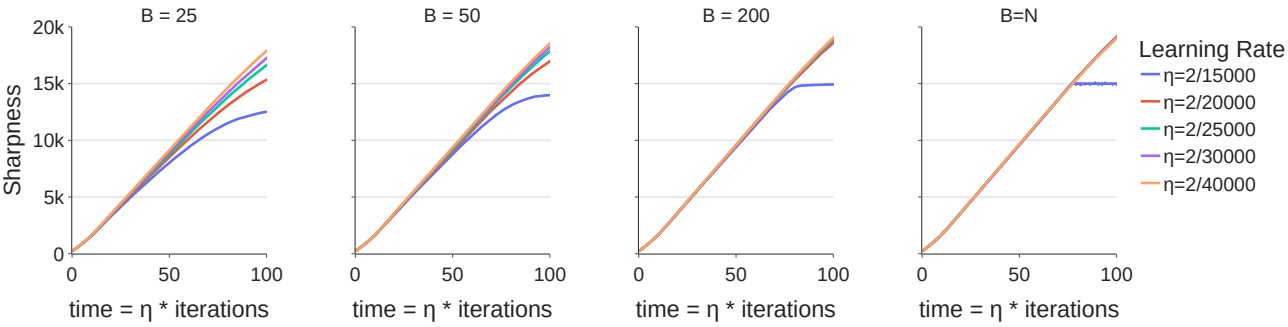

*Figure 12.* Effect of batch size, and learning rate in SGD and GD of minimal model, SVHN dataset.

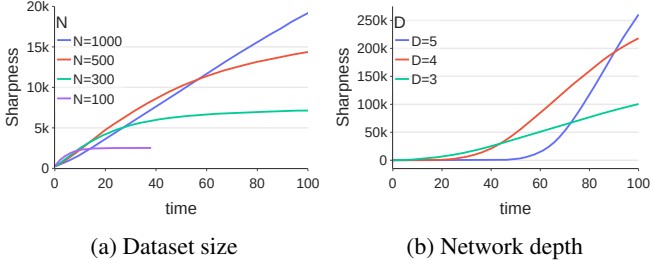

(a) Dataset size  (b) Network depth

*Figure 13.* Effect of dataset size, network depth of minimal model, SVHN dataset

## C.4.2. EXPERIMENTS ON GOOGLE SPEECH COMMANDS

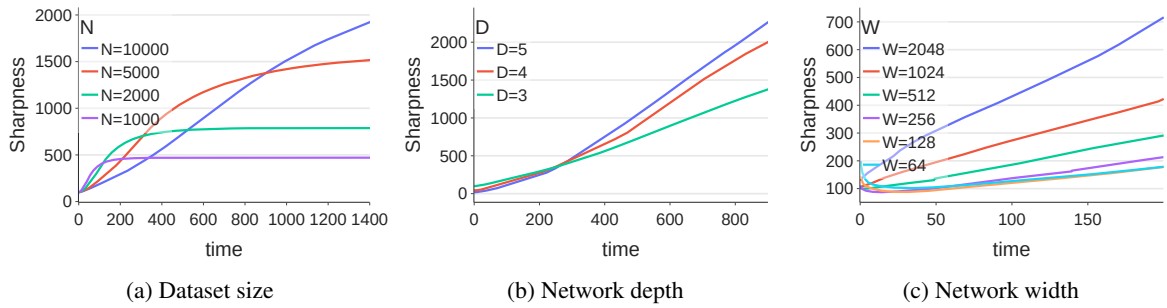

(a) Dataset size      (b) Network depth      (c) Network width

*Figure 14.* Effect of dataset size, network depth, and network width of tanh NN, Google speech commands dataset

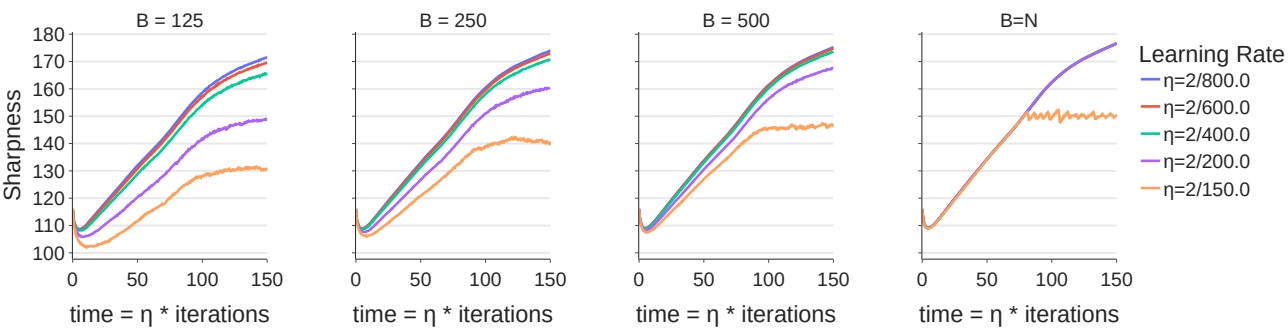

*Figure 15.* Effect of batch size, and learning rate in SGD and GD of tanh NN, Google speech commands dataset

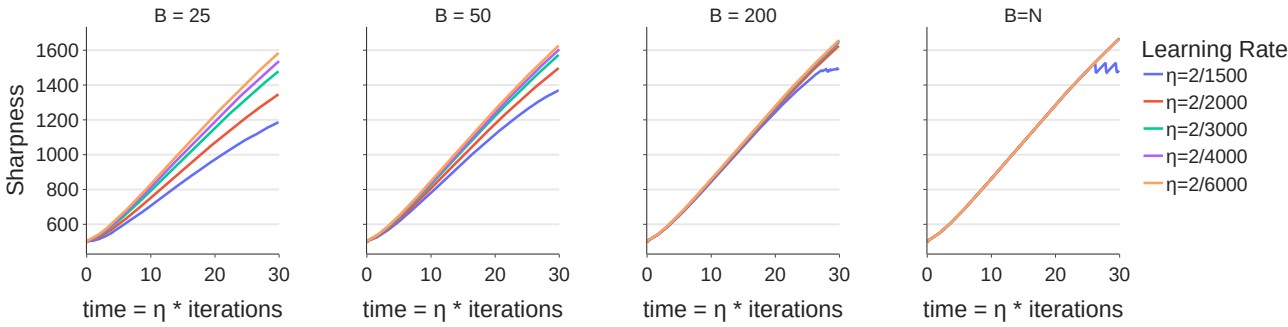

*Figure 16.* Effect of batch size, and learning rate in SGD and GD of minimal model, 2-label subsets of Google speech commands dataset (yes vs no)

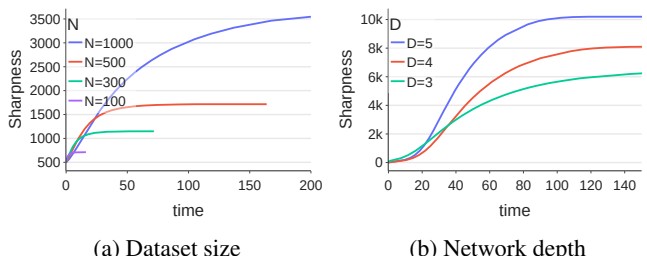

(a) Dataset size      (b) Network depth

*Figure 17.* Effect of dataset size, network depth of minimal model, 2-label subsets of google speech commands dataset (yes vs no)

## C.5. More Detailed Experimental Results for Figure 7 and Figure 8

We present more detailed experiments omitted from Figure 7 and Figure 8.

### C.5.1. EXPERIMENTS ON 2-LABEL (CAT VS DOG) SUBSETS OF CIFAR10

Table 5 shows values of correlation between empirical $S(\theta(\infty))$ vs predicted sharpness $\hat{S}_D$ and average empirical $S(\theta(\infty))$ for 2-label subsets of CIFAR10 with $N = 100$ and $N = 300$.

*Table 5.* Detailed results of GF experiments on 2-label subsets of CIFAR10.

(a) $N = 100$, correlation of empirical $S(\theta(\infty))$ vs predicted sharpness $\hat{S}_D$

| activation | identity | | | tanh | | | SiLU | | | ELU | | |
|---|---|---|---|---|---|---|---|---|---|---|---|---|
| width | 512 | 1024 | 2048 | 512 | 1024 | 2048 | 512 | 1024 | 2048 | 512 | 1024 | 2048 |
| depth=3 | 0.99 | 0.99 | 0.99 | 0.90 | 0.93 | 0.95 | 0.90 | 0.90 | 0.94 | 0.96 | 0.96 | 0.97 |
| depth=4 | 0.99 | 0.99 | 0.99 | 0.89 | 0.93 | 0.95 | 0.84 | 0.89 | 0.92 | 0.95 | 0.96 | 0.97 |
| depth=5 | 1.00 | 0.99 | 0.99 | 0.89 | 0.93 | 0.95 | 0.83 | 0.87 | 0.91 | 0.95 | 0.96 | 0.97 |

(b) $N = 300$, correlation of empirical $S(\theta(\infty))$ vs predicted sharpness $\hat{S}_D$

| activation | identity | | | tanh | | | SiLU | | | ELU | | |
|---|---|---|---|---|---|---|---|---|---|---|---|---|
| width | 512 | 1024 | 2048 | 512 | 1024 | 2048 | 512 | 1024 | 2048 | 512 | 1024 | 2048 |
| depth=3 | 0.99 | 0.99 | 0.99 | 0.74 | 0.76 | 0.83 | 0.86 | 0.86 | 0.87 | 0.85 | 0.89 | 0.91 |
| depth=4 | 0.99 | 0.99 | 0.99 | 0.73 | 0.75 | 0.85 | 0.85 | 0.84 | 0.86 | 0.85 | 0.89 | 0.90 |
| depth=5 | 0.99 | 0.99 | 0.99 | 0.80 | 0.78 | 0.84 | 0.80 | 0.82 | 0.85 | 0.87 | 0.89 | 0.90 |

(c) $N = 100$, average empirical $S(\theta(\infty))$

| activation | identity | | | tanh | | | SiLU | | | ELU | | |
|---|---|---|---|---|---|---|---|---|---|---|---|---|
| width | 512 | 1024 | 2048 | 512 | 1024 | 2048 | 512 | 1024 | 2048 | 512 | 1024 | 2048 |
| depth=3 | 495 | 518 | 566 | 248 | 266 | 293 | 97 | 97 | 101 | 266 | 283 | 313 |
| depth=4 | 478 | 493 | 527 | 236 | 250 | 270 | 73 | 71 | 72 | 240 | 254 | 273 |
| depth=5 | 468 | 479 | 505 | 231 | 243 | 259 | 62 | 59 | 58 | 228 | 239 | 255 |

(d) $N = 300$, average empirical $S(\theta(\infty))$

| activation | identity | | | tanh | | | SiLU | | | ELU | | |
|---|---|---|---|---|---|---|---|---|---|---|---|---|
| width | 512 | 1024 | 2048 | 512 | 1024 | 2048 | 512 | 1024 | 2048 | 512 | 1024 | 2048 |
| depth=3 | 1471 | 1502 | 1564 | 510 | 552 | 615 | 236 | 234 | 239 | 603 | 640 | 690 |
| depth=4 | 1701 | 1730 | 1791 | 573 | 620 | 680 | 215 | 204 | 204 | 647 | 683 | 724 |
| depth=5 | 1871 | 1897 | 1961 | 624 | 670 | 733 | 203 | 188 | 183 | 686 | 721 | 763 |

### C.5.2. EXPERIMENTS ON 2-LABEL (3 VS 5) SUBSETS OF SVHN

Table 6 shows values of correlation between empirical $S(\theta(\infty))$ vs predicted sharpness $\hat{S}_D$ and average empirical $S(\theta(\infty))$ for 2-label subsets of SVHN with $N = 100$.

*Table 6.* Detailed results of GF experiments on 2-label subsets of SVHN.

(a) N=100, correlation of empirical $S(\theta(\infty))$ vs $\hat{S}_D$

| activation | identity | | | tanh | | | SiLU | | | ELU | | |
|---|---|---|---|---|---|---|---|---|---|---|---|---|
| width | 512 | 1024 | 2048 | 512 | 1024 | 2048 | 512 | 1024 | 2048 | 512 | 1024 | 2048 |
| depth=3 | 1.00 | 1.00 | 1.00 | 0.74 | 0.82 | 0.86 | 0.78 | 0.79 | 0.83 | 0.85 | 0.89 | 0.92 |
| depth=4 | 1.00 | 0.99 | 1.00 | 0.78 | 0.81 | 0.86 | 0.72 | 0.76 | 0.78 | 0.86 | 0.87 | 0.91 |
| depth=5 | 0.99 | 0.99 | 1.00 | 0.80 | 0.84 | 0.87 | 0.66 | 0.73 | 0.77 | 0.86 | 0.88 | 0.91 |

(b) N=100, average empirical $S(\theta(\infty))$

| activation | identity | | | tanh | | | SiLU | | | ELU | | |
|---|---|---|---|---|---|---|---|---|---|---|---|---|
| width | 512 | 1024 | 2048 | 512 | 1024 | 2048 | 512 | 1024 | 2048 | 512 | 1024 | 2048 |
| depth=3 | 2759 | 2823 | 2946 | 1021 | 1086 | 1164 | 497 | 476 | 474 | 1029 | 1115 | 1219 |
| depth=4 | 3099 | 3153 | 3263 | 1113 | 1172 | 1241 | 478 | 432 | 408 | 1059 | 1134 | 1218 |
| depth=5 | 3352 | 3395 | 3493 | 1179 | 1246 | 1300 | 474 | 408 | 368 | 1089 | 1166 | 1240 |

### C.5.3. EXPERIMENTS ON 2-LABEL (YES VS NO) SUBSETS OF GOOGLE SPEECH COMMANDS

Table 7 shows values of correlation between empirical $S(\theta(\infty))$ vs predicted sharpness $\hat{S}_D$ and average empirical $S(\theta(\infty))$ for 2-label subsets of Google speech commands with $N = 100$.

*Table 7.* Detailed results of GF experiments on 2-label subsets of Google speech commands dataset(yes vs no).

(a) N=100, correlation of empirical $S(\theta(\infty))$ vs $\hat{S}_D$

| activation | identity | | | tanh | | | SiLU | | | ELU | | |
|---|---|---|---|---|---|---|---|---|---|---|---|---|
| width | 512 | 1024 | 2048 | 512 | 1024 | 2048 | 512 | 1024 | 2048 | 512 | 1024 | 2048 |
| depth=3 | 0.98 | 0.98 | 0.96 | 0.75 | 0.87 | 0.88 | 0.60 | 0.68 | 0.77 | 0.78 | 0.87 | 0.88 |
| depth=4 | 0.98 | 0.98 | 0.98 | 0.74 | 0.86 | 0.90 | 0.61 | 0.64 | 0.73 | 0.81 | 0.85 | 0.89 |
| depth=5 | 0.98 | 0.99 | 0.98 | 0.78 | 0.85 | 0.89 | 0.64 | 0.64 | 0.71 | 0.83 | 0.86 | 0.89 |

(b) N=100, average empirical $S(\theta(\infty))$

| activation | identity | | | tanh | | | SiLU | | | ELU | | |
|---|---|---|---|---|---|---|---|---|---|---|---|---|
| width | 512 | 1024 | 2048 | 512 | 1024 | 2048 | 512 | 1024 | 2048 | 512 | 1024 | 2048 |
| depth=3 | 500 | 532 | 598 | 252 | 282 | 328 | 178 | 162 | 154 | 303 | 323 | 366 |
| depth=4 | 491 | 514 | 561 | 238 | 265 | 301 | 151 | 133 | 119 | 278 | 295 | 325 |
| depth=5 | 487 | 504 | 541 | 230 | 256 | 287 | 136 | 116 | 102 | 261 | 280 | 304 |

**C.6. More Experiments for Table 2–4 on 2-label Subsets of SVHN (3 vs 5) and Google Speech Commands (yes vs no)**

We present more detailed experimental results omitted from Table 2 to Table 4. All experiments are carried out using 2-label (3 vs 5, yes vs no) subsets of SVHN and Google speech commands dataset.

Table 8 shows average values of $\tilde{C}$ computed for 2-label subsets of SVHN and Google speech commands of different $N$'s, omitted from Section 4.1. We observe the same drastic growth of $\tilde{C}$ as $N$ increases.

Table 8. Average $\tilde{C}$ that minimizes upper bound of Theorem 4.3, computed from 2-label subsets of SVHN and Google speech commands.

| Dataset | $N=100$ | $N=300$ | $N=500$ | $N=1000$ |
|---|---|---|---|---|
| SVHN | 49.29(11.76) | 897.24(127.75) | 4167.96(537.27) | 36556.80(3015.35) |
| Google speech commands | 7.10(1.12) | 43.96(4.86) | 119.38(10.01) | 688.18(60.07) |

Table 9 and Table 10 show values of the two terms that appear in Theorem 4.6, calculated using 2-label subsets of SVHN and Google speech commands. We observe the same trend as in Section 5.1: the predicted sharpness $\hat{S}_D$ dominates the bounds, making it a reliable proxy for the final sharpness.

Table 9. Average $\hat{S}_D = \frac{\sigma_1^2}{N} Q^{\frac{D-1}{D}}$ computed from 2-label subsets of the following datasets

(a) SVHN, with standard deviation in parenthesis

| $D$ | $N = 100$ | $N = 300$ | $N = 500$ | $N = 1000$ |
|---|---|---|---|---|
| 2 | 1936.80(302.71) | 8264.30(712.54) | 17851.97(1343.15) | 52935.36(2685.86) |
| 3 | 1988.28(379.48) | 13748.10(1470.64) | 38350.34(3640.47) | 163272.16(10276.97) |
| 4 | 2015.60(419.40) | 17735.64(2084.52) | 56218.58(5896.34) | 286762.97(19824.50) |
| 5 | 2032.51(443.82) | 20665.06(2560.53) | 70724.27(7841.86) | 402067.19(29295.27) |

(b) Google speech commands, with standard deviation in parenthesis

| $D$ | $N = 100$ | $N = 300$ | $N = 500$ | $N = 1000$ |
|---|---|---|---|---|
| 2 | 362.74 (34.85) | 891.27 (57.97) | 1468.31 (73.87) | 3500.62 (151.26) |
| 3 | 291.73 (34.35) | 971.45 (77.30) | 1890.28 (117.04) | 6038.67 (345.34) |
| 4 | 261.68 (33.80) | 1014.34 (88.64) | 2144.92 (145.91) | 7931.86 (510.36) |
| 5 | 245.18 (33.38) | 1041.02 (96.00) | 2313.93 (166.05) | 9342.15 (641.56) |

Table 10. Average $\frac{D-1}{N}\left(\sum_{i=1}^r d_i^2\right) Q^{-\frac{1}{D}}$ computed from 2-label subsets of the following datasets

(a) SVHN, with standard deviation in parenthesis

| $D$ | $N=100$ | $N=300$ | $N=500$ | $N=1000$ |
|---|---|---|---|---|
| 2 | 0.95(0.12) | 0.22(0.02) | 0.10(0.01) | 0.03(0.00) |
| 3 | 1.93(0.16) | 0.73(0.03) | 0.44(0.02) | 0.21(0.01) |
| 4 | 2.92(0.18) | 1.40(0.05) | 0.96(0.03) | 0.55(0.01) |
| 5 | 3.91(0.19) | 2.18(0.06) | 1.60(0.04) | 1.04(0.02) |

(b) Google speech commands, with standard deviation in parentheses

| $D$ | $N=100$ | $N=300$ | $N=500$ | $N=1000$ |
|---|---|---|---|---|
| 2 | 1.94(0.14) | 0.78(0.04) | 0.47(0.02) | 0.20(0.01) |
| 3 | 3.11(0.15) | 1.69(0.06) | 1.21(0.03) | 0.67(0.02) |
| 4 | 4.18(0.16) | 2.64(0.07) | 2.06(0.04) | 1.33(0.03) |
| 5 | 5.21(0.15) | 3.61(0.08) | 2.96(0.05) | 2.08(0.04) |

## C.7. Experiments for Theorem 5.6 and Theorem 5.7

To numerically validate the Theorems 5.6 and 5.7 and to assess the tightness of our bounds, we report Table 11–15, and Figure 18–20. We use $\alpha^2 = \frac{1}{3d}, \beta^2 = \frac{1}{3}$ so that the initialization matches the variance of PyTorch's default initialization.

Table 11 reports the lower bound, and Table 12 shows the corresponding gap from Theorem 5.6. Table 12 confirms that our theoretical bounds are tight. In Table 13, we empirically calculated the initial sharpness by sampling 400 random weight initializations for each 2-label subset of datasets, thereby calculating the average and standard deviation of 20000 different sharpness values. A comparison with the values in Table 11 reveals agreement between theory and experiment. We note that the standard deviation in Table 13 is relatively large, which is to be expected given the extremely small width of our minimalist model.

Table 14 reports the lower bound, and Table 15 shows the corresponding gap for Theorem 5.7. These numerical results demonstrate that the bound gap decreases, and thus the theoretical bounds become tighter as the dataset size $N$ grows.

To verify that the lower bound in Table 14 predicts the sharpness at convergence, we performed additional experiments following the setup of Figure 5b, but using the $\alpha\beta$ initialization with $\alpha^2 = \frac{1}{3d}, \beta^2 = \frac{1}{3}$ over three random weight seeds. Although individual runs exhibit variability in sharpness due to the model's extremely small width, Figures 18–20 show that, for small dataset sizes, the empirical sharpness saturates and its magnitude closely matches the prediction in Table 14. For larger $N$ (i.e., $N \geq 500$ for SVHN, and $N = 1000$ for CIFAR10 and Google speech commands), convergence to this saturation of sharpness would require more training iterations than are shown in our current plots.

By combining the results in Tables 11 and 14, we verify that the expected sharpness increases from initialization to convergence.

*Table 11.* Average lower bound of $\mathbb{E}[S(\theta^{(0)})]$ computed from 50 2-label subsets of datasets, with standard deviation in parenthesis, $\alpha^2 = \frac{1}{3d}, \beta^2 = \frac{1}{3}$.

| dataset | $N=100$ | $N=300$ | $N=500$ | $N=1000$ |
|---|---|---|---|---|
| CIFAR | 262.57(29.72) | 260.42(15.41) | 261.94(12.15) | 261.93(9.16) |
| SVHN | 602.25(39.35) | 600.50(23.20) | 601.97(15.63) | 602.17(12.55) |
| Google speech commands | 233.42(12.69) | 230.01(10.14) | 229.77(6.93) | 227.72(3.80) |

*Table 12.* Average gap between the upper bound and the lower bound of $\mathbb{E}[S(\theta^{(0)})]$ computed from 50 2-label subsets of datasets, with standard deviation in parenthesis, $\alpha^2 = \frac{1}{3d}, \beta^2 = \frac{1}{3}$.

| dataset | $N=100$ | $N=300$ | $N=500$ | $N=1000$ |
|---|---|---|---|---|
| CIFAR | 8.91(1.03) | 7.44(0.47) | 7.18(0.36) | 7.00(0.27) |
| SVHN | 12.54(1.43) | 11.66(0.71) | 11.49(0.42) | 11.33(0.25) |
| Google speech commands | 11.36(0.58) | 10.88(0.37) | 10.81(0.26) | 10.69(0.16) |

*Table 13.* Average of $S(\theta^{(0)})$ over 50 two-label subset of each dataset. For each subset, $S(\theta^{(0)})$ was computed across 400 random *weight* initializations (20,000 samples total); the standard deviation is shown in parentheses.

| dataset | $N=100$ | $N=300$ | $N=500$ | $N=1000$ |
|---|---|---|---|---|
| CIFAR | 266.56 (388.70) | 264.07 (383.16) | 265.37 (384.81) | 265.40 (384.68) |
| SVHN | 610.09 (886.94) | 608.07 (882.80) | 609.07 (883.88) | 609.38 (884.33) |
| Google speech commands | 237.50 (342.38) | 233.95 (337.19) | 233.68 (336.58) | 231.59 (333.44) |

*Table 14.* Average lower bound of $\mathbb{E}[S(\theta^{\star})]$ computed from 50 2-label subsets of datasets, with standard deviation in parenthesis, $\alpha^2 = \frac{1}{3d}, \beta^2 = \frac{1}{3}$.

| dataset | $N=100$ | $N=300$ | $N=500$ | $N=1000$ |
|---|---|---|---|---|
| CIFAR | 514.11(75.15) | 1141.41(97.49) | 1964.56(132.45) | 5325.42(298.65) |
| SVHN | 2250.11(316.08) | 8539.61(718.93) | 18105.67(1347.25) | 53138.75(2688.77) |
| Google speech commands | 490.39(38.74) | 994.79(60.39) | 1557.68(75.34) | 3559.35(151.39) |

*Table 15.* Average gap between the upper bound and the lower bound of $\mathbb{E}[S(\theta^{\star})]$ computed from 50 2-label subsets of datasets, with standard deviation in parenthesis, $\alpha^2 = \frac{1}{3d}, \beta^2 = \frac{1}{3}$.

| dataset | $N=100$ | $N=300$ | $N=500$ | $N=1000$ |
|---|---|---|---|---|
| CIFAR | 44.16(4.46) | 17.15(1.00) | 9.64(0.57) | 3.42(0.15) |
| SVHN | 47.14(5.06) | 11.15(0.81) | 5.19(0.30) | 1.75(0.07) |
| Google speech commands | 36.04(2.77) | 15.36(1.10) | 9.39(0.48) | 3.89(0.19) |

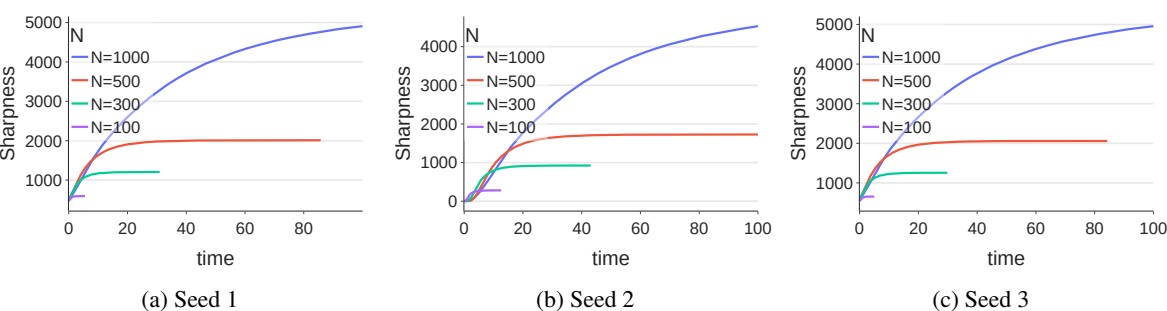

*Figure 18.* Minimalist model ($D = 2$) trained on different random weight seed, normal distribution with $\alpha^2 = \frac{1}{3d}, \beta^2 = \frac{1}{3}$ initialized, and the same 2-label subset of CIFAR10.

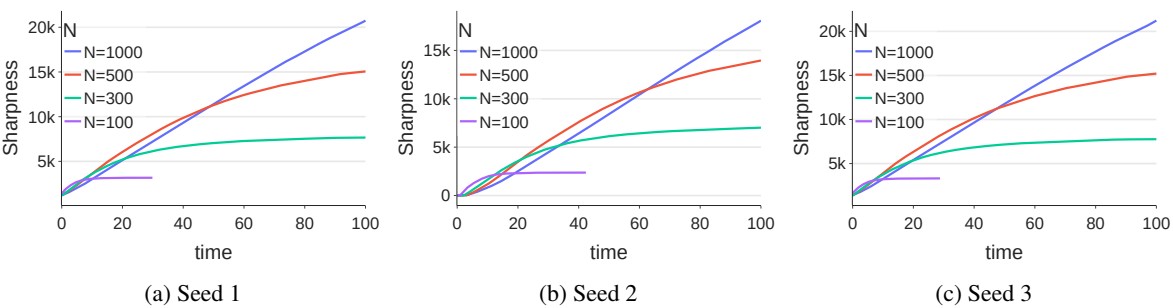

*Figure 19.* Minimalist model ($D = 2$) trained on different random weight seed, normal distribution with $\alpha^2 = \frac{1}{3d}, \beta^2 = \frac{1}{3}$ initialized, and the same 2-label subset of SVHN.

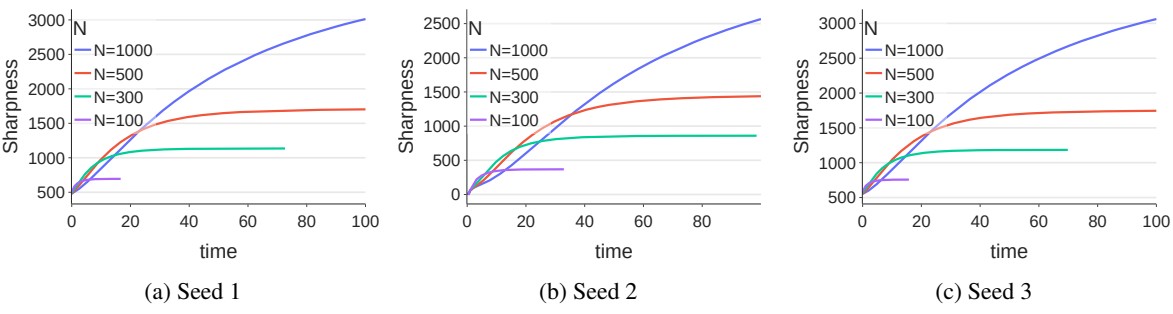

*Figure 20.* Minimalist model ($D = 2$) trained on different random weight seed, normal distribution with $\alpha^2 = \frac{1}{3d}, \beta^2 = \frac{1}{3}$ initialized, and the same 2-label subset of Google speech commands.

## C.8. Change of $T_1$, $T_2$, $\Psi_1$, $\Psi_2$, $\Omega_1$, and $\Omega_2$ for (S)GD

We present our experimental results for the terms we introduced in Theorem 5.9. Experimental settings are the same as Figure 4 and Figure 9.

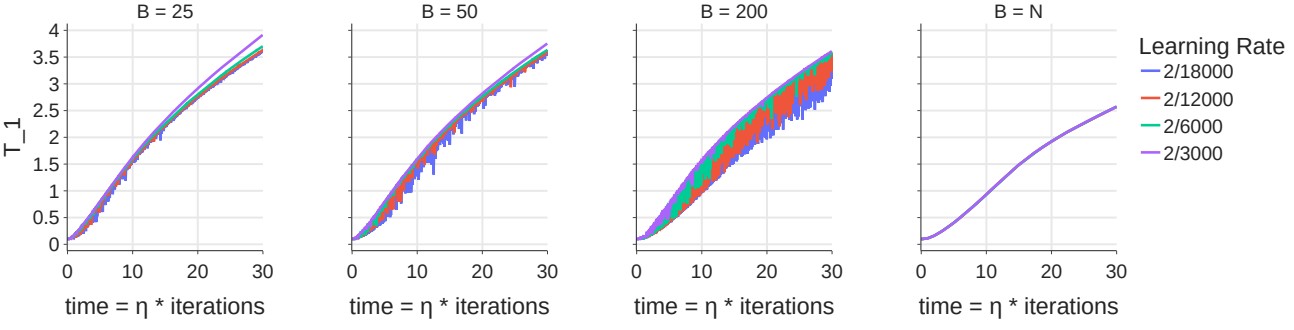

Figure 21. Change of $T_1$

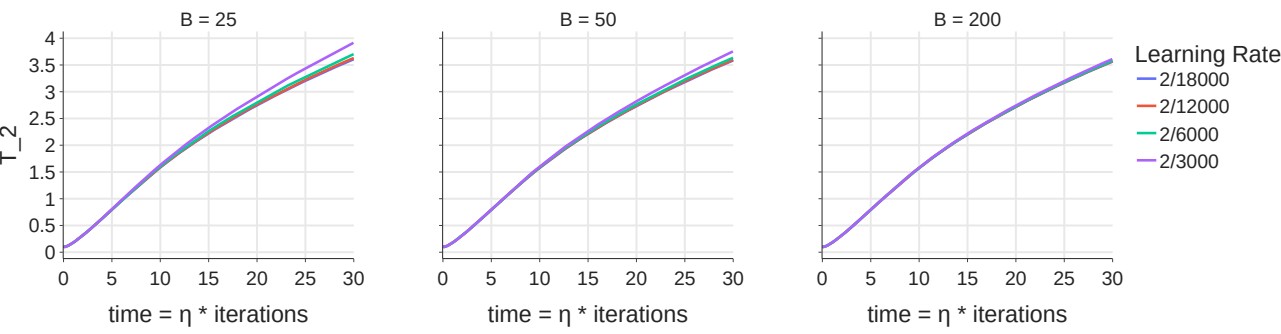

Figure 22. Change of $T_2$

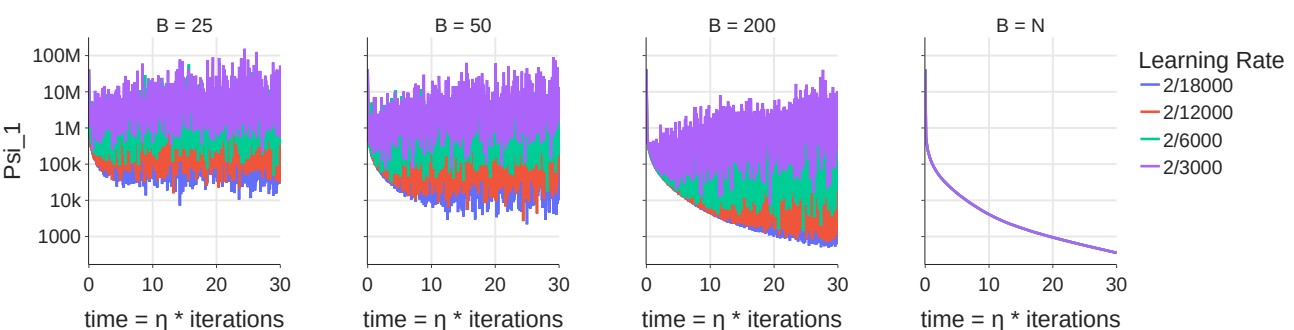

Figure 23. Change of $\Psi_1$

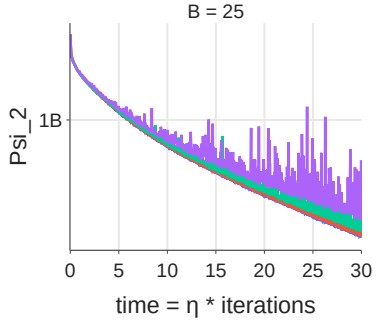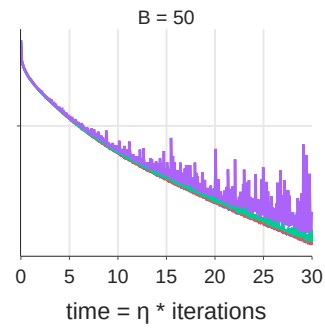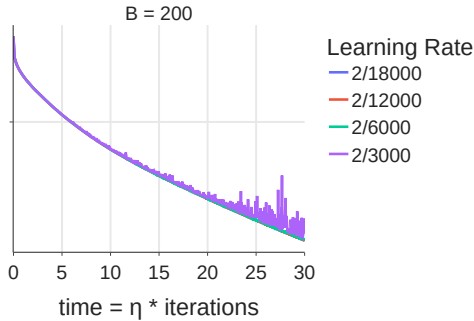

*Figure 24.* Change of $\Psi_2$

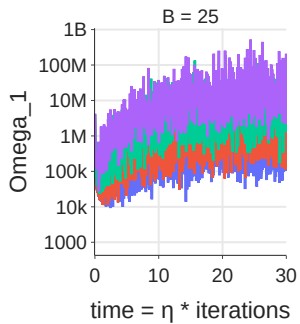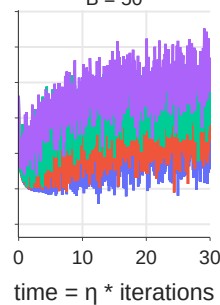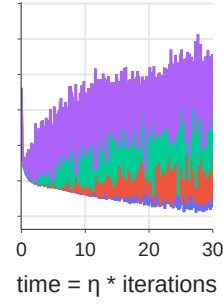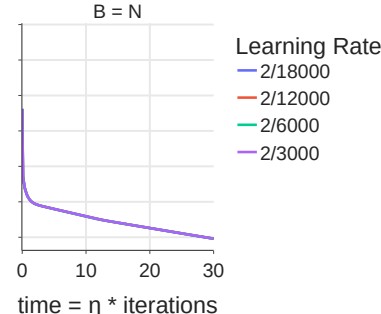

*Figure 25.* Change of $\Omega_1$

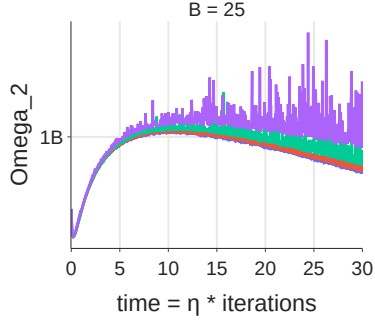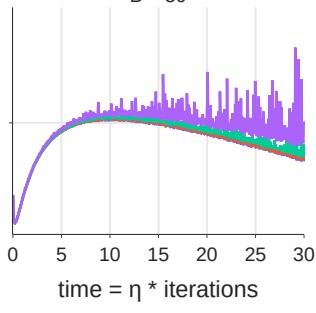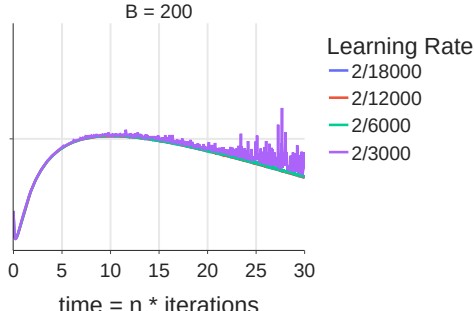

*Figure 26.* Change of $\Omega_2$

# D. Precision Dependence of EoS Behaviors

## D.1. Experiments in Synthetic Dataset

To check the precision dependence of EoS behaviors, we show our experimental results from JAX, and our own framework. For the training data, we generated our synthetic data with the code below.

```python
def minimal_data(d, common_size, signal_size, alpha, beta):

    np.random.seed(0)
    X = np.zeros((2, d))

    X_common = np.random.random((1, d))
    X_opposite = np.random.random((1, d))

    X_common /= np.linalg.norm(X_common)
    X_opposite -= X_common * (X_opposite.flatten() @ X_common.flatten())
    X_opposite /= np.linalg.norm(X_opposite) # Orthogonalizing X_common and X_opposite
    X[0] = common_size * X_common + signal_size * X_opposite
    X[1] = common_size * X_common - signal_size * X_opposite

    X_flat = X.reshape(2, -1)
    eigs, vecs = np.linalg.eigh(X_flat @ X_flat.T)

    y = vecs[:, 0] * alpha + vecs[:, 1] * beta
    return X, y
```

We used the result of minimal_data(100, 5.477, 0.233, 0.3, 1.414) for both cases. We trained our minimal model of $D = 2$ using $\eta = 0.02$ GD.

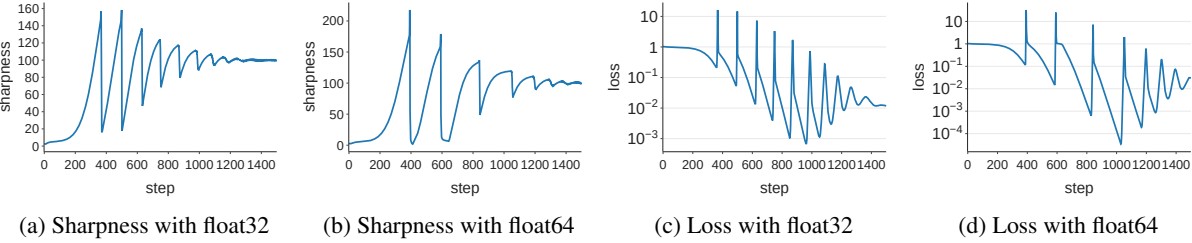

(a) Sharpness with float32     (b) Sharpness with float64     (c) Loss with float32     (d) Loss with float64

*Figure 27.* Effects of precision at the Edge of Stability, Experiment done in JAX

In Figure 27, we can see that our loss and sharpness plot of minimal model highly depends on the precision. We used the data type as jax.numpy.float32 for float32 case, and jax.numpy.float64 for float64 case.

To clearly see the impact of precision in training dynamics, for the same settings, we made a simple C++ program replicating the PyTorch training using MPFR C++ (Holoborodko, 2010). In Figure 28, we can clearly observe distinct trends across different precisions.

Notably, we find that high precision may lead to a blow up of the training loss instead of entering the edge of stability regime and converging non-monotonically.

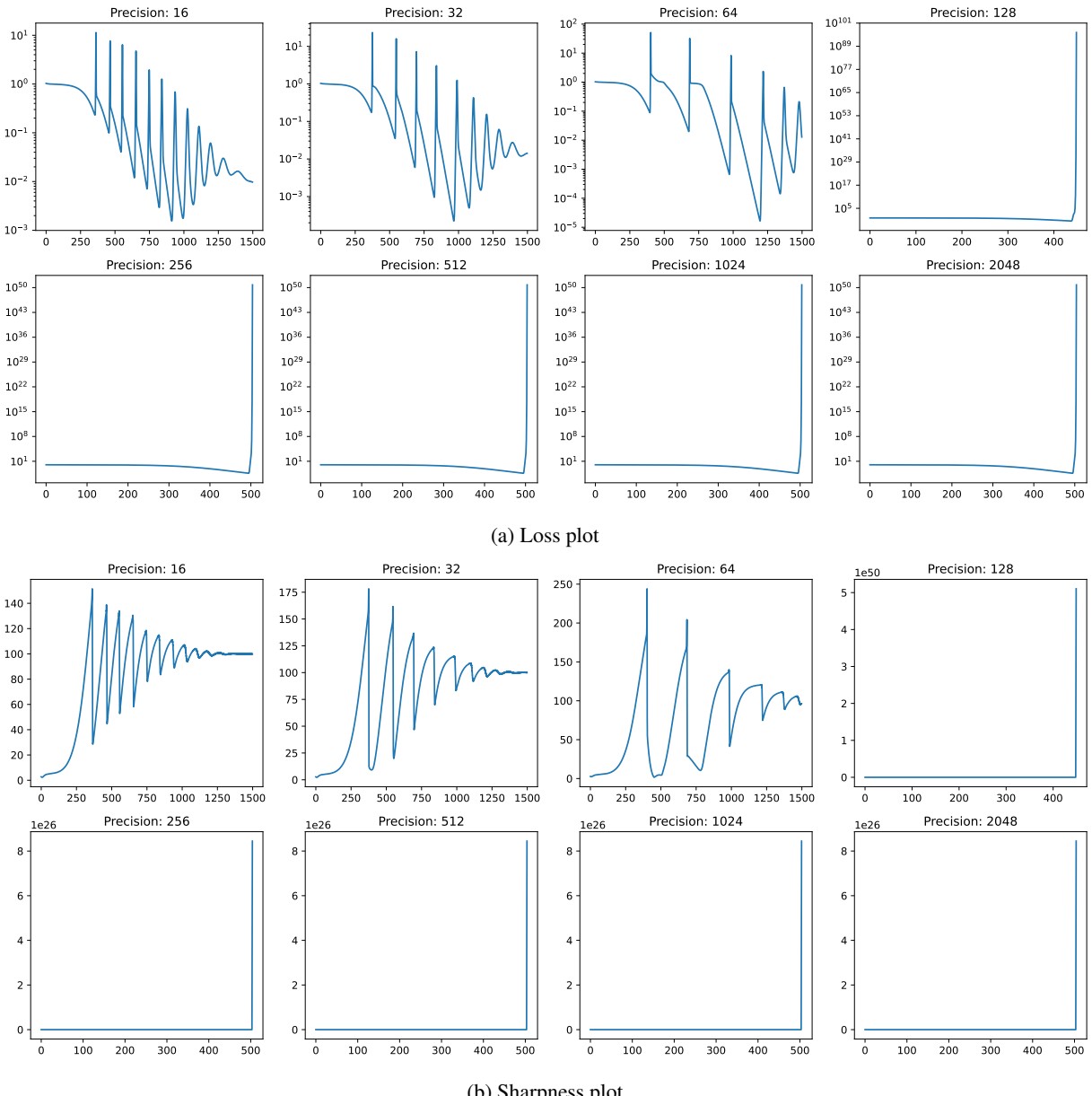

*Figure 28.* $D = 2$ minimal model in our C++ framework training, x axis is given as training step numbers

### D.2. Experiments in a 2-label Subset of CIFAR10 Dataset

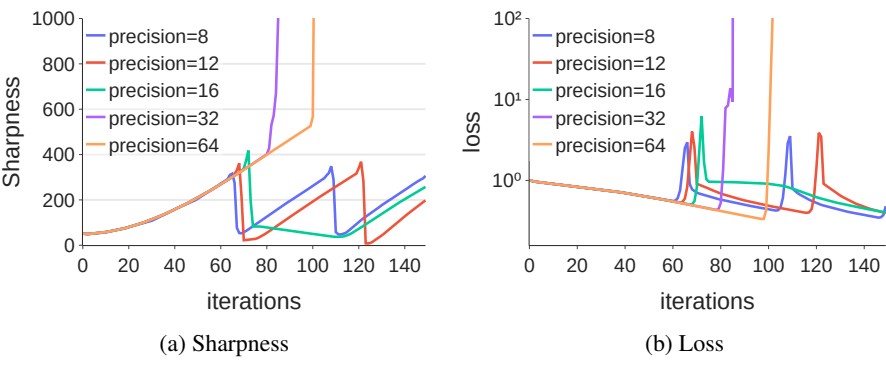

(a) Sharpness

(b) Loss

*Figure 29.* The effect of precision in minimalist model(D=2), with CIFAR-10 2 label subset $N = 300$, GD $\eta = \frac{2}{200}$

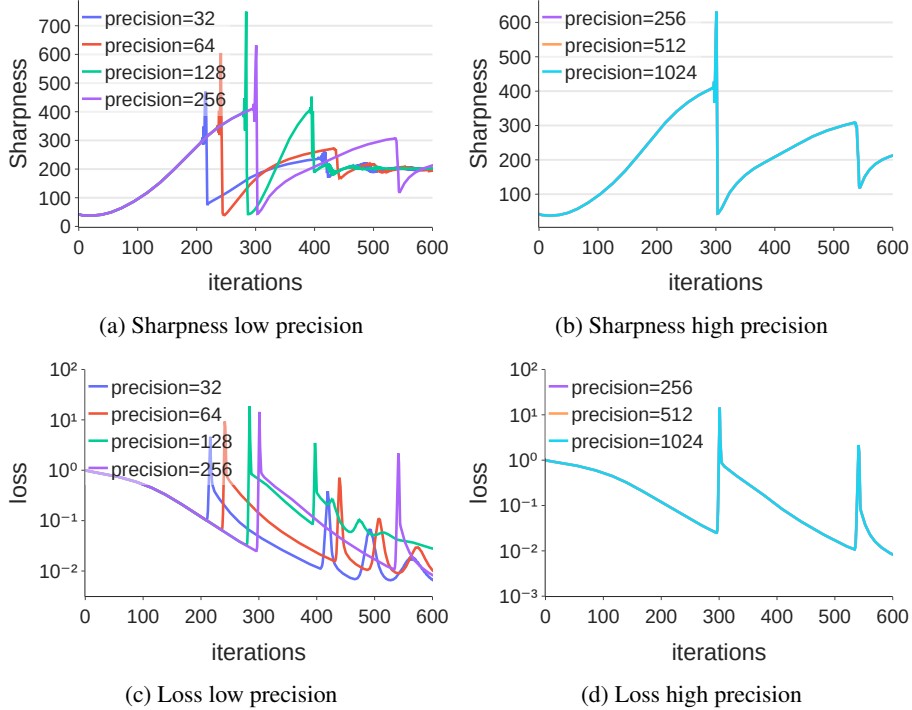

(a) Sharpness low precision

(b) Sharpness high precision

(c) Loss low precision

(d) Loss high precision

*Figure 30.* The effect of precision in realistic model (3 layer SiLU activated NN, width=32), and realistic data CIFAR 2 label subset $N = 300$, GD $\eta = \frac{2}{200}$.

To check the generality of the observations in Appendix D.1, we show our experimental results from our own framework. For the training data, we used a 2-label subset of CIFAR10 dataset.

For the experimental result in Figure 29, we trained our minimalist model (width=1, depth=2) on a CIFAR-2 subset ($N = 300$). The results confirm that higher precision leads to a blowup. Additionally, Figure 29a shows that increased precision postpones the sharpness drop (catapult phase).

For the experimental result in Figure 30, we trained a 3-layer SiLU-activated NNs (width=32) on a CIFAR-2 subset ($N = 300$). While all settings do not exhibit a blowup, higher precision delays the catapult phase until extremely high precision ($\geq 256$). Beyond this threshold, the training dynamics become nearly identical, with completely overlapping curves.

### D.3. Hypothesizing Precision Dependence in Self-Stabilization Framework

To elaborate on the intuition behind this dependency, we briefly introduce Section 4 of Damian et al. (2023)[3]. They introduce a concise 2-dimensional description of gradient descent dynamics near the edge of stability by Taylor-expanding the loss gradient $\nabla L$ around a fixed reference point $\theta^\star$ (which can be understood as the first $\theta$ iterate that reaches sharpness $2/\eta$). Two key quantities are defined as follows:

- $x_t = u \cdot (\theta_t - \theta^\star)$ measures the displacement from $\theta^\star$ along the unstable (top eigenvector of $\nabla^2 L(\theta^\star)$) direction $u$.

- $y_t = \nabla S(\theta^\star) \cdot (\theta_t - \theta^\star)$ quantifies the change in sharpness from its threshold value $2/\eta$.

While assuming that progressive sharpening happens at scale of $\alpha$: $-\nabla L(\theta^\star) \cdot \nabla S(\theta^\star) = \alpha > 0$, the dynamics are described in three stages cycling throughout:

- **Progressive Sharpening**: For small $x_t$ and $y_t$, the sharpness increases linearly: $y_{t+1} - y_t \approx \eta\alpha, (\alpha > 0)$. The update for $x_t$ is: $x_{t+1} \approx -(1 + \eta y_t)x_t$. Thus, once $y_t$ becomes positive, the factor $(1 + \eta y_t) > 1$ causes $|x_t|$ to grow. These updates rely on the 1st-order approximation of the gradient.

- **Blowup**: With $y_t > 0$, the multiplicative effect in the $x$-update leads to exponential growth in $|x_t|$, marking the blowup phase where the 1st-order approximation no longer suffices.

- **Self-Stabilization**: When $|x_t|$ is large, the 2nd-order approximation of gradient (which involves $\nabla^3 L$) yields: $y_{t+1} - y_t \approx \eta \left( \alpha - \frac{\beta}{2} x_t^2 \right)$, which provides a negative feedback that reduces $y_t$ and stops further growth of $|x_t|$.

Note from above that when $y_t < 0$, (i.e., before blowup), $|x_t|$ shrinks to zero exponentially. We hypothesize that higher numerical precision allows $|x_t|$ to remain small for longer iterations, postponing blowup and self-stabilization, resulting in abnormally high sharpness.

---

[3]We adopt here the notation of Damian et al. (2023).

# E. Preliminary Analysis of Nonlinear Activations

We present preliminary results extending our minimalist model to networks with nonlinear activation functions.

**Assumption E.1** (Orthogonal Data Assumption). $XX^\top$ is a diagonal matrix.

Note that each $e_i$ is given as the standard basis vector under the Assumption E.1. Then we define our two-layer non-linear minimalist model and its corresponding layer imbalance as follows:

**Definition E.2** (Two-layer Non-linear Minimalist Model). We define our two-layer non-linear minimalist model $f : \mathbb{R}^d \to \mathbb{R}$ as

$$f(x; \theta) := h(x^\top u)v_1,$$

where $\theta = (u, v_1)$ represents the collection of all model parameters and $h : \mathbb{R} \to \mathbb{R}$ is given as an activation function that satisfies the following conditions:

- $h$ is an injective function.
- $h \in C^1$.
- $h'(x) \neq 0, \forall x \in \mathbb{R}$.

We define $g$ as an antiderivate of $\frac{h}{h'}$.

**Definition E.3** (Layer imbalance in non-linear minimalist model). The layer imbalance of the two-layer non-linear minimalist model with parameter $\theta = (u, v_1)$ is defined by

$$C(\theta) := 2\sum_{i=1}^{r} \frac{g(\sigma_i o_i)}{\sigma_i^2} - v_1^2,$$

Definition E.3 can be viewed as the generalized version of Definition 4.2 [4]. For an arbitrary vector $x = (x_1, x_2, \ldots, x_n)^\top \in \mathbb{R}^n$, we denote $h(x) = (h(x_1), h(x_2), \ldots, h(x_n))^\top \in \mathbb{R}^n$. Then, the loss function is given by $L(\theta; X) = \frac{1}{2N}\|h(Xu)v_1 - y\|^2$, and $o_i = w_i^\top u$. We propose a useful lemma analogous to Lemma 5.2.

**Lemma E.4** (Conservation of layer imbalance in non-linear minimalist model under gradient flow). *Let $\theta(t)$ be a gradient flow trajectory trained on the two-layer non-linear minimalist model. Then, the layer imbalance remains constant along the gradient flow trajectory: $C(\theta(t)) = C(\theta(0))$ for all $t \geq 0$.*

*Proof.* It suffices to prove that $\dot{C}(\theta(t)) = 0$. Recall that the layer imbalance of two-layer non-linear minimalist model is defined as

$$C(\theta(t)) := 2\sum_{i=1}^{r} \frac{g(\sigma_i o_i(t))}{\sigma_i^2} - v_1(t)^2,$$

Differentiating both sides with respect to $t$ gives

$$\dot{C}(\theta(t)) = 2\left(\sum_{i=1}^{r} \frac{\dot{o}_i(t)}{\sigma_i} \frac{h(\sigma_i o_i(t))}{h'(\sigma_i o_i(t))}\right) - 2\dot{v}_1(t)v_1(t)$$

$$= -\frac{2}{N}\sum_{i=1}^{r} v_1(t)\left(e_i^\top z(t)\right)h(\sigma_i o_i(t)) + \frac{2}{N}\sum_{i=1}^{r} h(\sigma_i o_i)\left(e_i^\top z(t)\right)v_1(t)$$

$$= 0$$

$\square$

**Proposition E.5** (Solutions on non-linear activation). *Under Assumption E.1, for a two-layer non-linear minimalist model trained on a dataset $(X, y)$, the following equality holds at the global minimizer $\theta^\star = (u^\star, v^\star)$ of $L(\theta)$:*

$$(v_1^\star)^2 + C(\theta^\star) = 2\sum_{i=1}^{r} \frac{g(\sigma_i o_i^\star)}{\sigma_i^2} \tag{35}$$

---

[4]Given $h(x) = x, \forall x \in \mathbb{R}$, $g(x) = \frac{1}{2}x^2$. It results in the same definition of the imbalance.

*where* $o_i^\star = w_i^\top u^\star = \frac{1}{\sigma_i} h^{-1}\left(\frac{d_i}{v_1^\star}\right)$,

*Proof.* Let $\theta^\star = (u^\star, v_1^\star)$ be a global minimizer of $L(\theta)$ for a two-layer non-linear minimalist model trained on a dataset $(X, y)$ that holds Assumption E.1. We denote $o_i^\star = w_i^\top u^\star$ for each $i \in [r]$. Since $L(\theta^\star) = \frac{1}{2N}\|z(\theta^\star)\|^2$, the residual $z(\theta^*)$ is a zero vector. Combining with (37), we have

$$e_i^\top z(\theta^\star) = h(\sigma_i o_i^\star) v_1^\star - d_i = 0, \quad \forall i \in [r].$$

Moreover, we have

$$C(\theta^\star) = 2 \sum_{i=1}^{r} \frac{g(\sigma_i o_i^\star)}{\sigma_i^2} - (v_1^\star)^2,$$

by the definition of the layer imbalance. Substituting $o_i^\star = \frac{1}{\sigma_i} h^{-1}(\frac{d_i}{v_1^\star})$ finishes the proof. □

By solving the equation of Proposition E.5 based on Lemma E.4 under Assumption 5.1, we can specify the solution of the gradient flow dynamics. Based on the selection of $h$, Equation (35) can be expressed in the following form:

**tanh.** In this case, $g(\sigma_i o_i^\star) = \frac{1}{2}\cosh^2(\sigma_i o_i^\star)$ follows the definition of $g$. Then,

$$
\begin{aligned}
C(\theta^\star) &= \sum_{i=1}^{r} \frac{1}{\sigma_i^2} \cosh^2(\sigma_i o_i^\star) - (v_1^\star)^2 \\
&= \sum_{i=1}^{r} \frac{1}{\sigma_i^2} \frac{1}{1 - \tanh^2(\sigma_i o_i^\star)} - (v_1^\star)^2 \\
&= \sum_{i=1}^{r} \frac{1}{\sigma_i^2} \frac{(v_1^\star)^2}{(v_1^\star)^2 - d_i^2} - (v_1^\star)^2,
\end{aligned}
$$

where the last equality used $o_i^\star = \frac{1}{\sigma_i} \tanh^{-1}\left(\frac{d_i}{v_1^\star}\right)$.

**sigmoid.** In this case, $g(\sigma_i o_i^\star) = \exp(\sigma_i o_i^\star) + \sigma_i o_i^\star$ follows the definition of $g$. Then,

$$
\begin{aligned}
C(\theta^\star) &= 2 \sum_{i=1}^{r} \frac{\sigma_i o_i^\star + \exp(\sigma_i o_i^\star)}{\sigma_i^2} - (v_1^\star)^2 \\
&= 2 \sum_{i=1}^{r} \frac{\ln\left(\frac{d_i}{v_1 - d_i}\right) + \frac{d_i}{v_1 - d_i}}{\sigma_i^2} - (v_1^\star)^2,
\end{aligned}
$$

where the last equality used $o_i^\star = \frac{1}{\sigma_i} \ln(\frac{d_i}{v_1 - d_i})$.

While we consider these two cases to be relatively simple compared to other nonlinear activation functions, a thorough analysis under highly nonlinear equation remains necessary and is therefore deferred to future work.

### E.1. Reparameterization of the Two-layer Non-linear Minimalist Model

In this subsection, we reparameterize the gradient flow (GF) dynamics for the two-layer non-linear minimalist model (Definition E.2) under the Assumption E.1 in terms of $\sigma_i$, $d_i$, $o_i$ for $i \in [r]$, and $v_1$.

**Network output and residual.** We can decompose network output $f(X; \theta)$ into $e_1, \dots, e_r$ components by

$$f(X; \theta) = \sum_{i=1}^{r} (e_i^\top f(X; \theta)) e_i = \sum_{i=1}^{r} \left(e_i^\top h(Xu) v_1\right) e_i = \sum_{i=1}^{r} (h(\sigma_i o_i) v_1) e_i, \tag{36}$$

where the last equality is obtained by replacing $e_1^\top h(Xu)$ with $h(e_1^\top Xu)$, and $X$ with $\sum_{i=1}^r \sigma_i(e_i w_i^\top)$. Similarly, we can decompose residual $z(\theta)$ by

$$z(\theta) = f(X;\theta) - y = \sum_{i=1}^r (h(\sigma_i o_i)v_1 - d_i)\, e_i\,. \tag{37}$$

Hence, network output (36) and residual (37) can be reparameterized in terms of $\sigma_i$, $d_i$, $o_i$ for $i \in [r]$, and $v_1$.

**Gradient Flow.** Recall that the loss at $\theta$ is $L(\theta) = \frac{1}{2N}\|f(X;\theta) - y\|^2 = \frac{1}{2N}\|z(\theta)\|^2$. The GF dynamics is given by

$$\dot{u}(t) = -\frac{\partial L}{\partial u}(t) = -\frac{1}{N}v_1(t)X^\top[z(t) \odot h'(Xu(t))]\,, \tag{38}$$

$$\dot{v}_1(t) = -\frac{\partial L}{\partial v}(t) = -\frac{1}{N}h(Xu(t))^\top z(t)\,. \tag{39}$$

We can replace $X$ with $\sum_{i=1}^r \sigma_i(e_i w_i^\top)$ in (39) and obtain

$$\dot{v}_1(t) = -\frac{1}{N}\sum_{i=1}^r h(\sigma_i o_i(t)e_i)^\top z(t) = -\frac{1}{N}\sum_{i=1}^r h(\sigma_i o_i(t))e_i^\top z(t)\,.$$

Hence, we have

$$\dot{v}_1(t) = -\frac{1}{N}\sum_{i=1}^r h(\sigma_i o_i(t))e_i^\top z(t)\,. \tag{40}$$

Similarly, inner product with $w_i$ to both hand sides of (38) and replacing $X$ with $\sum_{i=1}^r \sigma_i(e_i w_i^\top)$ gives

$$\dot{o}_i(t) = w_i^\top \dot{u}(t) = -\frac{1}{N}\sigma_i\left(e_i^\top z(t)\right) h'(\sigma_i o_i)v_1\,, \quad \forall i \in [r]. \tag{41}$$

Therefore, (40) and (41) together give the reparameterization of the GF dynamics.

