# OpenReview forum: "Understanding Sharpness Dynamics in NN Training with a Minimalist Example: The Effects of Dataset Difficulty, Depth, Stochasticity, and More"
_ICML.cc/2025/Conference — ICML 2025 poster_

### Official Review · Reviewer_boFk · 2025-03-03

**Overall Recommendation:** 3

**Summary:**

The paper investigates sharpness dynamics in neural network training using a minimalist deep linear network with one neuron per layer. It theoretically and empirically shows that this model captures progressive sharpening and edge-of-stability behaviors observed in practice. Key contributions include:
1. Identifying dataset difficulty $Q$, depth, batch size, and learning rate as factors influencing sharpness.
2. Deriving theoretical bounds on sharpness at minima, dependent on $Q$ and depth.
3. Demonstrating that SGD’s stochasticity reduces progressive sharpening compared to GD via layer imbalance dynamics.


## update after rebuttal
The authors have done a good job explaining their work. My concerns about how to generalize this framework to non-linear models remain. Nevertheless, I believe this paper should be accepted, and I would retain my score.

**Claims And Evidence:**

Claim i: The minimalist model replicates sharpness dynamics (progressive sharpening, edge of stability).
- Evidence: Figures 3–5 show sharpness trajectories in the model matching practical networks (e.g., Transformers).

Claim ii: Sharpness at minima is bounded by $\sigma_1^2 Q^{(D-1)/D}/N$ (Theorem 4.6).
- Evidence: Table 3 and Figure 7 show strong correlation between predicted and empirical sharpness.

Claim iii: SGD increases layer imbalance $C(\theta)$, reducing sharpness (Theorem 5.6).
- Evidence: Figure 6 shows $C(\theta)$ grows faster with smaller batch sizes/larger learning rates, aligning with reduced sharpness in Figure 3.

**Essential References Not Discussed:**

I am not aware of.

**Experimental Designs Or Analyses:**

- GF/GD/SGD trajectories: Figures 3–6 validate sharpness dynamics and layer imbalance. The experiments look reasonable to me. In figure 4, I am puzzled by why RK4 is used.
- Figure 7 is good way to demonstrate the upper bound. It looks reasonable to me as well.

**Methods And Evaluation Criteria:**

The proposed methods and evaluation criteria are appropriate and well-justified for studying sharpness dynamics:
   - Synthetic Data: Validates theoretical bounds under controlled settings (Sec 4.1).
   - Real Data: 2-class subsets of CIFAR10/SVHN balance tractability and realism.
   - Correlation Analysis: Tests generalizability of $\hat{S}_D$ to nonlinear networks (Fig 7, Table 5).

**Other Comments Or Suggestions:**

It would be very interesting if the authors can try to run some experiments that are closer to real-world usages. Such as GPT-2 level small experiments or some proper image classification tasks. I would like to see how far this simple model can go.

**Other Strengths And Weaknesses:**

Strengths:
- Novel insights into dataset difficulty and depth’s role in sharpness.
- Clear empirical validation across architectures/activations (Table 5, Fig 7).

Weaknesses:
- Lack of nonlinear dynamics analysis (e.g., ReLU, attention).
- It is a good model but feels like a marginal improvement over the "convex-hull" of existing works.

**Questions For Authors:**

1. How does dataset difficulty $Q$ generalize to non-linear networks? I knew technically there is no difficulty defining it, but I wonder if the intuition still holds.
2. Why does $\hat{S}_D$ correlate with empirical sharpness even under non-balanced initializations (Fig 7)?  Could the authors provide some intuition?
3. Could authors provide any intuition on why the precision matters so much?

**Relation To Broader Scientific Literature:**

This work connects to:
- Progressive sharpening (Cohen et al., 2021) and edge of stability (Damian et al., 2023).
- SGD vs GD sharpness tradeoffs (Agarwala & Pennington, 2024).
- Similar minimalist example like Zhu et al. 2023, Kalra et al. 2023, where here the model captures convergence behavior.
Extends prior work by analyzing dataset difficulty $Q$ and depth $D$ in a tractable model.

**Theoretical Claims:**

The theoretical claims are solid.

I checked Sharpness Bounds (Theorems 4.3, 4.6):
   - Derived via NTK spectral analysis (Appendix A.2–A.3).
   - Key Step: Expressing the NTK matrix in terms of $\sigma_i$, $d_i$, $Q$, then bounding its spectral norm.

I did not carefully check Layer Imbalance Dynamics (Lemma 5.2, Theorem 5.6) but it looks reasonable to me.

The analysis assumes GF converges to global minima (Assumption 5.1). While empirically validated, formal convergence guarantees are not provided.

---

> ### Author Rebuttal · Authors · 2025-04-01
>
> We deeply appreciate your constructive feedback. Below are our responses to your questions and suggestions.
>
> ## Extending our work to more realistic scenarios
>
> Thank you for the insightful suggestion. To explore the applicability of our findings in more realistic settings, we conducted additional experiments training CNNs on the CIFAR-2 subset ($N=100$). The experimental setup follows the description in Section 5.1, with the only difference being that we fixed the random seed for weight initialization across runs. Our CNN architecture of depth $D=3$ consists of two convolutional layers (32 channels) with average pooling, followed by a fully connected layer.
>
> For predicted sharpness $\hat S_D$, we set $D=3$ that matches the depth of our CNN. The Pearson correlation between actual and predicted sharpness is summarized below:
>
> | identity | tanh | SiLU | ReLU |
> | --- | --- | --- | --- |
> | 0.7335 | 0.5723 | 0.3561 | 0.4314 |
>
> While the correlation decreases with non-linear activations, the identity activation still achieves a correlation above 0.7, suggesting that the intuition behind predicted sharpness may extend to CNNs. However, the behavior under non-linear activations is less clear. A deeper investigation into broader architectures remains an interesting direction for future work.
>
> ## How does $Q$ generalize to non-linear networks & Lack of nonlinear dynamics analysis
>
> We would like to clarify that $Q$ is a quantity that depends solely on the dataset, not on the architecture of the neural network nor the non-linearity of the activation function. Therefore, we interpret the question on $Q$ as asking whether a nonlinear dynamics analysis is possible. If our understanding is incorrect, please let us know.
>
> We now outline our initial attempt at analyzing non-linear networks. Assuming that $XX^\top$ is a diagonal matrix, we can analyze our minimal model with $D=2$. For such $X$’s, each $e_i$ is given as the standard basis vector.
>
> The loss function is given by $L(\theta; X) = \frac{1}{2N} \lVert h(Xu) v_1  - y \rVert^2$, and $o_i = e_i^\top h(Xu) = h(e_i^\top Xu)$. We consider a non-linear activation function $h$ that is injective, continuously differentiable, and has non-vanishing derivatives.
>
> The gradient flow dynamics for $o_i$ and $v_1$ are given by:
>
> $$
> \dot o_i = - \frac{1}{N}v_1 \sigma_i  \left[(v_1h(e_i^\top Xu) - y) \cdot h'( e_i^\top Xu) \right]
> = - \frac{1}{N}v_1 \sigma_i  \left[(v_1h(\sigma_i o_i) - d_i) \cdot h'(\sigma_i o_i) \right].
> $$
>
> and
>
> $$
> \dot v_1 = - \frac{1}{N} \sum_{i=1}^r h(e_i^\top Xu)  \left[e_i^\top \left(v_1h(Xu) - y \right) \right]
> = - \frac{1}{N} \sum_{i=1}^r h(\sigma_i o_i)  \left[ \left(v_1h(\sigma_i o_i) - d_i \right) \right] .
> $$
>
> We can derive the following equation:
>
> $$
> v_1 \dot v_1 = \sum_{i=1}^r \frac{\dot o_i}{\sigma_i} \frac{h(\sigma_i o_i)}{h'(\sigma_i o_i)} .
> $$
>
> Let $g$ be an antiderivative of $\frac{h}{h'}$. Integrating both sides gives:
>
> $$
> \frac{1}{2} v_1^2 =  C+ \sum_{i=1}^r \frac{1}{\sigma_i^2} g(\sigma_i o_i).
> $$
>
> Assuming the convergence of the loss, we have:
>
> $$
> o_i = \frac{1}{\sigma_i} h^{-1}\left( \frac{d_i}{v_1} \right).
> $$
>
> Therefore, at convergence, the following equation holds:
>
> $$
> \frac{1}{2}v_1^2 = C + \sum_{i=1}^r \frac{1}{\sigma_i^2} g\left(h^{-1}\left( \frac{d_i}{v_1} \right)\right).
> $$
>
> This results in a highly non-linear equation of $v_1$ which requires further investigation to fully understand its solutions $v_1^\star$. As such, we have decided to leave these parts for future work.
>
> ## Why does $\hat S_D$ correlate with empirical sharpness under non-balanced initialization?
>
> For the sake of theoretical analysis, our analysis employs a balanced initialization. Nevertheless, we anticipate that similar trends would be observed under a non-balanced initialization scheme. Moreover, owing to the network's relatively large width in our non-linear network experiments, we believe that the randomness in initialization is effectively "averaged out," leading to consistent results across different random seeds.
>
> ## Our minimalist model is a marginal improvement
>
> We agree that the minimalist model, on its own, builds upon previous works and is not the core novelty of our study. Instead, our primary contribution lies in rigorously connecting this model to the phenomenon of progressive sharpening—a relationship that has been underexplored in theoretical analyses. Through this, we derive novel insights into how problem parameters influence the degree of progressive sharpening during training. We believe these results advance the understanding of sharpening dynamics in ways that prior studies have not addressed.
>
> ## Why does precision matter so much?
>
> Please refer to our answer to Reviewer smTr.
>
> ## RK4 in Figure 4
>
> Please refer to our answer to Reviewer 2e3Z.

---

### Official Review · Reviewer_smTr · 2025-03-09

**Overall Recommendation:** 4

**Summary:**

This paper analyzes sharpness dynamics and characterizes the effect of dataset complexity, depth, batch size (Phenomena 1). Furthermore, they show a simplified model, a deep linear network with unit width trained on multiple examples, captures the Phenomena 1 and analyzes it in detail. By analyzing the sharpness at convergence and properties of the Gradient Flow, they show that the rate of progressive sharpening increases with depth and data complexity. They also argue the effect of batch noise and learning rate in SGD setting.

**Claims And Evidence:**

Claims:

* (Empirical) Phenomena 1: Sharpness increases with (1) Dataset size, (2) Depth and decreases at smaller batch sizes (Figure 1 and 2)
* (Empirical) The simplified model empirically captures Phenomena 1
* (Theoretical) The theoretical analysis (+ empirical analysis Tables) of the simplified model captures the effect of dataset complexity and depth.
* The effect of batch size and learning rate in SGD case is captured in restricted settings (depth 2)

**Essential References Not Discussed:**

I think the work adequately discusses existing works.

**Experimental Designs Or Analyses:**

The experimental design and analyses are sound.

**Methods And Evaluation Criteria:**

The method and evaluations are solid.

**Other Comments Or Suggestions:**

Suggestions:

* For equation 3, it might be better to use some other variable than d as its already used for input dimension

**Other Strengths And Weaknesses:**

Strengths

* The paper is clearly written and the claims are justified well.

Weakness

* For D > 2, only balanced initializations are examined
* The GD/SGD analysis is restricted to D = 2

**Questions For Authors:**

Questions:

* Effect of precision on EoS: if high precision can cause loss to diverge, then the theoretical analysis should suggest divergences. But typical theoretical analysis does not suggest that. Do the authors have some understanding of this?

* I am not familiar with step size selection for Runge Kutta methods. Why was the step size chosen to be proportional to inverse sharpness and recomputed periodically?

* In Figure 2, do the authors understand why does the sharpness decrease --> increase --> decrease --> finally increase? Typically, GD trajectories are known to show three types of dynamics [1]: (1) increase throughout, (2) decrease throughout, and (3) decrease then decrease. Is it because of the Runge-Kutta step size selection?

[1] Gradient Descent on Neural Networks Typically Occurs at the Edge of Stability
https://arxiv.org/abs/2103.00065

[2] Universal Sharpness Dynamics in Neural Network Training: Fixed Point Analysis, Edge of Stability, and Route to Chaos

**Relation To Broader Scientific Literature:**

The paper improves our understanding of progressive sharpness by examining the effect of depth, dataset complexity and batch size. The effect of smaller batch size reducing the rate of progressive sharpening has been known in prior work, such as Ref. [1].

[1] Agarwala, A. and Pennington, J. High dimensional analysis
reveals conservative sharpening and a stochastic edge of
stability. arXiv preprint arXiv:2404.19261, 2024

**Theoretical Claims:**

The theoretical claims are sound. I have verified the claims in the main text and also verified dynamical equations for GF, GD and SGD.

---

> ### Author Rebuttal · Authors · 2025-04-01
>
> We appreciate your valuable feedback. Below, we address your comments.
>
> ## Balanced initializations for D>2 & GD/SGD analysis only on D=2
>
> While it is true that the assumption of balanced initialization for $D > 2$ may appear restrictive, we believe it is justified for the following reasons:
>
> - In line 728 of our paper, we implicitly use that condition to derive the GF limit point $v_1^\star$. Without it, the resulting equation becomes a polynomial of degree $D$, whose closed-form solution can’t be derived when $D\geq 5$. For $2<D<5$ cases, the solutions are also hard to interpret. Although the roots might be solvable numerically, we prioritized presenting interpretable analytical bounds that highlight the effect of depth.
> - Balanced initialization is also a widely adopted assumption in the literature of deep linear networks [2, 3].
> - As shown in Figure 7, experimental results that don’t satisfy balanced initialization align with our theoretical predictions.
>
> In the case of GD/SGD analysis, the balanced constant $C$ does not remain fixed throughout training. As a result, even if training starts from a balanced initialization for $D > 2$, the balancedness may not be preserved during the optimization process.  For this reason, we limited our analysis to the two-layer case for intuitive understanding.
>
> ## Effect of precision on EoS in typical theory
>
> Few works mention the precision as a meaningful parameter for EoS phase. We found one work [1, Appendix D.3], where they also mention that EoS dynamics is sensitive to the precision.
>
> To elaborate the intuition, we’ll briefly introduce Section 4 of [1], which introduces a concise 2-dimensional description of gradient descent dynamics near the edge of stability by Taylor-expanding the loss gradient $\nabla L$ around a fixed reference point $\theta^\star$ (which can be understood as the first $\theta$ iterate that reaches sharpness $2/\eta$). Two key quantities are defined:
>
> - $x_t = u \cdot (\theta_t - \theta^\star)$ measures the displacement from $\theta^\star$ along the unstable (top eigenvector of $\nabla^2 L(\theta^\star)$) direction $u$.
> - $y_t = \nabla S(\theta^\star) \cdot (\theta_t - \theta^\star)$ quantifies the change in sharpness from its threshold value $2/\eta$.
>
> We assume that progressive sharpening happens at scale of $\alpha$: $-\nabla L(\theta^*) \cdot \nabla S(\theta^*) = \alpha > 0$. The dynamics are described in three stages cycling throughout:
>
> - **Progressive Sharpening:**
>
>     For small $x_t$ and $y_t$, the sharpness increases linearly: $y_{t+1} - y_t \approx \eta \alpha \, (\alpha > 0).$
>
>     The update for $x_t$ is: $x_{t+1} \approx -(1+\eta y_t) x_t .$
>
>     Thus, once $y_t$ becomes positive, the factor $(1+ \eta y_t) > 1$ causes $|x_t|$ to grow. These updates rely on the 1st-order approximation of the gradient.
>
> - **Blowup:**
>
>     With $y_t > 0$, the multiplicative effect in the $x$-update leads to exponential growth in $|x_t|$, marking the blowup phase where the 1st-order approximation no longer suffices.
>
> - **Self-Stabilization:**
>
>     When $|x_t|$ is large, the 2nd-order approximation of gradient (which involves $\nabla^3 L$) yields: $y_{t+1} - y_t \approx \eta \left( \alpha - \frac{\beta}{2} x_t^2\right),$ which provides a negative feedback that reduces $y_t$ and stops further growth of $|x_t|$.
>
>
> Note from above that when $y_t < 0$, (i.e., before blowup), $|x_t|$ shrinks to zero **exponentially**. We hypothesize that higher numerical precision allows $|x_t|$ to remain small for longer iterations, postponing blowup and self-stabilization, resulting in abnormally high sharpness. This is further supported by the experimental results discussed in our response to Reviewer 2e3Z.
>
> ## Why is RK4 step size proportional to the inverse sharpness?
>
> The RK4 step size is chosen based on its linear stability range. For the ODE $\frac{dy}{dt} = \lambda y$ with $\lambda \in \mathbb{R}$, RK4 is stable if $-2.785 \leq \lambda \eta \leq 0$, where $\eta$ is the step size. Since we already flip the sign in gradient flow, using a step size proportional to the inverse sharpness ensures stable integration.
>
> Further rationale on our choice of RK4 can be found in our response to Reviewer 2e3Z.
>
> ## Sharpness dynamics in Figure 2
>
> To clarify, Figure 2 illustrates GD and SGD dynamics (Appendix B.1), not RK4. The observed sharpness patterns are thus likely due to problem-specific characteristics of the dynamics. However, our work focuses on the end-to-end behavior, not intermediate dynamics, leaving this as an open question.
>
> ---
>
> [1] Damian et al., 2023, Self-Stabilization: The Implicit Bias of Gradient Descent at the Edge of Stability, ICLR.
>
> [2] Arora et al., 2018, On the Optimization of Deep Networks: Implicit Acceleration by Overparameterization, ICML.
>
> [3] Arora et al., 2019, A Convergence Analysis of Gradient Descent for Deep Linear Neural Networks, ICLR.

---

> > ### Comment · Reviewer_smTr · 2025-04-03
> >
> > I thank the authors for their rebuttal. Most of my concerns have been resolved. As my initial score was already high, I would like to keep my score.

---

> > > ### Author Response · Authors · 2025-04-09
> > >
> > > Thank you for your positive feedback and for taking the time to review our work. We are glad to hear that our rebuttal addressed most of your concerns. We are grateful for your thoughtful assessment throughout the review process.

---

### Official Review · Reviewer_2e3Z · 2025-03-10

**Overall Recommendation:** 4

**Summary:**

This paper first shows that a "minimalist model"--one that has a single unit per layer with linear activations--can effectively captures a recently observed phenomenon called "progressive sharpening", where the sharpness of the loss increases as training progresses. This sharpness then stabilizes around $2/\eta$ ($\eta$ being the learning rate), a regime called the "edge of stability". The paper then defines several metrics: dataset difficulty and layer imbalance, which are used in theoretical bounds on the sharpness at the global minimum.

## Update after rebuttal

I am now more confident in recommending acceptance. The rebuttal text should be added to the paper or the supplementary, and would improve its presentation and overall quality.

**Claims And Evidence:**

The authors test the effect of dataset size, network depth, batch size, and learning rate in progressive sharpening, and the theory derived lines up well with their experiments.

While the datasets used in the paper are common and with good reason (for example, SVHN is "harder" than CIFAR-10, and this paper's results align with that), I'm concerned that *only* using these two limits the generalizability of the results. Most of this paper's results solely rely on CIFAR-10 (Figure 1-4), so it's unclear how well they translate to other datasets, such as those with different modalities.

Another issue along these lines is in Appendix C, where the authors seem to have run the experiment for a single seed, on a single, randomly-generated dataset, with one set of parameters passed to the `minimal_data` function, and only using the minimal model with $D=2$. This limits how representative the results are.

**Essential References Not Discussed:**

I did not find any crucially missing references, though the authors should cite the LOBPCG paper since they use and mention it explicitly:

[1] Knyazev, A. V. (2001). Toward the optimal preconditioned eigensolver: Locally optimal block preconditioned conjugate gradient method. SIAM journal on scientific computing, 23(2), 517-541.

**Experimental Designs Or Analyses:**

I have detailed these concerns under Claims and Evidence.

**Methods And Evaluation Criteria:**

This is an interesting paper. For how surprising some results are, the authors do a good job designing experiments to show claimed effects.

However, I believe the paper misses one aspect surrounding the variables it tests. While the paper tests the effect of learning rate and batch size separately on the sharpness, recent work has shown that these two determine the dynamics of SGD *jointly* as a ratio instead, see [1-2] below. Therefore, this paper would also benefit from performing its experiments with the *ratio* as a parameter.

[1] Jastrzębski, S., Kenton, Z., Arpit, D., Ballas, N., Fischer, A., Bengio, Y., & Storkey, A. (2017). Three factors influencing minima in sgd. arXiv preprint arXiv:1711.04623.
[2] Smith, S. L., Kindermans, P.-J., Ying, C., & Le, Q. V. (2018). Don’t Decay the Learning Rate, Increase the Batch Size. International Conference on Learning Representations.

**Other Comments Or Suggestions:**

None.

**Other Strengths And Weaknesses:**

This paper could benefit from better presentation in a couple of instances:

* On page 4, under "Progressive Sharpening", the authors state: "In Figure 3 and Figure 4, we can observe the same trend described in Phenomenon 1 [...]". This would be a good chance for Figure 3 to also show the edge of stability in the same plots. Figure 4 already does this fairly well.
* In Section 4, the $o_i$ terms are used without definition.

**Questions For Authors:**

1. For Figure 4 (and Figure 1, as stated in Appendix B.1), why do you use the Runge-Kutta method? It's possible I misunderstood, but my read of these figures was that they plot sharpness over optimizer iterations; it's curious to not use something like SGD.
2. For sharpness, why do you specifically choose the maximum Hessian eigenvalue instead of other established measures such as the robustness under adversarial perturbations [1] or the volume of a polytope with a thresholded height [2]? These, especially the former, have been endorsed in the literature [3-4].

[1] Keskar, N. S., Mudigere, D., Nocedal, J., Smelyanskiy, M., & Tang, P. T. P. (2017). On Large-Batch Training for Deep Learning: Generalization Gap and Sharp Minima. International Conference on Learning Representations.
[2] Hochreiter, S., & Schmidhuber, J. (1997). Flat minima. Neural Computation, 9(1), 1–42.
[3] Dinh, L., Pascanu, R., Bengio, S., & Bengio, Y. (2017, July). Sharp minima can generalize for deep nets. In International Conference on Machine Learning (pp. 1019-1028). PMLR.
[4] Andriushchenko, Maksym, Francesco Croce, Maximilian Müller, Matthias Hein, and Nicolas Flammarion. “A Modern Look at the Relationship between Sharpness and Generalization.” In International Conference on Machine Learning, 840–902. PMLR, 2023.

**Relation To Broader Scientific Literature:**

The contributions of this paper are significant. Although the paper's theory and experiments are currently limited to a small architecture, the authors effectively argue why it is still relevant. Moreover, this paper is an advancement in the field, and other papers in the future will build on it to larger architectures.

**Theoretical Claims:**

I checked the proofs in the main paper and did not find any issues. I skimmed the proofs of the Appendix, where results seem to follow naturally from prior work.

---

> ### Author Rebuttal · Authors · 2025-04-01
>
> We appreciate your insightful feedback. We have provided figures for additional experiments in the supplementary PDF file [[Link]](https://anonymous.4open.science/r/understand_progressive_sharpening-E2F5/Experimental_Results_2e3Z.pdf). Below, we address your comments.
>
> ## Applicability of results across datasets
>
> We have included additional experimental results on SVHN, shown in Figures 1–4 of the supplementary file, confirming that the observed trends hold across both CIFAR-10 and SVHN. To further test our results on a different modality, we conducted experiments on the Google Speech Commands dataset [3]. The results in Figures 7–10 and Tables 1–5 of the supplementary file again exhibit similar trends. These findings suggest that our observations capture general training dynamics rather than being specific to a particular dataset or modality.
>
> ## Additional Experiments on Precision (Appendix C)
>
> We appreciate the reviewer’s concern regarding the generality of our precision experiments in Appendix C. We have conducted additional runs on deep neural networks and a real-world dataset. The updated results are provided in Figures 5 and 6 of the supplementary PDF file.
>
> - **Figure 5:** We trained our minimalist model (width=1, depth=2) on a CIFAR-2 subset ($N=300, \eta=0.01$). The results confirm that higher precision leads to a blowup. Additionally, Figure 5a shows that increased precision postpones the sharpness drop (catapult phase).
> - **Figure 6:** We trained a 3-layer SiLU-activated NNs (width=32) on a CIFAR-2 subset ($N=300, \eta=0.01$). While all settings do not exhibit a blowup, higher precision delays the catapult phase until extremely high precision ($\geq 256$). Beyond this threshold, the training dynamics become nearly identical, with completely overlapping curves. We suspect that a larger width prevents excessive progressive sharpening and blowup.
>
> For further intuition, please refer to our response to Reviewer smTr.
>
> ## Ratio of batch size and learning rate as a key parameter
>
> We appreciate the suggestion to analyze the ratio $\eta/B$ as a key parameter. While we did not explicitly frame our experiments in terms of this ratio, our results in Figure 2 already allow for an implicit comparison. For example, the cases with $B=125, \eta=2/400$ and $B=250, \eta=2/200$ share the same ratio but exhibit slightly different sharpness trends, suggesting that $\eta/B$ alone may not fully capture the observed phenomena.
>
> In large-batch settings ($B \approx N$), our results align with the GD Edge of Stability (EoS) literature [1], showing that sharpness evolution remains largely unchanged before EoS, implying that the ratio has little effect on sharpness dynamics in this regime. Given these observations, we are uncertain about the additional insights gained by treating the ratio as the primary parameter, but we welcome further discussion.
>
> ## Use of Runge-Kutta in Figures 1 & 4
>
> Our choice of the Runge-Kutta (RK4) method follows the setup in [1] (Appendix I.5). As shown in their Figures 29, 31, 33, 35, 37, and 39, RK4 gradient flow closely tracks GD dynamics before EoS. This allows us to analyze training dynamics independently of the learning rate schedule, providing a cleaner theoretical comparison. We will add the relevant citations in the revised manuscript.
>
> ## Choice of sharpness measure
>
> Our sharpness definition follows [1] and [2], as our primary goal is to study progressive sharpening observed in [1] and the transition into the EoS regime. While alternative measures offer valuable insights, they primarily focus on generalization rather than training dynamics. Since our study is centered on the optimization perspective, the Hessian maximum eigenvalue remains the most relevant metric.
>
> ## Minor concerns
>
> >Cite the LOBPCG paper
>
> Thank you for pointing this out. We will add the citation.
>
> >Modify Figure 3 to also show the edge of stability. Figure 4 already does this fairly well.
>
> We would like to clarify that Figure 4 does not depict the edge of stability. The observed sharpness saturation in Figure 4 is not due to the edge of stability but rather reflects sharpness dynamics under gradient flow, simulated using the Runge-Kutta (RK4) method instead of gradient descent. The edge of stability is explicitly presented in Figure 2 for the case where $B = N$, where sharpness saturates at 200 for a learning rate of $\eta = 2/200$.
>
> >$o_i$ is used without definition
>
> In Section 4, we specify that the first-layer weight $u$ can be decomposed as $u=\sum_{i=1}^r o_i w_i + \Pi_W^{\perp}u$, where $w_i$ are the right singular vectors of $X$. Thus, $o_i$ represents the component of $u$ along $w_i$.
>
> ---
>
> [1] Cohen et al., Gradient Descent on Neural Networks Typically Occurs at the Edge of Stability, ICLR 2021.
>
> [2] Damian et al., Self-Stabilization: The Implicit Bias of Gradient Descent at the Edge of Stability, ICLR 2023.
>
> [3] Warden P., Speech Commands: A Dataset for Limited-Vocabulary Speech Recognition, 2018.

---

> > ### Comment · Reviewer_2e3Z · 2025-04-02
> >
> > Thank you for your detailed responses. The authors' rebuttal clearly addresses all of my concerns. Much of their rebuttal text to other reviewers would also be useful additions to the supplementary material. I am updating my score to recommend acceptance.

---

> > > ### Author Response · Authors · 2025-04-09
> > >
> > > We appreciate your careful reading of our response. Your feedback has greatly contributed to improving the quality of our work. Also, we would like to thank you for raising the score. We will incorporate the discussion here into the revised manuscript.

---

### Official Review · Reviewer_47Zn · 2025-03-11

**Overall Recommendation:** 3

**Summary:**

This paper employs a minimalist model to investigate the progressive sharpening phenomenon in the training of deep neural networks. Progressive sharpening is a widely observed phenomenon characterized by the enhancement of sharpness during training using gradient descent or stochastic gradient descent, before reaching a saturation point at the boundary of stability. The primary objective of this study is to explore the relationship between progressive sharpening and various problem parameters, such as dataset size, network depth, batch size, and learning rate. The minimalist model is a regression problem with a single neuron per layer and identity activation.

**Claims And Evidence:**

The provided paper introduces two crucial quantities: the dataset difficulty, denoted as $Q$, and the layer imbalance, denoted as $C(\theta)$. It establishes a bound on the sharpness of the minimizer in terms of these two quantities under specific assumptions. The paper substantiates these claims through both proofs and empirical evidence.

**Essential References Not Discussed:**

I recommend that the paper conduct a more comprehensive literature review on the training dynamics of neural networks and the previous studies of gradient descent (GD) and stochastic gradient descent (SGD). While the authors have made it evident that their study focuses on the influence of problem settings on the progressive sharpening, I believe it would be essential to discuss the significance of this sharpening (implicit regularization) and how it could be utilized to enhance training (sharpness-aware minimization, SGD). Additionally, as previously mentioned, I believe the connection between the proposed minimalist model and previous works should be further elaborated.

To list a few:
(1) Implicit Regularization of SGD: [4, 5, 6]
(2) SAM: [3, 7, 8]
(3) Connections to linear diagonal networks (and its variants): [1, 2, 5, 6, 9, 10]

[3] Long, Philip M., and Peter L. Bartlett. "Sharpness-aware minimization and the edge of stability." Journal of Machine Learning Research 25.179 (2024): 1-20.
[4] Wu, Jingfeng, et al. "Direction matters: On the implicit bias of stochastic gradient descent with moderate learning rate." arXiv preprint arXiv:2011.02538 (2020).
[5] HaoChen, Jeff Z., et al. "Shape matters: Understanding the implicit bias of the noise covariance." Conference on Learning Theory. PMLR, 2021.
[6] Ren, Yinuo, Chao Ma, and Lexing Ying. "Understanding the generalization benefits of late learning rate decay." International Conference on Artificial Intelligence and Statistics. PMLR, 2024.
[7] Andriushchenko, Maksym, and Nicolas Flammarion. "Towards understanding sharpness-aware minimization." International conference on machine learning. PMLR, 2022.
[8] Foret, Pierre, et al. "Sharpness-aware Minimization for Efficiently Improving Generalization." International Conference on Learning Representations.
[9] Pesme, Scott, Loucas Pillaud-Vivien, and Nicolas Flammarion. "Implicit bias of sgd for diagonal linear networks: a provable benefit of stochasticity." Advances in Neural Information Processing Systems 34 (2021): 29218-29230.
[10] Cai, Yuhang, et al. "Large stepsize gradient descent for non-homogeneous two-layer networks: Margin improvement and fast optimization." Advances in Neural Information Processing Systems 37 (2024): 71306-71351.

**Experimental Designs Or Analyses:**

Empirical observations are systematically integrated throughout the paper to substantiate the theoretical assertions.

**Methods And Evaluation Criteria:**

The empirical methods appear to be sound and provide a valid basis for the verification of their theoretical results.

**Other Comments Or Suggestions:**

See questions.

**Other Strengths And Weaknesses:**

The paper is generally well-written and easy to follow. Heuristic claims are frequently validated with empirical observations, and the theorems are rigorously mathematically constructed.

**Questions For Authors:**

- I am somewhat perplexed by the reasoning presented in Section 5.2. It is asserted that whenever $C(\theta) \leq 0$, the corresponding layer imbalance of both GD and SGD is inevitably bound to increase. However, as previously mentioned by the authors, $C(\theta)$ continues to rise even when $C(\theta)$ is positive, which is clearly not elucidated by Theorem 5.6. Could you provide a possible explanation for this discrepancy? In this context, how does the batch size $B$ and the step size $\eta$ influence the progressive sharpening process?
- The balanced condition (Assumption 4.5) appears to be quite stringent. Could you elucidate the underlying rationale for this assumption?
- As previously mentioned, how does progressive sharpening impact the generalization and the model’s quality after training? Are there any practical implications of the findings to the actual training of neural networks?

**Relation To Broader Scientific Literature:**

The studied phenomenon is closely related to the ongoing research on the edge-of-stability phenomenon and the potential implicit regularization effect of SGD. The minimalist examples also appear to have drawn inspiration from several previous works [1, 2], which should be included in the literature review (see also the next question).

[1] Gunasekar, Suriya, et al. "Implicit bias of gradient descent on linear convolutional networks." Advances in neural information processing systems 31 (2018).
[2] Woodworth, Blake, et al. "Kernel and rich regimes in overparametrized models." Conference on Learning Theory. PMLR, 2020.

**Theoretical Claims:**

The proofs are grounded in conventional eigendecompositions of the data matrix, which appears sound to me.

---

> ### Author Rebuttal · Authors · 2025-04-01
>
> We appreciate your thoughtful comments. Below, we address your recommendations and questions.
>
> ## Comprehensive Literature Review & Implications
>
> Thank you for highlighting relevant literature that we initially omitted. We agree that incorporating a more comprehensive review will strengthen our work.
>
> - Linear diagonal networks: Our minimalist model shares similarities with diagonal linear networks in a sparse regression setting. [1] showed that SGD leads to solutions with better generalization than GD. Similarly, our results show that SGD induces less progressive sharpening than GD, leading to lower sharpness at convergence. Considering that lower sharpness correlates with improved generalization in diagonal linear networks [2], they both unveil how stochasticity can help generalization.
> - Potential practical implications on learning rate scheduling: The study by [6] highlights that the catapult mechanism contributes positively to model generalization, and catapults can be induced by designing a proper learning rate schedule. In light of this, predicting sharpness evolution can offer practical value when designing such schedulers.
> - Connection between sharpness and generalization: SAM [3] was introduced under the hypothesis that minimizing sharpness improves generalization. Moreover, GD with a large learning rate has been shown to implicitly find flatter solutions [4], which often generalize better than those obtained with small learning rates. While these works suggest a correlation between sharpness and generalization, [5] showed that this relationship is data-dependent.
>
> The link between sharpness and generalization remains an active research area, though our study focuses on optimization dynamics rather than generalization. Nevertheless, our findings on progressive sharpening may relate to generalization, as larger learning rates and smaller batch sizes yield less progressive sharpening, effectively regularizing toward lower-sharpness solutions. This also connects our analysis to the broader literature on the implicit bias of GD and SGD, including the works mentioned by the reviewer.
>
> ## Increase of $C(\theta)$ when it is positive
>
> We acknowledge that Theorem 5.6 may have caused confusion, especially when $C(\theta) > 0$. Our simplified version emphasized the effect of batch size and learning rate while avoiding complexity. Below is the complete result.
>
> In Appendix A.5, instead of applying the Cauchy-Schwarz inequality in line 847 (Eq. 23), we derive the following exact characterization:
>
> For SGD,
>
> $$
> \\begin{align*}
> &\mathbb{E}\_P[C(\theta_\text{SGD}^+)] - C(\theta) \\\\
> &= \underbrace{\frac{\eta^2}{N^2} [ - \Psi_1(\theta) C(\theta) + \Omega_1(\theta) ]}\_{C(\theta_\text{GD}^+) - C(\theta)} + \frac{\eta^2(N-B)}{BN^2(N-1)}[- (\Psi_2(\theta) - \Psi_1(\theta)) C(\theta) + (\Omega_2(\theta) - \Omega_1(\theta))]
> \\end{align*}
> $$
>
> We denote, $\Omega_1(\theta) \triangleq \sum_{i} \sum_{j>i} \left[ \sigma_i (z(\theta)^\top  e_i) o_j - \sigma_j (z(\theta)^\top  e_j) o_i \right ]^2$, $\Omega_2(\theta) \triangleq N\sum_{i} \sum_{j>i} \lVert \sigma_i(z(\theta) \odot e_i) o_j - \sigma_j(z(\theta) \odot e_j) o_i \rVert^2$, and use $\Psi_1$, $\Psi_2$ from our theorem statement.
>
> Then, $C(\theta)$  increases for GD ($C(\theta_\text{GD}^+) - C(\theta) \geq 0$) if and only if
>
> $$
> C(\theta) \leq  \frac{\Omega_1(\theta)}{\Psi_1(\theta)}=: T_1(\theta).
> $$
>
> Moreover, $C(\theta)$  increases for SGD faster than GD ($\mathbb{E}\_P [C(\theta\_\text{SGD}^+)] - C(\theta\_\text{GD}^+) \geq 0$) if and only if
>
> $$
> C(\theta) \leq  \frac{\Omega_2(\theta)- \Omega_1(\theta)}{\Psi_2(\theta) - \Psi_1(\theta)}=: T_2(\theta).
> $$
>
> We numerically verified that $C(\theta) \leq T_1(\theta)$ and $C(\theta) \leq T_2(\theta)$ holds throughout the GD/SGD training. The experiment results in PDF [[link]](https://anonymous.4open.science/r/understand_progressive_sharpening-E2F5/Experimental_Results_47Zn.pdf) illustrate how $C(\theta)$ increases even when positive. Moreover, we observe the same dependence on learning rate $\eta$ and batch size $B$—specifically, $C(\theta)$ increases more with a larger learning rate or a smaller batch size, as long as $C(\theta) \leq T_1(\theta)$ and $C(\theta) \leq T_2(\theta)$ hold.
>
> We will incorporate this discussion into the revised manuscript.
>
> ## Underlying rationale for balanced condition
>
> Please refer to our response to Reviewer smTr.
>
> ---
>
> [1] Implicit Bias of SGD for Diagonal Linear Networks: a Provable Benefit of Stochasticity, NeurIPS 2021
>
> [2] Implicit Bias of the Step Size in Linear Diagonal Neural Networks, ICML 2022
>
> [3] Sharpness-Aware Minimization for Efficiently Improving Generalization, ICLR 2021
>
> [4] Gradient Descent on Neural Networks Typically Occurs at the Edge of Stability, ICLR 2021
>
> [5] A Modern Look at the Relationship between Sharpness and Generalization, ICML 2023
>
> [6] Catapults in SGD: spikes in the training loss and their impact on generalization through feature learning, ICML 2024

---

> > ### Comment · Reviewer_47Zn · 2025-04-02
> >
> > I would like to express my gratitude to the reviewers for their thoughtful responses, which have partially clarified my concerns. In light of their feedback, I have revised my recommendation from 2 (Weak Reject) to 3 (Weak Accept). Nevertheless, I believe that certain modifications are necessary to enhance the readability and presentation of this paper.

---

> > > ### Author Response · Authors · 2025-04-09
> > >
> > > We appreciate your careful consideration of our work and are glad to hear that our response has helped address some of your concerns. We will also make further efforts to improve readability and presentation in the revised version.
> > >
> > > In addition to incorporating the author-reviewer discussion in our revision, we plan to add several results that will strengthen the paper. These include:
> > >
> > > - For $D=2$, under a standard random initialization scheme, we have analyzed the (expected) initial sharpness and sharpness at convergence. We will include the resulting quantitative characterization of the “sharpness increase” over the trajectory of training. We also provide a [[link]](https://anonymous.4open.science/r/ups-BCF4/Comment_to_47Zn.pdf) to the corresponding empirical results, which show that the expected sharpness increment closely matches the actual sharpness increase.
> > > - As part of our literature review, we will also include a discussion of [1], contextualizing it relative to our Theorem 5.6. They consider a scalar linear network with loss $\mathcal{L}(\boldsymbol w) \triangleq \frac{1}{2} \left( \prod_{i=1}^D \boldsymbol w_i - 1 \right)$, for depth $D \in \mathbb{N}$ and weights $\boldsymbol w \in \mathbb{R}^D$.  Their Theorem 3.2 shows that gradient descent does not increase the sharpness of the gradient flow solution initialized at GD iterates (referred to as GFS sharpness). Similarly, our Theorem 5.6 and the rebuttal show that $C(\theta)$ increases over time when mild conditions on $C(\theta)$ are satisfied. Together with our Remark 4.4, these results imply that GFS sharpness decreases as training progresses under GD/SGD in our minimalist model.
> > >
> > > We sincerely thank you again for your valuable feedback and helpful suggestions.
> > >
> > > ---
> > >
> > > [1] Kreisler et al., Gradient Descent Monotonically Decreases the Sharpness of Gradient Flow Solutions in Scalar Networks and Beyond, ICML 2023.

---

### Decision · Program_Chairs · 2025-05-01

**Decision:**

Accept (poster)

**Comment:**

This paper investigates the phenomenon of progressive sharpening in deep neural network training using a minimalist model. The work provides theoretical analysis and empirical validation for how sharpness evolves during training, and how problem parameters like depth, data difficulty, learning rate, and batch size influence this evolution. There is strong consensus among the reviewers that the paper makes a valuable contribution to understanding sharpness dynamics — a actively studied topic in optimization and deep learning theory. Some reviewers pointed out that the analysis is primarily limited to linear models or highly simplified settings. While this is acknowledged by the authors, and some preliminary attempts at nonlinear analysis were discussed, further exploration in this direction would strengthen future work. Given the overall reviewer consensus, the quality of the technical contributions, the clarity of the presentation, and the constructive engagement during the rebuttal phase, I recommend acceptance.